# HSG-12M: A Large-Scale Benchmark of Spatial Multigraphs from the Energy Spectra of Non-Hermitian Crystals

**Xianquan Yan**[1,2*] **Hakan Akgün**[1] **Kenji Kawaguchi**[2] **N. Duane Loh**[1,3,4†] **Ching Hua Lee**[1‡]

Department of {[1]Physics, [2]Computer Science, [3]Biological Sciences}, National University of Singapore
[4]NUS Centre for Bioimaging Sciences, National University of Singapore

## ABSTRACT

AI is transforming scientific research by revealing new ways to understand complex physical systems, but its impact remains constrained by the lack of large, high-quality domain-specific datasets. A rich, largely untapped resource lies in non-Hermitian quantum physics, where the energy spectra of crystals form intricate geometries on the complex plane—termed as *Hamiltonian spectral graphs*. Despite their significance as fingerprints for electronic behavior, their systematic study has been intractable due to the reliance on manual extraction. To unlock this potential, we introduce **Poly2Graph**[1]: a high-performance, open-source pipeline that automates the mapping of 1-D crystal Hamiltonians to spectral graphs. Using this tool, we present **HSG-12M**[2]: a dataset containing 11.6 million static and 5.1 million dynamic Hamiltonian spectral graphs across 1401 characteristic-polynomial classes, distilled from 177 TB of spectral potential data. Crucially, HSG-12M is the first large-scale dataset of *spatial multigraphs*—graphs embedded in a metric space where multiple geometrically distinct trajectories between two nodes are retained as separate edges. This simultaneously addresses a critical gap, as existing graph benchmarks overwhelmingly assume simple, non-spatial edges, discarding vital geometric information. Benchmarks with popular GNNs expose new challenges in learning spatial multi-edges at scale. Beyond its practical utility, we show that spectral graphs serve as universal topological fingerprints of polynomials, vectors, and matrices, forging a new algebra-to-graph link. HSG-12M lays the groundwork for data-driven scientific discovery in condensed matter physics, new opportunities in geometry-aware graph learning and beyond.

## 1 INTRODUCTION

The integration of AI into scientific research is transforming how complex physical systems are understood (Carleo et al., 2019). However, this transformation is often hindered by a shortage of high-quality, domain-specific datasets, particularly in physical sciences. Recent breakthroughs in protein folding (Jumper et al., 2021; Varadi et al., 2024), materials discovery (Merchant et al., 2023; Li et al., 2025), and many-body physics (Yang et al., 2024; Torlai et al., 2018) underscore how well-curated scientific datasets can unlock AI's full potential, enabling discoveries that would otherwise remain inaccessible.

A rich resource lies in the *Hamiltonian spectral graph*, a fascinating and diverse object emerging from recent advances in non-Hermitian physics. Recent advances have shown that the energy spectrum of one-dimensional crystals under open boundary conditions[3] forms arcs and loops on the complex energy plane. These spectral loci can be naturally represented as spatial graphs embedded

---

[*]yanx@u.nus.edu

[†]duaneloh@nus.edu.sg

[‡]phylch@nus.edu.sg

[1]https://github.com/sarinstein-yan/Poly2Graph

[2]https://github.com/sarinstein-yan/HSG-12M

[3]To be precise, it is 1-D crystal (lattice) Hamiltonian, under open boundary conditions (OBC), in the thermodynamic limit (i.e. large-size limit, the length of the lattice $\to \infty$).

in the two-dimensional complex plane ($\mathbb{C}$-plane). Moreover, these *spectral graphs* (Tai & Lee, 2023; Lin et al., 2023; Xiong & Hu, 2023; Wang et al., 2024) serve as fingerprints with far more intricate structures than conventional topological signatures for electronic behavior (e.g., $\mathbb{Z}/\mathbb{Z}_2$ invariants, Chern number (Hasan & Kane, 2010)). Figure 2&A4 show examples of these graphs, featuring a kaleidoscope of nontrivial edge geometries and multiplicities that form diverse patterns beyond existing graph datasets.

Despite their theoretical significance, spectral graph extraction has traditionally relied on manual plotting and visual inspection—an approach limited to toy examples and small-scale investigations. In the absence of any automated workflow or large curated dataset, its systematic studies have remained out of reach.

To overcome the reliance on manual inspection, we developed `Poly2Graph`: an open-source pipeline that combines algebraic geometry, non-Bloch band theory, and morphological image processing to fully automate spectral graph extraction. By delivering unprecedented speed and memory efficiency, `Poly2Graph` enabled us to introduce **HSG-12M** (Hamiltonian Spectral Graphs, 12 Million). This dataset distills 177 TB of spectral potential data into 12M spatial multigraph representations (256 GB), spanning 1401 characteristic polynomial classes. Each graph is derived from the *energy band structure* of a crystal Hamiltonian, encoding the complex energy spectrum's full geometry[4]. In condensed matter physics, energy band structure is a fundamental concept, key to understanding insulators, conductors, phase transitions, electron dynamics, and system symmetries. We additionally provide 5.1M *temporal spatial graphs* capturing continuous deformations of spectral graphs.

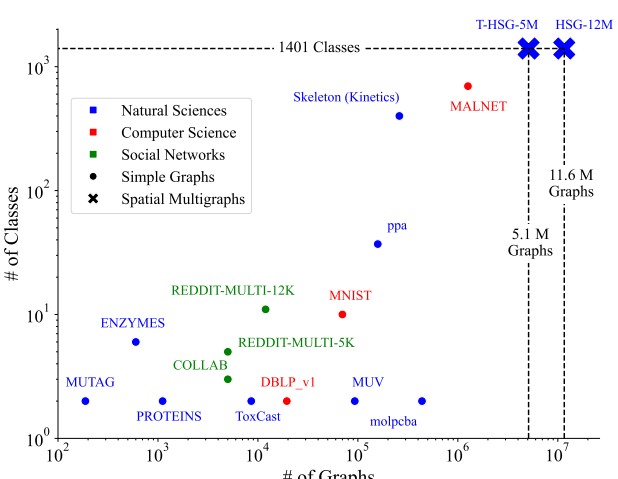

Figure 1: Number of graphs v.s. number of classes in HSG-12M compared to other graph-classification datasets. HSG-12M is the only large-scale *multigraph* (i.e. unlike *simple* graph that only allows one edge between any node pair) dataset, with exceptional class diversity that even exceeds all other simple graph datasets. T-HSG-5M holds temporal spatial multigraphs. Table A4 lists comprehensive comparison against 45 other datasets.

Moreover, the Hamiltonian spectral graph is inherently a *spatial multigraph*—a type of graph resource fundamentally different from existing datasets. Graph representation learning (Hamilton et al., 2017; Corso et al., 2024) has emerged as a powerful paradigm for modeling structured data, yet a critical limitation persists: virtually all public benchmarks treat data as *simple* graphs, allowing at most one edge between any node pair (Hu et al., 2020b; Ranveer & Hiray, 2015; Freitas et al., 2021). Even when source data contains multi-edges, these are typically aggregated into a single attributed edge, discarding crucial spatial information. Due to the absence of such datasets, the development of methodologies for spatial multigraphs has been severely hindered.

In contrast, many real-world networks are spatial multigraphs, i.e., graphs embedded in a metric space, where entities may connect through multiple distinct geometrically meaningful paths. Such **spatial graphs** or **geometric networks** (Barthélemy, 2011; Guo et al., 2021) naturally arise in urban street networks (Kujala et al., 2018; Boeing, 2019), biological neural networks (Weiner et al., 2010; Di Martino et al., 2014), protein structures (Anand & Huang, 2018; Guo et al., 2020), and beyond (Bullmore & Sporns, 2009; Caldarelli, 2007). When the properties of interest include both connectivity *topology* and connection *geometry*, collapsing intrinsically distinct multi-edges results in critical information loss.

---

[4]In mathematical terms, the energy spectrum refers to the set of eigenvalues of the Hamiltonian matrix. Within this work, energy band structure can be considered the same as energy spectrum.

HSG-12M addresses this critical gap in graph representation learning, being the first large-scale database of spatial multi-graphs, grounded in non-Hermitian physics. Furthermore, the temporal component establishes the first large-scale temporal (dynamic) spatial graph dataset for graph-level tasks. Our benchmark with popular GNNs expose new challenges in learning spatial multi-edges at scale.

In summary, this work introduces a large-scale spatial multigraph dataset and methodology at the intersection of condensed-matter physics and graph representation learning. Our key contributions include:

1. **Open-source, High-performance, End-to-End Automated Pipeline.** We release Poly2Graph that can map arbitrary 1D Hamiltonians to spectral graphs, providing the first automated tool to study spectral graphs with high speed and efficiency. Poly2Graph not only enables us to produce HSG-12M, but also empowers researchers to generate custom spectral graph datasets, vastly expanding the possibilities for future study.
2. **Large Scale & Exceptional Class Diversity.** 11.6 million static and 5.1 million dynamic spatial multigraphs spanning 1401 classes, distilled from 177 TB of spectral potential data. HSG-12M is the first large-scale multigraph dataset for graph-level tasks (Figure 1&A6) with class diversity exceeding all simple graph datasets.
3. **Novel Graph Type & New Challenges.** Spatial multigraphs simultaneously capture connection topology with edge multiplicity preserved and geometry of multiedges & nodes in the embedding space. This first large-scale collection introduces new challenges for developing geometry-aware graph learning algorithms.
4. **New Domain, Physics-grounded, Universal Relevance.** Spectral graphs are firmly grounded in theories of non-Hermitian quantum physics, introducing an abundant database *from* and *for* an entirely new domain. Physically, spectral graph encapsulates information about quantum state dynamics and topology, Hamiltonian symmetry class, response strength, quantum sensing capability, and more. Thus our database paves the way for accelerating discovery of exotic phases, enabling rational design of materials with desired quantum properties.

    Additionally, we identify *Hamiltonian spectral graph* as a new class of topological object deserving attention in its own right—in section 5 we show that vectors, matrices, and polynomials, be they real or complex, admit spectral graphs as their topological fingerprint, bridging graph and ubiquitous algebra objects.

## 2 POLY2GRAPH: AUTOMATING SPECTRAL GRAPH EXTRACTION

Poly2Graph is high-performance and the first *end-to-end* automated pipeline that converts an arbitrary one-dimensional crystal Hamiltonian into its *spectral graph* representation. It operationalizes the mathematical construction reviewed in appendix B by integrating non-Bloch band theory, algebraic geometry, and morphological image processing.

Full algorithmic details are deferred to appendix C. Here we highlight the design choices that make Poly2Graph $10^5 \times$ **faster** and more memory-efficient than the best available code (empirical benchmark in appendix C.5), thereby enabling the construction of HSG-12M.

**From Hamiltonians to Characteristic Polynomials.** Poly2Graph initializes with either a Bloch Hamiltonian matrix $H(z)$ or its characteristic polynomial. For a $s$-band tight-binding crystal chain, the Bloch Hamiltonian reads

$$\boldsymbol{H}(z) = \sum_{j=-p}^{q} \boldsymbol{T}_j \, z^j, \qquad z = e^{ik}, \; k \in [-\pi, \pi), \; \boldsymbol{T}_j \in \mathbb{C}^{s \times s}, \tag{1}$$

Its open-boundary spectrum solely depends on the roots of the Laurent *characteristic polynomial*:

$$P(z, E) := \det \left[ \boldsymbol{H}(z) - E \, \boldsymbol{I}_s \right] = \sum_{n=-p}^{q} a_n(E) \, z^n. \tag{2}$$

We choose an energy domain $\Omega \subset \mathbb{C}$ in the complex energy plane (the minimal square) that encloses the entire spectral graph $\mathcal{G}$. By default, Poly2Graph estimates $\Omega$ by diagonalising a small real-space

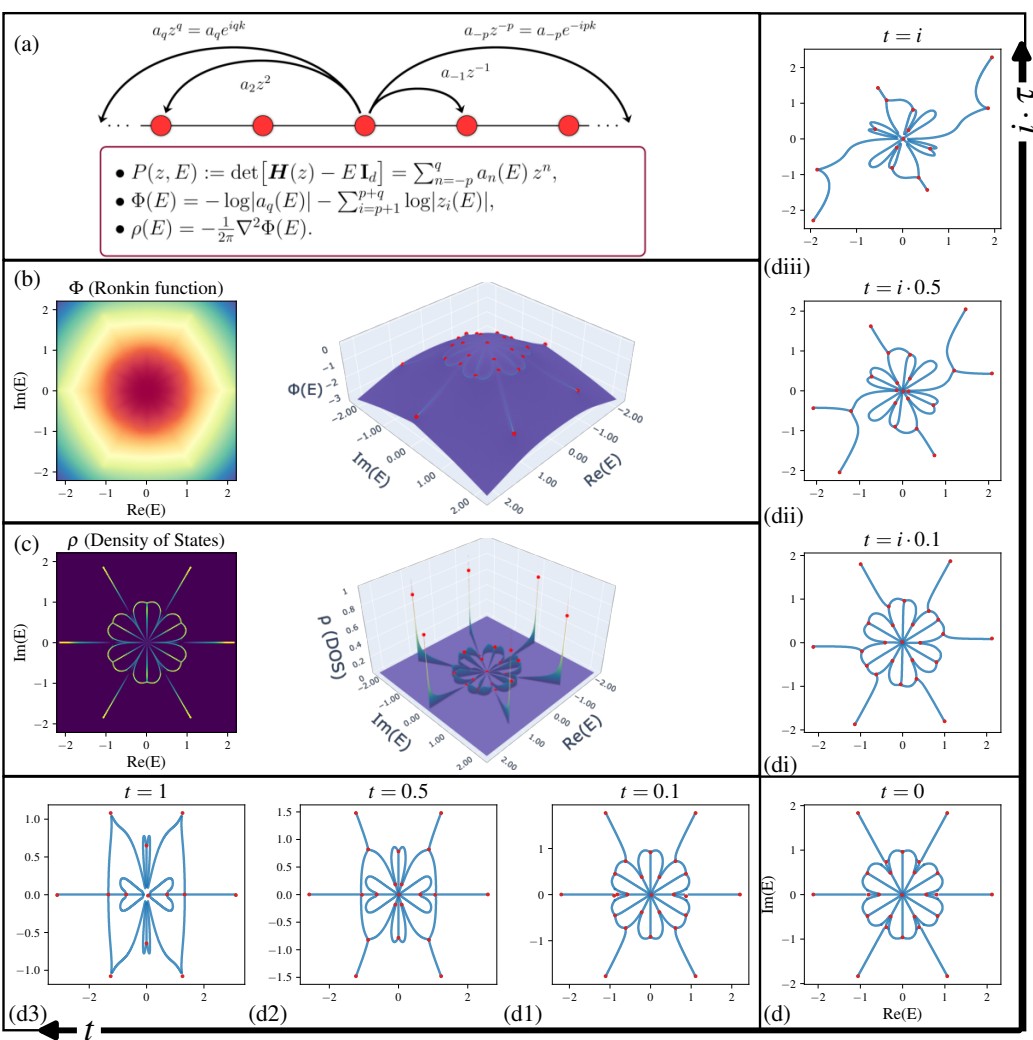

Figure 2: **Poly2Graph pipeline.** (a) Starting from a 1-D crystal Hamiltonian $H(z)$ in momentum space—or, equivalently, its *characteristic polynomial* $P(z, E) = \det[\boldsymbol{H}(z) - E\boldsymbol{I}]$. The crystal's open-boundary spectrum solely depends on $P(z, E)$. (b) The *spectral potential* $\Phi(E)$ (Ronkin function) is computed from the roots of $P(z, E) = 0$, following recent advances in non-Bloch band theory (Tai & Lee, 2023; Xiong & Hu, 2023; Wang et al., 2024). (c) The density of states $\rho(E)$ is obtained as the Laplacian of $\Phi(E)$. (d) The spectral graph extracted from $\rho(E)$ via a morphological computer-vision pipeline. Varying the coefficients of $P(z, E)$ produces diverse graph morphologies in the real domain (d1)-(d3) and imaginary domain (di)-(diii).

Hamiltonian with $L = 40$ unit cells, though users may optionally specify a custom domain and resolution. The resultant domain $\Omega$ is discretized into a grid of complex energy values. In HSG-12M, we used a default resolution of 256 (initial) × 4 (adaptive enhancement) = 1024 points along each axis.

For each sample energy $E \in \Omega$, we solve the roots $\{z_i(E)\}$ of $P(z, E) = 0$ (treating $E$ as constant) and then sort them by magnitude $|z_1(E)| \leq |z_2(E)| \leq \cdots \leq |z_{p+q}(E)|$. This is the *computational bottleneck* in naive approaches—solving roots of an enormous batch (by default $1024^2 \approx 10^6$) of polynomials for every grid point is extremely expensive. To tame this bottleneck, we implement a custom, optimized root-solver based on Frobenius companion matrices and parallel eigen-solvers with auto-backend detection for optional GPU acceleration, cutting wall-time from hours to milliseconds.

**Spectral Potential & Density-of-States (as 2D Images).** With the roots $\{z_i(E)\}$ computed, we leverage non-Bloch band theory (Tai & Lee, 2023; Xiong & Hu, 2023; Wang et al., 2024) and reliably compute the *spectral potential*[5] as:

$$\Phi(E) = -\log|a_q(E)| - \sum_{i=p+1}^{p+q} \log|z_i(E)|, \tag{3}$$

where $a_q(E)$ is the leading coefficient of the characteristic polynomial. The Laplacian of this potential yields the *Density of States* (DOS):

$$\rho(E) = -\frac{1}{2\pi}\nabla^2\Phi(E). \tag{4}$$

where $\nabla^2 = \partial^2_{\text{Re}E} + \partial^2_{\text{Im}E}$. Physically, $\rho(E)$ quantifies the number of eigenstates per unit area at energy $E$ in the complex plane. $\rho(E) > 0$ **region traces out** *spectral graph* (Figure 2c). Geometrically, since DOS is defined as the second derivative (curvature), the spectral graph in other words corresponds to the "ridges" of the spectral potential landscape (Figure 2b).

In addition, we exploit inherent symmetries in special polynomials. For example, the complex conjugate root theorem guarantees that if $P(z, E)$ has purely real coefficients, its spectral graph is symmetric about the real axis; similarly, purely imaginary coefficients produce symmetry about the imaginary axis. By calculating only the relevant half-plane and mirroring the results, we reduce computation time by up to 50% for qualifying polynomials.

**Image-to-Graph Routine.** To extract the spectral graph from the DOS image, we binarize the DOS and apply skeletonization to obtain a one-pixel-wide graph skeleton.

However, we face a resolution-computation tradeoff: insufficient resolution results in lost topological features (small loops, adjacent nodes, etc), while uniform high-resolution calculation across the entire energy domain $\Omega$ is prohibitively expensive, especially since the spectral graph typically occupies only a small fraction of this area.

We resolve this challenge with a *two-stage adaptive resolution* approach:

1. *Coarse identification*: We first compute the DOS on a moderately-resolved grid ($256 \times 256$), threshold to binarize the image, and perform morphological dilation with a $2 \times 2$ disk. This generates a conservative binary mask that envelops the spectral graph while excluding approximately 95-99% of non-contributive region.

2. *Refined calculation*: Within only the masked region, we subdivide each pixel into an $m \times m$ grid (default $m = 4$), recalculating the spectral potential and DOS at this higher resolution. This targeted approach achieves an effective resolution of $1024 \times 1024$ while computing just 1-5% of the grid points.

The high-resolution DOS is then re-binarized and subjected to iterative morphological thinning operations (Lee et al., 1994) until a one-pixel-wide skeleton remains, preserving topological features ready to be distilled into a graph representation.

For the final graph extraction, we analyze this skeleton to identify three point types: (1) junction nodes where three or more paths intersect, (2) leaf nodes where paths terminate, and (3) edge points along continuous segments. The output is an `NetworkX MultiGraph` object. Crucially, each edge stores its complete geometric information as an ordered sequence of $(\text{Re}(E), \text{Im}(E))$ coordinates, preserving not just connectivity but the exact shape of each spectral curve.

**Quality Assurance and Limitations.** We validated Poly2Graph on hundreds of characteristic polynomials, by visually checking that the spectral graph from Poly2Graph agrees with the energy spectrum from exact diagonalization. In rare complicated cases, numerical instabilities can still arise close to the junction nodes whose surrounding edges have extremely low DOS (see appendix C.6). Poly2Graph will attempt to mitigate such cases by merging nearby nodes and contracting edges shorter than a predefined tolerance.

---

[5]The spectral potential is also known as the Ronkin function, an algebro-geometric property of $P(z, E)$ (Wang et al., 2024)

**Open-Source Release and Broader Impact.** Poly2Graph[1] is released under the MIT licence. See a tutorial in appendix G. Poly2Graph establishes a turn-key mechanism for translating linear operators into machine-learning-ready graphs, bridging condensed matter physics and graph representation learning. The same principle extends to any vector, matrix, and univariate/bivariate polynomial, opening an new "algebra-as-graph" perspective, broadening the applicability of Poly2Graph to a wide range of other areas (section 5, appendix F).

## 3   HSG-12M DATASET DESCRIPTION

The speed and memory efficiency of Poly2Graph make large-scale spatial multigraph research practical for the first time. Figure 1&A6 illustrate the scale of HSG-12M, showing #graphs vs. #classes and #graphs vs. total #nodes relative to other graph classification datasets. To our knowledge, HSG-12M is not only the largest dataset by number of graphs and classes but also the only large-scale spatial multigraph dataset available.

Moreover, each graph class corresponds a particular Hamiltonian family (hopping pattern). A well-trained graph neural network could therefore potentially serve as a surrogate model to predict material structure from a desired spectral graph, thereby facilitating inverse design of materials with targeted properties—e.g. design of acoustic metamaterial, electrical circuit, or photonic crystal with desired spectral response.

In Table A4 we provide a comprehensive comparison with existing graph datasets and benchmarks. Most prior popular graph-classification datasets are non-spatial, simple graphs. A few are spatial, e.g., some superpixels and molecular graphs have node coordinates in 2D / 3D, but their edges remain an abstract connection defined by adjacency. HSG-12M uniquely provides *spatial multigraphs*, where the intricate geometric structure of multi-edges carries essential information that cannot be simplified without loss. The most relevant resource, OpenStreetMap (Boeing, 2019) is much smaller, less diverse, and lacks associated ML tasks in comparison.

Furthermore, while temporal graph datasets exist (Huang et al., 2023), they typically focus on node/edge-level tasks or involve small numbers of graphs and classes. Our T-HSG-5M represents the first large-scale collection of dynamic graphs for graph-level tasks, capturing the continuous evolution of spectral graphs over Hamiltonian parameters.

**Data Format and Accessibility.** To maximize accessibility and flexibility, we release HSG-12M under a permissive **CC BY 4.0** license. The dataset is publicly available via `Dataverse` (Yan et al., 2025a). Users can download the full dataset or select specific subsets using the code provided at `https://github.com/sarinstein-yan/HSG-12M`. In companion, we release an auxiliary package `HSG-12M`[2] for effortless data handling, benchmark reproduction, custom featurization and dataset generation, interactive tutorial, and more.

The dataset comprises 1401 separate Python `npz` files, each containing graphs from one class with relevant metadata. Raw files use `NetworkX MultiGraph` format, preserving full node and edge geometry— ❶ *Node attributes:* complex coordinates, spectral potential, and density of states. ❷ *Edge attributes:* edge length (also serving as weight), coordinate sequences along the edge, average spectral potential and average DOS over the edge.

We provide this descriptive format because representation learning on spatial multigraphs remains nascent, with no agreed-upon standard for representing continuous multi-edge geometry. Rather than imposing a particular featurization, we encourage researchers to explore various approaches, e.g., treating edge curves as sequences, computing summary features like curvature, or developing novel and more sophisticated neural network-based representations. Moreover, the attribute-rich format here aids interpretability and is relevant to researchers interested in the underlying physics rather than solely ML.

That said, for convenience, we propose our own featurization scheme and include a conversion API that transforms raw data into PyTorch Geometric (PyG) datasets for graph classification benchmarking. Particularly, to manage the inhomogeneity of edge coordinates and make the spectral graphs compatible with standard GNN input, our reference conversion uses *fixed-sized, direction-ignorant edge summary features* (appendix E.1): length, the straight-line distance between start and end nodes, middle point coordinates, average spectral potential, and average DOS along the edge.

Table 1: Key statistics of the HSG dataset variants. #Graphs: number of graphs; #Classes: number of classes; Ratio: the #Graphs of the largest class / #Graphs of the smallest class; Temporal: whether the graphs are temporal. All other five datasets are derived from HSG-12M; thus all datasets are **spatial** and irreducibly **multigraph**. `HSG-topology` contains non-isomorphic graphs in each class and is the only *imbalanced* dataset; `T-HSG-5M` is the *temporal* spectral graph collection; the rest four teal-colored datasets are balanced, static datasets.

| Name | #Graphs | #Classes | Ratio | Temporal |
|------|---------|----------|-------|----------|
| `HSG-one-band` | 198,744 | 24 | 1.0 | - |
| `HSG-two-band` | 2,277,275 | 275 | 1.0 | - |
| `HSG-three-band` | 9,125,662 | 1102 | 1.0 | - |
| `HSG-topology` | 1,812,325 | 1401 | 660.2 | - |
| `T-HSG-5M` | 5,099,640 | 1401 | 1.0 | ✓ |
| `HSG-12M` | 11,601,681 | 1401 | 1.0 | - |

**Dataset Construction.** Graphs are grouped by different Hamiltonian families (i.e. characteristic polynomial classes) as detailed in appendix B. We systematically sample polynomial classes while respecting mathematical symmetries to avoid spurious abundance. For instance, if a polynomial exhibits $z$-reciprocity—i.e. $P(z) = z^{p+q}P(1/z)$—this reciprocal transformation physically means flipping the crystal chain from left to right, which leaves the spectrum unchanged and yields the same spectral graph.

Specifically, we start from a base polynomial with hopping range $p + q$ and $s$ energy bands:

$$\hat{P}(z, E) = -E^s + z^{-p} + z^q .\qquad(5)$$

We then set the degree of $E^k : k \in \{0, 1, \dots, s-1\}$ for each $z^i : i \in \{-p+1, \dots, q-1\}$. Subsequently, we assign two free coefficients $(a, b)$ to two chosen monomials $z^j : j \in \{-p+1, \dots, -1, 1, \dots, q-1\}$—excluding $z^0$, since varying the constant term only raise or lower the entire spectral potential landscape, no effect exerted on the spectral graph.

For example, a two-band polynomial with $p = 3$ and $q = 3$ may take the form:

$$\hat{P}(z, E) = -E^2 + z^{-3} + \left(a\, z^{-1} + b\, E\, z + E\, z^2\right) + z^3, \quad a, b \in \mathbb{C} .\qquad(6)$$

Under such a sampling scheme, we iterate over all combinations for one-band to three-band polynomials, with hopping ranges varied from four to six. This range has well covered all realistic 1D tight-binding crystals (typically $p + q \leq 4$ and less than three bands).

After removing symmetric redundancy, we collect 24 one-band classes, 275 two-band classes, and 1102 three-band classes, amounting to a total of 1401 unique classes.

Finally, we vary the two free coefficients from $-10 - 5i$ to $10 + 5i$ respectively, with 13 real and 7 imaginary values, yielding $(13 \times 7)^2 = 8281$ samples per class.

**Dataset Variants.** We provide six datasets tailored to different research needs.

`HSG-one-band`: Small-to-medium scale, the collection of all one-band polynomials, balanced subset with 198,744 graphs across 24 classes. These graphs in this subset display simpler patterns ideal for rapid prototyping and algorithm validation.

`HSG-two-band` and `HSG-three-band`: Medium-to-large scale, the collection of all two-band and three-band polynomials respectively, balanced datasets with increasing complexity, containing 2.3M and 9.1M graphs across 275 and 1,102 classes, respectively.

`HSG-12M`: The complete dataset spanning all 1,401 classes with balanced sampling, totaling 11.6M static graphs, designed for large-scale challenge.

`HSG-topology`: An imbalanced subset preserving only *topologically* distinct (i.e. non-isomorphic) graphs within each class. This filtered dataset removes isomorphic duplicates, resulting in highly skewed class distributions (max class size ratio 660.2), useful for analyzing spectral graph topology diversity and benchmarking graph algorithms on imbalanced datasets.

`T-HSG-5M`: Our temporal multigraph collection capturing continuous spectral graph evolution. As shown in figure 2d, varying either the real or imaginary part of a coefficient in the characteristic polynomial continuously morphs the *geometry* of the spectral graph; at certain transition points, one can observe the graph *topology* changes discontinuously. For each class, we collect all sequences of the variation in real (or imaginary) parts of one free coefficient, adding up to 5.1M temporal graphs across 1401 classes. T-HSG-5M is suitable for evaluating temporal graph-level tasks such as temporal extrapolation and classification on early sequences. Functionality to select any desired sequence or subset is provided in the package.

## 4 BENCHMARKING RESULTS

To assess the capabilities of existing graph learning methods on the new challenges introduced by our HSG datasets, particularly their spatial nature, edge multiplicities, class imbalance, and scale, we benchmark popular GNNs and discuss the implications.

**Baseline Models.** We benchmark eight popular graph neural networks (GNNs)—GCN (Kipf & Welling, 2017), GAT (Veličković et al., 2018), GATv2 (Brody et al., 2022), GIN (Xu et al., 2019), GINE (Hu et al., 2019), MF (Duvenaud et al., 2015), CGCNN (Xie & Grossman, 2018), and Graph-SAGE (Hamilton et al., 2017).

**Experimental Setup.** Full details supporting reproducibility are provided in appendix E.2—including data preprocessing and split; model architecture and hyperparameters; optimizer setting and learning rate schedule; hardware and trainer specifics. In particular, to ensure fair comparison, for each dataset we tune each model's convolution hidden dimension to equalize the total number of learnable parameters, and we cap the training budget by max epochs and max steps.

**Evaluation Metrics.** Given the high class diversity, we report Top-1 accuracy, Top-10 accuracy (relevant for scenarios where multiple plausible answers are acceptable), and Macro-averaged $F_1$-score (which weights every class equally and exposes performance on minority classes). Additional evaluation including test loss, Top-5 accuracy, peak GPU memory utilization, throughput are reported in Tables A5-A8.

**Results and Analysis.** The graph-level classification results are presented in Table 2. Seed variance is small across variants, indicating stable training. Additional results and analysis are in appendix E.4. Several observations emerge:

*Performance degrades with task difficulty.* Test metrics decay monotonically from simpler to harder dataset, consistent with increasing graph size, richer multi-edge geometry, more complex isomorphisms, and growing class diversity. Memory usage likewise grows with complexity (e.g., SAGE $\sim$0.067→0.511 MB/graph).

*Edge attributes matter.* Edge-aware GINE consistently outperforms edge-agnostic GIN (e.g., on `HSG-12M`, Accuracy $0.460_{\pm0.025}$ vs. $0.063_{\pm0.031}$), reflecting that multi-edge spatial geometry (length, straight-line distance, midpoint, average potential/DOS) carries irreducible signal.

*Top-$k$ is high—promising signal for inverse design.* On the full dataset which covers all realistic cases, despite moderate Top-1 accuracy, Top-10 accuracy is high (e.g., on `HSG-12M`: SAGE 95.2%, CGCNN 94.8%) and near-saturated on easier subsets (99%+ on `one-band`), enabling retrieval of a small candidate set of Hamiltonian families for downstream expert screening.

*GraphSAGE excels under limited budget and comparable parameter counts.* Under matched parameters ($\leq$4% variance) and fixed budget (`max_epochs`=100, `max_steps`=1000), **GraphSAGE** attains the best performance on all subsets, indicating either stronger inductive bias for spatial multigraphs or better sample/compute efficiency than more expressive baselines under tight budgets.

*Take-away.* With our proposed edge summaries, popular GNNs already capture substantial discrimination; nonetheless, the Top-1 vs. Top-10 gap and degradation at high class diversity suggest that richer edge geometry encoding (e.g., curvature/torsion, higher-order moments, spline/Bezier encodings, or polyline sequences) could potentially improve performance, especially on the more challenging datasets.

Table 2: Graph-level classification results on the HSG dataset variants. Test metrics shown as $\text{mean}_{\pm\text{std}}$ over three random seeds; best result in **Bold**.

| Model | Test Metric | one-band | two-band | three-band | topology | HSG-12M |
|---|---|---|---|---|---|---|
| GCN[53] | Accuracy | $.711_{\pm.010}$ | $.478_{\pm.012}$ | $.337_{\pm.014}$ | $.397_{\pm.009}$ | $.365_{\pm.021}$ |
| | Macro $F_1$ | $.704_{\pm.009}$ | $.465_{\pm.015}$ | $.323_{\pm.013}$ | $.392_{\pm.011}$ | $.349_{\pm.022}$ |
| | Top-10 Acc. | $.999_{\pm.000}$ | $.931_{\pm.005}$ | $.816_{\pm.011}$ | $.825_{\pm.006}$ | $.841_{\pm.015}$ |
| GAT[96] | Accuracy | $.677_{\pm.002}$ | $.462_{\pm.008}$ | $.344_{\pm.012}$ | $.434_{\pm.015}$ | $.365_{\pm.010}$ |
| | Macro $F_1$ | $.671_{\pm.003}$ | $.449_{\pm.007}$ | $.327_{\pm.014}$ | $.431_{\pm.011}$ | $.347_{\pm.010}$ |
| | Top-10 Acc. | $.998_{\pm.000}$ | $.922_{\pm.004}$ | $.825_{\pm.013}$ | $.855_{\pm.010}$ | $.846_{\pm.006}$ |
| GATv2[6] | Accuracy | $.644_{\pm.005}$ | $.444_{\pm.004}$ | $.282_{\pm.030}$ | $.401_{\pm.003}$ | $.351_{\pm.002}$ |
| | Macro $F_1$ | $.635_{\pm.007}$ | $.430_{\pm.005}$ | $.265_{\pm.031}$ | $.397_{\pm.001}$ | $.330_{\pm.002}$ |
| | Top-10 Acc. | $.997_{\pm.001}$ | $.914_{\pm.003}$ | $.765_{\pm.032}$ | $.833_{\pm.005}$ | $.835_{\pm.001}$ |
| GIN[109] | Accuracy | $.799_{\pm.005}$ | $.343_{\pm.084}$ | $.050_{\pm.021}$ | $.095_{\pm.059}$ | $.063_{\pm.031}$ |
| | Macro $F_1$ | $.796_{\pm.006}$ | $.323_{\pm.087}$ | $.030_{\pm.016}$ | $.082_{\pm.060}$ | $.042_{\pm.024}$ |
| | Top-10 Acc. | $1.000_{\pm.000}$ | $.868_{\pm.060}$ | $.295_{\pm.089}$ | $.390_{\pm.148}$ | $.339_{\pm.135}$ |
| GINE[40] | Accuracy | $.764_{\pm.006}$ | $.518_{\pm.049}$ | $.379_{\pm.013}$ | $.533_{\pm.017}$ | $.460_{\pm.025}$ |
| | Macro $F_1$ | $.758_{\pm.007}$ | $.501_{\pm.053}$ | $.362_{\pm.012}$ | $.531_{\pm.011}$ | $.445_{\pm.027}$ |
| | Top-10 Acc. | $1.000_{\pm.000}$ | $.958_{\pm.015}$ | $.872_{\pm.008}$ | $.927_{\pm.008}$ | $.921_{\pm.011}$ |
| MF[22] | Accuracy | $.589_{\pm.012}$ | $.308_{\pm.014}$ | $.271_{\pm.004}$ | $.348_{\pm.012}$ | $.295_{\pm.010}$ |
| | Macro $F_1$ | $.572_{\pm.012}$ | $.287_{\pm.016}$ | $.254_{\pm.001}$ | $.343_{\pm.012}$ | $.274_{\pm.011}$ |
| | Top-10 Acc. | $.997_{\pm.000}$ | $.838_{\pm.019}$ | $.754_{\pm.005}$ | $.793_{\pm.010}$ | $.791_{\pm.012}$ |
| CGCNN[106] | Accuracy | $.796_{\pm.008}$ | $.585_{\pm.029}$ | $.478_{\pm.011}$ | $.566_{\pm.016}$ | $.531_{\pm.004}$ |
| | Macro $F_1$ | $.792_{\pm.010}$ | $.572_{\pm.029}$ | $.464_{\pm.015}$ | $.563_{\pm.018}$ | $.518_{\pm.003}$ |
| | Top-10 Acc. | $1.000_{\pm.000}$ | $.975_{\pm.005}$ | $.923_{\pm.006}$ | $.940_{\pm.005}$ | $.948_{\pm.002}$ |
| GraphSAGE[33] | Accuracy | $\mathbf{.854_{\pm.003}}$ | $\mathbf{.678_{\pm.004}}$ | $\mathbf{.523_{\pm.020}}$ | $\mathbf{.620_{\pm.003}}$ | $\mathbf{.546_{\pm.004}}$ |
| | Macro $F_1$ | $\mathbf{.853_{\pm.002}}$ | $\mathbf{.670_{\pm.005}}$ | $\mathbf{.512_{\pm.020}}$ | $\mathbf{.622_{\pm.001}}$ | $\mathbf{.534_{\pm.005}}$ |
| | Top-10 Acc. | $\mathbf{1.000_{\pm.000}}$ | $\mathbf{.988_{\pm.001}}$ | $\mathbf{.940_{\pm.006}}$ | $\mathbf{.958_{\pm.001}}$ | $\mathbf{.952_{\pm.001}}$ |

## 5 DISCUSSION

**Benchmarking and algorithmic opportunities.** HSG-12M fills four key gaps at once: ❶ it is the first systematic resource of Hamiltonian spectral graphs, ❷ it is the first *large-scale multigraph* dataset, ❸ it is the first *spatial multigraph* resource, retaining edge multiplicity with rich continuous geometry, and ❹ it provides both static and *dynamic* multigraph sequences.

These traits open a suite of tasks that are under-served by current methods: multi-edge featurization, geometry-aware message passing, spatio-temporal prediction, etc. Beyond supervised learning, the dataset is sufficiently large and expandable with Poly2Graph to support topology-conditioned generation and pre-training foundation models for rational inverse-design of materials.

**Universal Relevance of Spectral Graphs.** While HSG-12M is rooted in non-Hermitian band theory, its reach extends well beyond condensed-matter physics.

1. Any **bivariate Laurent polynomial** $P(z, E)$ has a spectral graph.

2. Any **univariate polynomial** $h(z)$ can be viewed as a one-band Bloch Hamiltonian; and any **vector** can be treated as a symmetrised coefficient list of a univariate polynomial.

3. Any **matrix** can be decomposed into a product of one-band Hamiltonian matrices (Ye & Lim, 2015), and thus in general has a *multiset* of spectral graphs (detailed in appendix F)

Hence polynomials, vectors, and matrices all admit spectral graphs as their topological fingerprints. This establishes a universal bridge between algebraic objects and graphs, inviting graph-based methods to problems in linear algebra.

Since the characteristics of most scientific systems can be expressed as the aforementioned algebraic objects, our approach offers a novel analytical lens. This opens up new avenues for research across numerous fields, a direction we are actively exploring and invite the broader community to join.

## 6  CONCLUSION

We present `Poly2Graph`, an open-source, high-performance pipeline that automatically extracts Hamiltonian spectral graphs, and **HSG-12M**, a large-scale dataset of 11.6M static and 5.1M dynamic Hamiltonian spectral graphs. HSG-12M collects physics-grounded data, offering the first large-scale database for *spatial multigraph* with irreducible edge multiplicity and spatial geometry. The construction generalizes to arbitrary polynomials, vectors, and matrices. Our benchmark on popular GNNs indicate the need for methodological advances. We release `Poly2Graph` and HSG-12M under permissive licences and invite the community to build on this resource for new models, tasks, more comprehensive and carefully designed benchmarks, and insights across machine learning, condensed-matter physics, and beyond.

### ACKNOWLEDGMENTS

This research is supported by the Singapore Ministry of Education (MOE) through the Academic Research Fund Tier-II Grants (MOE-T2EP50222-0003 and MOE-T2EP50224-0007), the Tier-I grants (WBS No. A-8002656-00-00 and Award No. T1 251RES2509). Additional support was provided by the Air Force Office of Scientific Research (Award No. FA2386-24-1-4011).

### REPRODUCIBILITY STATEMENT

We release **Poly2Graph** https://github.com/sarinstein-yan/Poly2Graph (section 2, appendix C&G), **HSG-12M** (section 3, appendix D) and an auxiliary dataset package https://github.com/sarinstein-yan/HSG-12M under permissive open-source licences to facilitate reproducibility and further customization. The packages include detailed documentation and tutorials for easy starting, full reproduction, and extension.

### LLM USAGE

We used large language models, such as ChatGPT and Gemini, to aid in the writing of this paper. All generated content has been reviewed and edited by the authors. The authors are solely responsible for any errors or inaccuracies.

### ETHICS STATEMENT

This research was conducted in an ethical manner, with no direct negative societal impact identified from the implementation of Poly2Graph or the creation and benchmark of the HSG-12M dataset. The computation pipeline and dataset are derived from physical and mathematical principles. The authors have no known conflicts of interest or financial incentives beyond standard academic publication.

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

APPENDIX TABLE OF CONTENTS

## A   RELATED WORK

### A.1   GRAPH REPRESENTATION LEARNING, DATASETS, AND BENCHMARKS.

Graph learning has seen a rapid rise in recent years, driven by advances in graph neural networks (GNNs) (Hamilton, 2020; Bronstein et al., 2021; Ma & Tang, 2021; Wu et al., 2022; Corso et al., 2024) and proliferation of datasets and benchmarks (Ranveer & Hiray, 2015; Freitas et al., 2021; Chen & Wang, 2019; Yanardag & Vishwanathan, 2015; Hu et al., 2020b). HSG-12M addresses critical gaps in existing benchmarks by introducing not only the first large-scale spatial multigraph dataset[6], but also one of the largest known graph machine learning datasets and natural science-based datasets. This work sets a new standard in terms of scale and class diversity.

### A.2   GRAPH LEARNING IN MULTIGRAPHS.

In contrast to *simple* graphs, *multi*graphs permit multiple edges between the same pair of nodes. Apart from a handful of exploration on multigraph learning algorithms (Butler et al., 2023; Egressy et al., 2024), progress has been hampered by the absence of large-scale data sources.

In many practical settings, multiple edges are typically collapsed into a single edge—often sacrificing valuable information. This simplification may be acceptable when edge-level details can be represented as aggregated attributes, as is often the case in heterogeneous graphs (Chaari et al., 2022; Youssef et al., 2023), multi-modular models (Said et al., 2024; Ding et al., 2022), or multiplex networks (Horvat & Zweig, 2018).

However, in spatial multigraphs (Barthélemy, 2011), where edges carry rich geometric information such as distances, directions, or physical observable information, such aggregation results in significant information loss. This critical issue has remained underexplored due to the lack of datasets where edge aggregation is inherently infeasible. HSG-12M addresses this gap by providing the first benchmark where capturing both multi-edge relationships and edge geometry is essential.

### A.3   GRAPH LEARNING IN SPATIAL GRAPHS.

A spatial (or geometric) graph is a network in which nodes and edges are spatial entities living in a metric space (Barthélemy, 2011; Iddianozie & McArdle, 2021). Such networks emerge naturally in domains where spatial embedding is fundamental to structure and function: urban, transportation, and communication networks are shaped by physical distances and road geometries (Buhl et al., 2006; Wang et al., 2020); biological systems like neural and vascular networks are constrained by surrounding tissue geometry (Runions et al., 2005; Bullmore & Sporns, 2009); and river networks

---

[6]To our knowledge, this is also the first *large-scale multigraph* dataset–*large-scale* conforms to OGB criteria (Hu et al.).

evolve through interactions of gravity and topography (Caldarelli, 2007; Rodriguez-Iturbe & Rinaldo, 1997). In all these cases, spatial graph structure encodes essential information that cannot be inferred from connectivity alone or reconstructed from non-spatial data.

Despite growing recognition of spatial information in Spatial and Geo AI (Papadimitriou; Gao, 2021), its importance remains underappreciated in graph learning. Currently, no benchmark exists with sufficiently rich geometric structure to exhibit intricate spatial patterns, let alone one of **spatial multigraphs**. As a result, despite many efforts to develop algorithms for spatial graphs (Guo et al., 2021; Iddianozie & McArdle, 2021; Danel et al., 2019; Ingraham et al., 2019; Yan et al., 2018), the field has lacked a standardized, large-scale testbed.

### A.4    NON-BLOCH BAND THEORY

Non-Hermitian band theory has developed a mature language for classifying complex spectra and their topological phases, including point-/line-gap notions, symmetry-based classifications, and associated boundary phenomena (Lee & Thomale, 2019; Kawabata et al., 2019; Gong et al., 2018; Li & Lee, 2022). A central theme is the breakdown (and restoration) of bulk–boundary correspondence via non-Bloch descriptions, which underpin the non-Hermitian skin effect (Kunst et al., 2018; Jin & Song, 2019; Yang et al., 2020; Lin et al., 2023; Xiong et al., 2024). Recent advances further sharpen the geometric viewpoint on complex band structures through generalized Brillouin-zone constructions and higher-dimensional non-Bloch theory, e.g., the amoeba formulation (Wang et al., 2024; 2025b; Kim et al., 2025; Meng et al., 2026). In parallel, there is growing interests on "spectral geometry" and morphology in the complex plane—from graph-topological characterizations of spectra and band morphology to the structure of self-intersections and spectral transitions (Tai & Lee, 2023; Xiong & Hu, 2023; Yan et al., 2025b; Pi et al., 2025; Zhong et al., 2025). Real-space geometry dependence also appears in experimental settings and in general theories (Fang et al., 2022; Wang et al., 2025a). Beyond classification, there exists dynamical and response-based viewpoints (Li et al., 2021; Lee et al., 2020; Xu et al., 2025). Finally, a growing experimental and digital-quantum literature has begun to directly probe these effects (Wang & Chen, 2025; Zhang et al., 2025; Shen et al., 2025; Koh et al., 2025; Yang et al., 2022).

## B    MATHEMATICAL BACKGROUND - *Hamiltonian Spectral Graph*

### B.1    THE HAMILTONIAN AND ENERGY SPECTRUM IN 1D TIGHT-BINDING SYSTEMS

In physical sciences, it is customary to represent and study a system through its Hamiltonian matrix. The **energy spectrum**, which refers to the set of eigenvalues of this matrix, reveals the energy band structure—a central object of study in condensed matter physics. Let us consider a generic 1D tight-binding Hamiltonian with $s$ internal degrees of freedom (bands or orbitals) per unit cell:

$$\boldsymbol{H} = \sum_{x,j} \boldsymbol{T}_j \hat{c}_x^\dagger \hat{c}_{x+j} \tag{7}$$

where $x$ and $j$ index unit cells and hopping lengths, respectively; $\hat{c}_x$ is the annihilation operator (vector of length $s$) for the $x$-th unit cell. Each term $\boldsymbol{T}_j \in \mathbb{C}^{s \times s}$ represents the transition amplitude matrix for a particle hopping from site $x + j$ to $x$. The $L^2$ norm of the amplitude corresponds to the transition probability. These hopping terms can generically be complex.

The matrix representation of this Hamiltonian in real space, $\boldsymbol{H}_{\text{real}}$, for which the block at $(x, x')$ is $(\boldsymbol{H}_{\text{real}})_{x,x'} = \boldsymbol{T}_{x'-x}$, is a block Toeplitz matrix—a matrix in which each descending block diagonal from left to right is constant:

$$\boldsymbol{H}_{\text{real}} = \begin{pmatrix} \boldsymbol{T}_0 & \boldsymbol{T}_1 & \boldsymbol{T}_2 & & \cdots & & 0 \\ \boldsymbol{T}_{-1} & \boldsymbol{T}_0 & \boldsymbol{T}_1 & \boldsymbol{T}_2 & & & \\ \boldsymbol{T}_{-2} & \boldsymbol{T}_{-1} & \boldsymbol{T}_0 & \boldsymbol{T}_1 & \ddots & & \vdots \\ & \boldsymbol{T}_{-2} & \boldsymbol{T}_{-1} & \boldsymbol{T}_0 & \ddots & & \\ \vdots & & \ddots & \ddots & \ddots & & \boldsymbol{T}_2 \\ & & & & & \boldsymbol{T}_0 & \boldsymbol{T}_1 \\ 0 & & \cdots & & \boldsymbol{T}_{-2} & \boldsymbol{T}_{-1} & \boldsymbol{T}_0 \end{pmatrix} \tag{8}$$

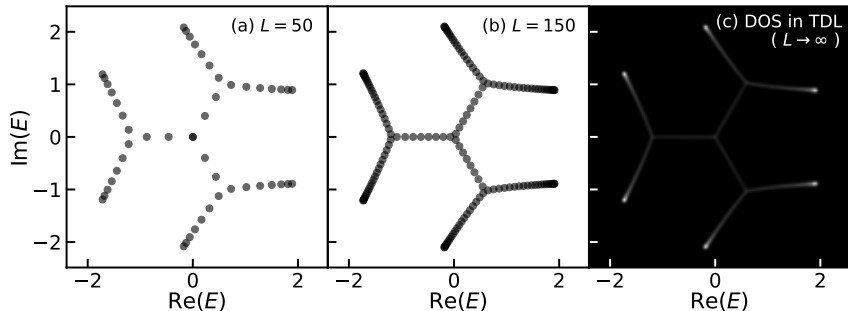

Figure A3: **The emergence of spectral graphs. (a)-(b)** show the OBC energy spectra with increasing system size $L = [50, 150]$, of the non-Hermitian lattice whose characteristic polynomial is $P(z, E) = -z^{-2} - E - z + z^4$. In the thermodynamic limit ($L \to \infty$), the spectra becomes a *band continuum* and the energy loci traces out a planar graph on the complex plain, namely the *spectral graph*. For this particular example, it is a 3-Cayley tree. **(c)** shows the corresponding density of states when $L \to \infty$.

If there are $L$ sites in total, $\boldsymbol{H}_{\text{real}} \in \mathbb{C}^{Ls \times Ls}$. In general, $\boldsymbol{T}_j \neq \boldsymbol{T}^\dagger_{-j}$ (where $\boldsymbol{T}^\dagger_{-j}$ is the conjugate transpose of $\boldsymbol{T}_{-j}$), which breaks the Hermiticity of the Hamiltonian, i.e., $\boldsymbol{H}^\dagger \neq \boldsymbol{H}$. Consequently, the eigenvalues can take on complex values. The energy spectrum is obtained by diagonalizing $\boldsymbol{H}_{\text{real}}$.

## B.2 Hamiltonian Spectral Graph: Emergent Topology in the Thermodynamic Limit

For non-Hermitian systems, the energy eigenvalues form intricate patterns in the complex plane. The **spectral graph** $\mathcal{G}$ emerges from the energy spectra under open boundary conditions (OBC) in the **thermodynamic limit** (i.e., as the system size $L \to \infty$). In this limit, the discrete energies become continuous, and their loci trace out a planar graph on the complex plane (Tai & Lee, 2023; Xiong & Hu, 2023). Figure A3 illustrates this emergence: the OBC energy spectra for finite system sizes $L = 50$ and $L = 150$ of a non-Hermitian lattice (whose characteristic polynomial, defined later, is $P(z, E) = -z^{-2} - E - z + z^4$) clearly approach a 3-Cayley tree as $L$ increases. Figure A3c shows the corresponding **density of states** (DOS) in the $L \to \infty$ limit.

The spectral graphs of different lattices exhibit a kaleidoscope of geometries, including arcs, loops, and more exotic shapes resembling stars, kites, braids, and even rockets (Tai & Lee, 2023; Lin et al., 2023), as showcased in Figure A4. These structures represent an uncharted band topology, embedding hidden symmetries and graph topological transitions that lie beyond standard homotopy-based frameworks (Hasan & Kane, 2010). In effect, *a new class of topological invariants appears*—those tied to the global geometry of the eigenvalue loci.

However, accurately diagonalizing a large non-Hermitian matrix $\boldsymbol{H}_{\text{real}}$ to obtain the OBC spectrum is notoriously difficult (Yang et al., 2020), let alone for an infinite-sized matrix (i.e. an operator). This necessitates a more sophisticated theoretical approach.

## B.3 Theoretical Framework: Non-Bloch Band Theory

The standard approach to analyze such systems, guided by non-Bloch band theory, begins with a Fourier transformation and the analysis of the resulting characteristic polynomial.

### B.3.1 The Bloch Hamiltonian and Characteristic Polynomial $P(z, E)$

Fourier transforming the real-space Hamiltonian (second quantized form—equation 7 or its matrix form—equation 8) yields the Bloch Hamiltonian:

$$\boldsymbol{H}(z) = \sum_j \boldsymbol{T}_j z^j, \quad z := e^{ik} \tag{9}$$

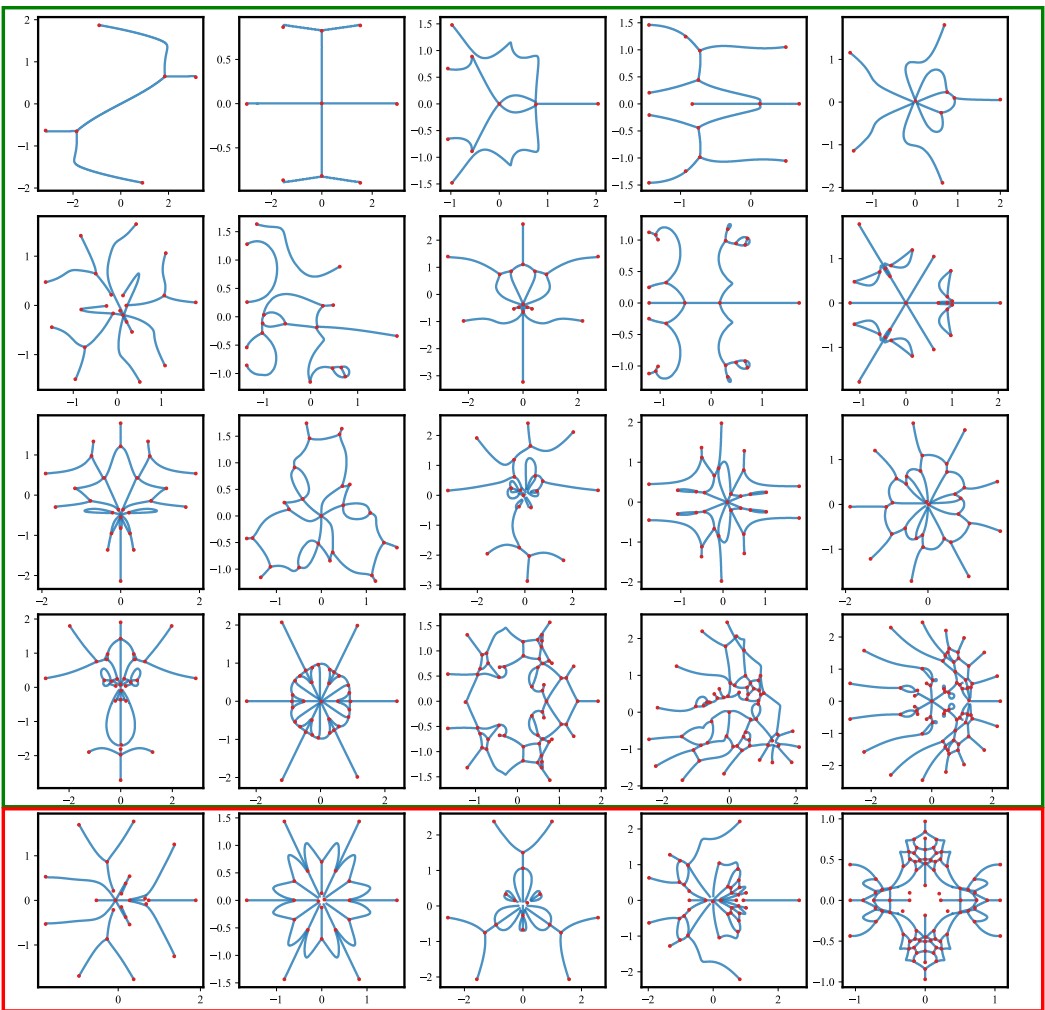

Figure A4: **A Gallery of Spectral Graphs.** The top four rows highlight the intricate structures characteristic of spectral graphs. The bottom row illustrates the distinct phenomenon we refer to as *component fragmentation* (Section 5)—some nodes in theory should be connected, however its surrounding low density of states limits accurate edge detection, causing certain nodes to be *fragmented* into disjoint nodes, often leading to fragmentation of an otherwise connected component. The phenomena often occurs for high-band and long-range hopping crystals.

with $\boldsymbol{T}_j \in \mathbb{C}^{s \times s}$. Let the hopping range of $\boldsymbol{H}$ be $[-p_{\boldsymbol{H}}, q_{\boldsymbol{H}}]$, such that $\boldsymbol{T}_j = 0$ for $j \notin [-p_{\boldsymbol{H}}, q_{\boldsymbol{H}}]$. $\boldsymbol{H}(z)$ is a matrix-valued Laurent polynomial of the phase factor $z = e^{ik}$, where $k$ is the crystal momentum.

The energy dispersion relation is found by solving the secular equation. The **characteristic polynomial** of the Hamiltonian is defined as:

$$P(z, E) := \det\big[\boldsymbol{H}(z) - E\,\boldsymbol{I}\big] = \sum_{n=-p_P}^{q_P} a_n(E)\, z^n. \tag{10}$$

This is a finite *Laurent* polynomial in $z$ whose coefficients $a_n(E)$ are themselves scalar polynomials in $E$ of degree $\leq s$. This equation is also known as the **energy-momentum dispersion**.

It is sometimes convenient to clear the negative powers in equation 10 by defining the ordinary polynomial in $z$,

$$\widetilde{P}(z, E) := z^{p_P} P(z, E) \in \mathbb{C}[z, E], \tag{11}$$

whose degree in $z$ equals $d_z := p_P + q_P$.

**Degree bounds in $z$.** The highest and lowest degree bounds of the associated $P(z, E)$ satisfy:

$$p_P \leq s \times p_{\boldsymbol{H}}, \qquad q_P \leq s \times q_{\boldsymbol{H}}, \tag{12}$$

with equality holding *generically* (i.e., unless the leading/minor determinants vanish because of special symmetries or parameter fine-tuning). In particular, $d_z = p_P + q_P \leq s(p_{\boldsymbol{H}} + q_{\boldsymbol{H}})$.

**Roots in $z$ at fixed $E$.** For any fixed $E \in \mathbb{C}$, the equation $P(z, E) = 0$ has exactly $d_z$ solutions $\{z_\alpha(E)\}_{\alpha=1}^{d_z}$ (equivalently, $\widetilde{P}(z, E)$ has $d_z$ roots in $z$). We will order them by modulus,

$$|z_1(E)| \leq |z_2(E)| \leq \cdots \leq |z_{d_z}(E)|.$$

**Examples.**

1. *One-band, nearest-neighbor ($s = 1$, $p_{\boldsymbol{H}} = q_{\boldsymbol{H}} = 1$).* Let

$$\boldsymbol{H}(z) = t_{-1} z^{-1} + t_0 + t_{+1} z, \qquad P(z, E) = \boldsymbol{H}(z) - E.$$

Here $p_P = q_P = 1$ and $d_z = 2$. The periodic boundary condition (PBC) dispersion is $E(k) = t_0 + t_{+1} e^{ik} + t_{-1} e^{-ik}$.

2. *Two-band SSH-type model ($s = 2$, $p_{\boldsymbol{H}} = q_{\boldsymbol{H}} = 1$).*

$$\boldsymbol{H}(z) = \begin{pmatrix} 0 & t_1 + t_2 z \\ t_1 + t_2 z^{-1} & 0 \end{pmatrix}, \quad P(z, E) = \det[\boldsymbol{H}(z) - E\boldsymbol{I}] = E^2 - (t_1 + t_2 z)(t_1 + t_2 z^{-1}).$$

Here $p_P = q_P = 1$ (so $d_z = 2$), which is *strictly smaller* than the generic bound $s(p_{\boldsymbol{H}} + q_{\boldsymbol{H}}) = 4$.

**Interpretation of $z$ exponents.** The exponents of $z$ in $\boldsymbol{H}(z)$ have a direct hopping interpretation (a nonzero $\boldsymbol{T}_j$ encodes $j$-th neighbor hopping). After taking the determinant to form $P(z, E)$, individual $z^n$ terms no longer correspond to a single hopping distance; rather, they arise from products of matrix entries and thus encode *composite* hopping paths. This explains why the bounds in equation 12 can be strict: symmetries (e.g., the chiral symmetry in the SSH example above) can cause cancellations of the leading powers of $z$ in $P(z, E)$.

### B.3.2 Limitations of Standard Bloch Theory (PBC)

Under periodic boundary conditions (PBC), Bloch theory applies, utilizing the constraint $|z| = 1$ (real momentum $k$). If $\boldsymbol{H}(z)$ is Hermitian on the unit circle, the PBC spectrum consists of real $E$ values forming continuous bands (the usual Bloch dispersion). For non-Hermitian systems, the PBC spectrum typically forms loops or closed curves in the complex energy plane.

However, for non-Hermitian systems under OBC, eigenstates often exhibit the **non-Hermitian skin effect** (NHSE) (Lin et al., 2023), where a macroscopic number of eigenstates localize at the boundaries. Consequently, the PBC and OBC spectra can be qualitatively different. As one evolves the system from PBC to OBC (e.g., by turning off boundary hoppings), the PBC spectrum (loops) often *collapses* inwards to form the skeletal structure of the OBC spectral graph, as depicted in Figure A5.

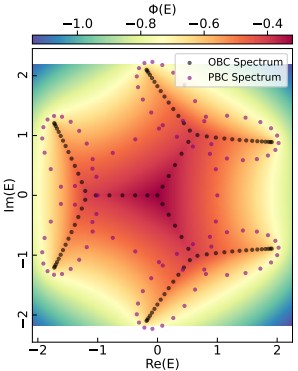

Figure A5: **Spectral Collapse & Spectral Potential.** PBC spectrum usually appears as circles and loops; changing to OBC, the spectrum *collapses* into a graph skeleton. The spectral graph resides on the *ridges* of the potential landscape, $\Phi(E)$.

### B.3.3 NON-BLOCH BAND THEORY AND THE GENERALIZED BRILLOUIN ZONE (GBZ)

Since standard Bloch theory is inapplicable under OBC due to the NHSE, **non-Bloch band theory** is employed. This theory introduces the concept of the **generalized Brillouin zone** (GBZ).

Under OBC, $z$ is allowed to leave the unit circle, meaning $k$ can be complex. The imaginary part of $k$, $\kappa := \text{Im}(k) = -\log|z|$, is the inverse decay length (or inverse skin depth), quantifying the localization of skin modes.

Non-Bloch band theory establishes that the OBC spectrum in the thermodynamic limit is determined by those $E \in \mathbb{C}$ for which the $z$ roots of $P(z, E) = 0$ satisfy a specific *equal-modulus* condition. Given the characteristic polynomial $P(z, E)$ with degrees $p_P$ and $q_P$ as defined in equation 10, the condition for $E$ to be in the OBC spectrum is:

$$|z_{p_P}(E)| = |z_{p_P+1}(E)|. \tag{13}$$

The corresponding loci of $z$'s is the *generalized Brillouin zone* (GBZ). The spectral graph $\mathcal{G}$ is precisely the set of $E$ values that satisfy this condition; equivalently, it is the image of the GBZ under the map $(z \mapsto E : P(z, E) = 0)$.

### B.3.4 CHARACTERISTIC POLYNOMIAL CLASS $\mathcal{C}_P$

The algebraic structure of $P(z, E)$ plays a crucial role in determining the topology of the spectral graph. To understand this relationship, we must consider $P(z, E)$ as a bivariate polynomial, examining the interplay between the powers of $z$ (representing spatial structure) and the powers of $E$ (representing energy bands). We expand $P(z, E)$ fully as:

$$P(z, E) = \sum_{n=-p_P}^{q_P} a_n(E) z^n = \sum_{n=-p_P}^{q_P} \sum_{m=0}^{s} c_{n,m} E^m z^n. \tag{14}$$

We define the **characteristic polynomial *class* $\mathcal{C}_P$** based on the **monomial support** of $P(z, E)$. The support identifies which specific monomials $E^m z^n$ are present in the polynomial structure, regardless of the exact values of their coefficients $c_{n,m}$.

Formally, we define the support $\boldsymbol{S}_P$ as the set of index pairs $(n, m)$ for which the coefficient $c_{n,m}$ is structurally non-zero (i.e., it is allowed to vary, rather than being identically zero):

$$\boldsymbol{S}_P = \{(n, m) \mid c_{n,m} \not\equiv 0\}. \tag{15}$$

Crucially, the spectral graph is invariant under parity transformation[7], which corresponds to the transformation $z \to z^{-1}$ in the polynomial. Let $P'(z, E) = P(z^{-1}, E)$. The support of this parity-

---

[7]I.e., spatial inversion about the origin ($x \to -x$), or flipping the 1D lattice from left to right. In terms of $\boldsymbol{H}_{\text{real}}$, this corresponds to $\boldsymbol{T}_j \to \boldsymbol{T}_{-j}$, which is equivalent to transposing the matrix ($\boldsymbol{H}_{\text{real}} \to \boldsymbol{H}_{\text{real}}^T$). The transpose does not change the eigenvalues.

transformed polynomial is:

$$\boldsymbol{S}_{P'} = \{(-n, m) \mid (n, m) \in \boldsymbol{S}_P\}. \tag{16}$$

The characteristic polynomial class $\mathcal{C}_P$ is defined as the equivalence class represented by this pair of supports:

$$\mathcal{C}_P = \{\boldsymbol{S}_P, \boldsymbol{S}_{P'}\}. \tag{17}$$

This classification ensures that polynomials related by spatial inversion, which necessarily yield the same spectral graph, belong to the same class. If the polynomial structure is palindromic in $z$ (i.e., $\boldsymbol{S}_P = \boldsymbol{S}_{P'}$), the class is simply identified by the single support set $\boldsymbol{S}_P$.

We find that the characteristic polynomial class $\mathcal{C}_P$ is a key criterion for classifying spectral graph topologies and is thus the target for inverse classification tasks. Varying the specific values of the coefficients $c_{n,m}$ within a fixed class $\mathcal{C}_P$ may deform the geometry of the spectral graph but typically preserves its fundamental topology.

**Examples:**

1. *Single-band example ($s = 1$) from Figure A3.* $P(z, E) = -z^{-2} - z + z^4 - E$. The monomials present are $E^0 z^{-2}$, $E^0 z^1$, $E^0 z^4$, and $E^1 z^0$. The support is $\boldsymbol{S}_P = \{(-2, 0), (1, 0), (4, 0), (0, 1)\}$. The parity-transformed polynomial is $P(z^{-1}, E) = -z^2 - z^{-1} + z^{-4} - E$. The transformed support is $\boldsymbol{S}_{P'} = \{(2, 0), (-1, 0), (-4, 0), (0, 1)\}$. The class is $\mathcal{C}_P = \{\boldsymbol{S}_P, \boldsymbol{S}_{P'}\}$.

2. *Two-band example ($s = 2$).* Consider a class defined by the structure (similar to the dataset construction example): $P(z, E) = -E^2 + z^{-3} + z^3 + c_1 z^{-1} + c_2 E z + E z^2$. The support is $\boldsymbol{S}_P = \{(0, 2), (-3, 0), (3, 0), (-1, 0), (1, 1), (2, 1)\}$. This support explicitly captures the interplay between hopping range and energy bands, defining the class.

### B.3.5 RECAP OF KEY CONCEPTS (HIERARCHY OF ABSTRACTIONS)

**Objects and their roles.**

- **Real-space Hamiltonian $\boldsymbol{H}_{\text{real}}$:** an infinite banded (block) Toeplitz operator acting on $\ell^2(\mathbb{Z}) \otimes \mathbb{C}^s$, formed from hopping blocks $\{\boldsymbol{T}_j\}$.

- **Bloch Hamiltonian $\boldsymbol{H}(z)$:** the $s \times s$ matrix Laurent polynomial in equation 9, the Fourier transform of $\boldsymbol{H}_{\text{real}}$.

- **Characteristic polynomial $P(z, E)$:** the scalar Laurent polynomial equation 10; a bivariate polynomial in $z$ and $E$ (after clearing denominators in $z$).

- **ChP class $\mathcal{C}_P$:** an equivalence class of characteristic polynomials defined by fixing the *monomial support* in $(z, E)$ and accounting for parity symmetry (see equation 17).

- **Spectral graph $\mathcal{G}$:** the subset of $\mathbb{C}$ traced by eigen-energies $E$ in the thermodynamic limit under OBC,

$$\mathcal{G} = \{E \in \mathbb{C} : \forall z \in \text{GBZ}, \ P(z, E) = 0\}.$$

A specific Hamiltonian (or specific $P$) maps to a specific spectral graph; a ChP class maps to a *family* of spectral graphs parameterized by its coefficients.

**How the abstractions relate.**

$$\boldsymbol{H}_{\text{real}} \xrightarrow{\text{Fourier}} \boldsymbol{H}(z) \xrightarrow{\det[\cdot - E\boldsymbol{I}]} P(z, E) \xrightarrow{\text{Support \& Symmetry}} \mathcal{C}_P$$

$$P(z, E) \xrightarrow{\text{solve in } z \text{ (GBZ)}} \mathcal{G} \subset \mathbb{C}.$$

Each arrow forgets inessential microscopic details while preserving spectral information relevant at the next level. This hierarchy clarifies terminology used in this work.

| Object | Symbol | Mathematical type | Typical size |
|---|---|---|---|
| Real-space Hamiltonian | $\boldsymbol{H}_{\text{real}}$ | banded Toeplitz operator | $L \times L$ (finite) or infinite |
| Bloch Hamiltonian | $\boldsymbol{H}(z)$ | Laurent-poly. matrix | $s \times s$ ($s$ bands/orbitals) |
| Characteristic polynomial | $P(z, E)$ | Laurent poly. in $z$ & Poly. in $E$ | $(p_P, q_P)$ in $z$; degree $\leq s$ in $E$ |
| ChP class | $\mathcal{C}_P$ | set of $P$'s with fixed support up to parity symmetry | — |
| Spectral graph | $\mathcal{G}$ | spatial planar multigraph on $\mathbb{C}$ (energy plane) | — |

### B.4 THE SHORTCUT TO SPECTRAL GRAPH VIA ELECTROSTATIC ANALOGY.

#### B.4.1 DENSITY OF STATES $\rho(E)$ AND THE SPECTRAL POTENTIAL $\Phi(E)$

The **density of states** (DOS) describes the continuous spectrum. It is defined as the density of eigenstates on the complex energy plane. An example of DOS is shown in Figure A3c.

Recent developments in non-Bloch theory map the problem of finding the spectral graph and DOS to a classic 2D electrostatic problem (Xiong & Hu, 2023; Yang et al., 2022; Wang et al., 2024). If we treat the eigenvalues $\epsilon_n$ (for a system of size $N$) as electric charges of strength $1/N$, we can define the Coulomb potential $\Phi(E)$, also called the **spectral potential**, at a point $E \notin \mathcal{G}$:

$$\Phi(E) = -\lim_{N \to \infty} \frac{1}{N} \sum_{\epsilon_n} \log |E - \epsilon_n|$$

$$= -\int \rho(E') \log |E - E'| \, d^2 E' \tag{18}$$

The DOS is related to the potential by the Poisson equation:

$$\rho(E) = -\frac{1}{2\pi} \nabla^2 \Phi(E) \tag{19}$$

where $\nabla^2 = \partial^2_{\text{Re}E} + \partial^2_{\text{Im}E}$ is the Laplacian operator on the complex energy plane. The Laplacian extracts curvature. Geometrically, this implies that the loci of the spectral graph $\mathcal{G}$, where the DOS is concentrated, reside on the *ridges* of the Coulomb potential landscape $\Phi(E)$, as suggested in figure A5 and figure 2.

#### B.4.2 EFFICIENT CALCULATION OF $\Phi(E)$

Leveraging Szegö's strong limit theorem, the spectral potential $\Phi(E)$ in equation 18 can be reduced to a computationally efficient form based directly on the characteristic polynomial $P(z, E)$:

$$\Phi(E) = -\log |a_{q_P}(E)| + \sum_{i=p_P+1}^{p_P+q_P} \kappa_i(E) \tag{20}$$

Here, $p_P$ and $q_P$ are the lowest and highest degrees of $z$ in $P(z, E)$, respectively (see equation 10). $a_{q_P}(E)$ is the coefficient of $z^{q_P}$. $\kappa_i(E) = -\log |z_i(E)|$ are the inverse decay lengths associated with the $q_P$ largest roots of $P(z, E) = 0$ (these are $z_{p_P+1}, \ldots, z_{p_P+q_P}$ in the sorted list).

Although equation 18 is strictly defined for $E \notin \mathcal{G}$, the expression in equation 20 can be analytically continued to the entire complex plane (Xiong & Hu, 2023). This allows the construction of the potential landscape $\Phi(E)$ merely by knowing the characteristic polynomial $P(z, E)$, thereby obviating the need for direct diagonalization of large real-space Hamiltonians and avoiding the numerical errors associated with such diagonalizations.

## C POLY2GRAPH PIPELINE DETAILS

Armed with the above theoretical guidance, we implement the transformations numerically, and then integrate a few computer vision techniques (Lee et al., 1994; Wang et al., 2018; Nunez-Iglesias et al., 2018) to construct the spectral graph given its characteristic polynomial (or Bloch Hamiltonian). This appendix complements section 2. The core procedure of `Poly2Graph` algorithm ("Characteristic **Poly**nomial **to** Spectral **Graph**") is summarized in algorithm 1.

---

**Algorithm 1:** `Poly2Graph`: Characteristic **Poly**nomial **to** Spectral **Graph**

---

**Input:** *(1)* $\boldsymbol{H}(z)$ *or* $P(z, E) := \det[\boldsymbol{H}(z) - E\,\boldsymbol{I}]$
`# ↑ Hamiltonian or its characteristic polynomial`
**Input:** *(2) Energy Domain:* $\Omega \subset \mathbb{C}$ *such that* $\Omega \supsetneq \mathcal{G}$ *(spectral graph)*
**Output:** *Spectral Graph:* $\mathcal{G} \in$ `networkx.MultiGraph`

**begin**
  `# Build the characteristic polynomial if only` $\boldsymbol{H}(z)$ `was given`
  **if** *input* $\boldsymbol{H}(z)$ **then**
    $\;\;\lfloor\; P(z, E) = \det[\boldsymbol{H}(z) - E\,\boldsymbol{I}] = \sum_{n=-p}^{q} a_n(E)\, z^n$

  `# Stage 1:  Coarse computation over initial energy grid` $\Omega$
  **(Parallel) for** $E \in \Omega$ **do**
    `# Solve roots`
    $\{z_i(E)\} = \text{Sort}[\text{Roots}(P(z, E))]$ s.t. $|z_1| \leq \cdots \leq |z_{p+q}|$
    `# Compute spectral potential`
    $\Phi(E) = -\log|a_q(E)| - \sum_{i=p+1}^{p+q} \log|z_i(E)|$
    `# Compute Density of States (DOS)`
    $\lfloor\; \rho(E) = -\frac{1}{2\pi} \nabla^2 \Phi(E)$

  `# Identify regions of interest`
  $\text{coarse\_mask} = \text{dilate}(\text{binarize}(\{\rho(E)\}_{E \in \Omega}))$
  `# Define refined energy grid`
  $\Omega' = \text{get\_masked\_subgrid}(\text{coarse\_mask})$

  `# Stage 2:  Refined computation within masked regions` $\Omega'$
  **(Parallel) for** $E \in \Omega'$ **do**
    `# Re-solve roots at higher resolution`
    $\{z_i(E)\} = \text{Sort}[\text{Roots}(P(z, E))]$
    `# Recompute spectral potential`
    $\Phi'(E) = -\log|a_q(E)| - \sum_{i=p+1}^{p+q} \log|z_i(E)|$
    `# Recompute DOS`
    $\lfloor\; \rho'(E) = -\frac{1}{2\pi} \nabla^2 \Phi'(E)$

  `# Combine coarse and refined DOS for full high-resolution`
  ` image`
  $\rho_{\text{final}}(E) = \text{combine}(\{\rho(E)\}_{E \in \Omega \setminus \Omega'}, \{\rho'(E)\}_{E \in \Omega'})$
  `# Binarize high-resolution DOS`
  $\text{final\_binarized\_image} = \text{binarize}(\{\rho_{\text{final}}(E)\}_{E \in \Omega})$
  `# Extract one-pixel-wide skeleton`
  $\text{graph\_skeleton} = \text{skeletonize}(\text{final\_binarized\_image})$
  `# Convert skeleton to graph object`
  $\mathcal{G} = \text{skeleton2graph}(\text{graph\_skeleton})$
  `# Post-processing`
  $\mathcal{G} = \text{merge\_nearby\_nodes}(\mathcal{G}, \text{tolerance})$
  $\lfloor\; \mathcal{G} = \text{remove\_isolated\_nodes}(\mathcal{G})$

---

## C.1 INITIALIZATION AND INPUT

Poly2Graph accepts diverse input formats for the 1-D tight-binding crystal. It can initialize from a Bloch Hamiltonian $\boldsymbol{H}(z)$ or directly from its characteristic polynomial $P(z, E)$. Supported formats for $P(z, E)$ include `sympy.Matrix` (for $\boldsymbol{H}(z)$, $\boldsymbol{H}(k)$), `sympy.Poly`, or a raw string expression of the polynomial. During initialization, Poly2Graph automatically computes a full set of different representations and properties. See the tutorial section appendix G or visit our repository https://github.com/sarinstein-yan/Poly2Graph.

The energy domain $\Omega \subset \mathbb{C}$, which must fully contain the spectral graph $\mathcal{G}$, is also required. While users can specify a custom $\Omega$ and its discretization, by default Poly2Graph can automatically esti-

mate a suitable region by diagonalizing a small real-space Hamiltonian (typically $L = 40$ unit cells) and applying a small padding.

**Sidenotes: notions of "size" appeared so far.**

1. *System size* — $L$ (real-space lattice length):
   The number of unit cells. The operator $\boldsymbol{H}_{\text{real}}$ is $L \times L$ for finite $L$ and becomes an infinite operator in the thermodynamic limit $L \to \infty$.

2. *Internal size* — $s$ (band/orbital count):
   The matrix dimension of $\boldsymbol{H}(z)$. This equals the maximal degree in $E$ of $P(z, E)$.

3. *Laurent degree in $z$* of $P(z, E)$ — the pair $(p_P, q_P)$ (or total $d_z = p_P + q_P$):
   Governing the number of $z$ roots at fixed $E$. It is controlled by the degree range of $\boldsymbol{H}(z)$'s elements via equation 12.

4. *Rasterization resolution of spectral images* — $N$ (pixels):
   The size of the spectral images that serve as intermediate construct between input characteristic and output spectral graph. Not a property of the crystal.

## C.2 Accelerated Root Finding

As detailed in section 2, solving for the roots $\{z_i(E)\}$ of $P(z, E) = 0$ for each energy $E$ in the discretized domain $\Omega$ is the primary computational bottleneck. To achieve the reported performance gains (five orders of magnitude speedup and higher memory efficiency over previous methods (Tai & Lee, 2023) on default settings), we employ a specialized root-finding strategy.

For a characteristic polynomial $P(z, E) = \sum_{j=-p}^{q} a_j(E) z^j$, its roots are equivalent to the eigenvalues of its Frobenius companion matrix $\boldsymbol{F}(E)$. For a polynomial of degree $d = p + q$, the companion matrix is a $d \times d$ matrix constructed from the coefficients:

$$\boldsymbol{F}(E) = \begin{pmatrix} 0 & 0 & \cdots & 0 & -a_{-p}(E)/a_q(E) \\ 1 & 0 & \cdots & 0 & -a_{-p+1}(E)/a_q(E) \\ 0 & 1 & \cdots & 0 & -a_{-p+2}(E)/a_q(E) \\ \vdots & \vdots & \ddots & \vdots & \vdots \\ 0 & 0 & \cdots & 1 & -a_{q-1}(E)/a_q(E) \end{pmatrix}. \tag{21}$$

This formulation holds for complex coefficients $a_j(E) \in \mathbb{C}$.

Poly2Graph constructs a batch of these companion matrices for each $E \in \Omega$, where $\Omega$ is the discretized grid of energy values. This batch is then processed by a parallelized eigensolver. The implementation automatically detects the availability of `TensorFlow` or `PyTorch` backends, leveraging them for hardware acceleration, including optional GPU support via CUDA. To optimize memory and computation, calculations are performed using single precision (float32), which has been found sufficient for high-fidelity spectral graph extraction.

## C.3 Adaptive Resolution and Image Processing

The adaptive resolution strategy, outlined in section 2, is crucial for computational tractability.

1. **Coarse Identification**: The spectral potential $\Phi(E)$ (equation 3) and DOS $\rho(E)$ (equation 4) are computed on an initial, moderately resolved grid (e.g., $256 \times 256$). The DOS image is binarized and morphologically dilated to create a conservative mask $\Omega'$ covering potential graph regions.

2. **Refined Calculation**: Within this mask $\Omega'$, each pixel is subdivided (e.g., into a $4 \times 4$ subgrid), and $\Phi(E)$ and $\rho(E)$ are recomputed at this higher resolution.

This two-stage process achieves high effective resolution (e.g., $1024 \times 1024$) while minimizing redundant calculations in empty regions of the complex energy plane.

The resulting high-resolution DOS image is again binarized. We currently employ a *global mean threshold* ($\rho(E) > \langle \rho(E') \rangle_{E' \in \Omega}$) for binarization, as it has empirically outperformed a pool of other common global and adaptive thresholding heuristics, including Otsu, Li, Yen, Triangle, Isodata,

local adaptive, and hysteresis variants for our datasets. Subsequently, iterative morphological thinning operations (Lee et al., 1994) are applied to reduce the binarized features to a one-pixel-wide skeleton, revealing the graph topology.

## C.4 GRAPH EXTRACTION AND POST-PROCESSING

The `skeleton2graph` submodule converts the binary skeleton into a graph representation. It identifies pixels as junction nodes (three or more neighbors), leaf nodes (one neighbor), or edge points (two neighbors). The output is a `NetworkX MultiGraph` object, where each edge in particular stores its geometric path as an ordered sequence of $(\mathrm{Re}(E), \mathrm{Im}(E))$ coordinates.

To handle numerical artifacts, two post-processing steps are implemented as shown in algorithm 1:

1. `merge_nearby_nodes`: Nodes within a predefined Euclidean distance tolerance are merged. This helps consolidate fragmented junctions.
2. `remove_isolated_nodes`: Nodes with no connecting edges after the merging step are removed.

## C.5 BENCHMARK POLY2GRAPH SPEED-UP

The primary innovation is the *end-to-end automation* of the spectral graph extraction process, which was previously reliant on manual inspection. This automation makes systematic research on spectral graphs feasible.

The speed and memory efficiency are secondary, but *critical*, feature that make *large-scale* research on spectral graphs feasible for the first time.

Since Poly2Graph is the first tool of its kind that fully automates the entire pipeline, a true predecessor for an end-to-end comparison does not exist. Here, in table A3, we only benchmark the *computation bottleneck (spectral potential calculation)* against the best available code from Ref. (Tai & Lee, 2023), which does not automate the graph extraction.

Table A3: Speed comparison between Poly2Graph and the best available code from Ref. (Tai & Lee, 2023). The time used per sample (i.e. for an input $H(z)/P(z, E)$) for each method, and the speed-up that the Poly2Graph obtained are reported. The comparison is conducted on the computation bottleneck (spectral potential calculation), as Ref. (Tai & Lee, 2023) does not automate the graph extraction. Poly2Graph's time complexity is polynomial in degree range ($d_z = p + q$).

| degree range | Poly2Graph | Ref. (Tai & Lee, 2023) | Speed-up |
|---|---|---|---|
| p+q=2 | 13.1 ± 0.3 ms | 3025 s | 2.3e5 × |
| p+q=3 | 20.8 ± 0.1 ms | 3423 s | 1.6e5 × |
| p+q=4 | 28.6 ± 0.3 ms | 3921 s | 1.4e5 × |
| p+q=5 | 38.8 ± 0.2 ms | 5177 s | 1.3e5 × |
| p+q=6 | 50.8 ± 0.3 ms | 6199 s | 1.2e5 × |

## C.6 CAVEATS: COMPONENT FRAGMENTATION

A notable challenge, particularly for systems with large degree ranges or many bands, is a phenomenon we termed *component fragmentation*. As illustrated in the bottom row of figure A4, this refers to the spurious disconnection of spectral branches that should ideally form a single connected component. Fragmentation typically arises at junction nodes where the surrounding DOS is exceptionally low. In such cases, the spectral potential landscape ($\Phi(E)$) around these junctions is virtually flat, making the corresponding ridges (which correspond to edges) fall below the detection threshold of the binarization and thinning processes, due to finite floating-point precision.

While the current global mean thresholding for binarization is a robust general choice, it may struggle with complicated spectra. Ultra-low-DOS edges can be missed, leading to missing pixels in the skeleton and thus fragmentation. While more sophisticated ridge-following or adaptive local thresholding algorithms might offer improvements, they often come at a significant cost to Poly2Graph's

speed and memory efficiency. Addressing this intrinsic limitation robustly remains an area for future development.

# D    DATASET DETAILS

## D.1    COMPARISON WITH 45 OTHER DATASETS

We present a comprehensive comparison of our dataset in terms of both structural properties and statistical metrics. Table A4 compiles all prominent graph datasets to the best of our knowledge. Each column is described in the caption. As illustrated, while some spatial graph datasets do exist, they generally lack rich connection geometry (RCG. Nontrivial edge patterns beyond a simple straight-line link) or support for multiple parallel edges between nodes. The dataset most similar to HSG-12M in these respects is OpenStreetMap; however, it is not designed with any ML downstream tasks, contains far fewer graphs, and is medium-scale judged by OGB criteria [39]. Moreover, although it supports non-linear edge shapes—streets connecting a pair of destinations are usually not straight-lines—the complexity of its connectivity is limited. In contrast, the edge geometries in our setting exhibit much richer geometric variation. Consequently, prior to this work, the absence of a *large-scale* multigraph learning challenge remain unaddressed.

Moreover, as shown in the table, our large-scale T-HSG-5M dataset is the only temporal dataset that includes class labels. This is particularly valuable in our setting, as different classes may exhibit distinct temporal evolution patterns. We leave the investigation of these dynamics to future research.

### D.1.1    COMPARISON WITH OTHER DATASETS IN PHYSICAL SCIENCES

Additionally, as illustrated in figure 1, HSM-12M stands out as the largest classification dataset in terms of both graph count and class diversity. Although our dataset is designed for classification, it is still informative to compare it with others based on the total number of graphs and nodes. By these metrics, certain large-scale computer science datasets—such as TpuGraphs, Tenset, and Mal-Net—contain larger overall volumes. However, a fairer comparison emerges when we evaluate our dataset alongside those from the natural sciences, as they are constructed using similar methodologies.

To facilitate this comparison, we selected the largest datasets from the table and visualized them in figure A6. The results show that even among natural science datasets not constrained to classification tasks, ours stands out as not only competitive but also the largest in scale. These findings highlight the scope and impact of this work.

## D.2    CLARIFICATION ON `HSG-topology` DATASET VARIANT

The `HSG-topology` variant contains only pairwise non-*isomorphic* graphs. Within each ChP class we retain exactly one representative for every unique *connectivity pattern*—equivalently, one per isomorphism class that have the same adjacency matrix (i.e., up to a relabeling of nodes)—and discard the rest isomorphic samples. This construction makes the dataset purely combinatorial.

This filtering induces a pronounced class imbalance: different Hamiltonian families generate markedly different numbers of distinct topologies (i.e. isomorphisms).

This design is physically motivated: we are interested in how spectral-graph *topology* varies with the underlying Hamiltonian parameters.

Table A4: Overview of graph-level benchmark datasets. Each row corresponds to one dataset: **#Graphs** gives the total number of graphs; **#Classes** is the number of target labels; **#Nodes** and **#Edges** report the average number of nodes and edges per graph; **Scale** column follows the OGB Large-Scale Challenge definitions [39]: Small (S) datasets fall below the large-scale thresholds; Medium (M) datasets contain > 1 million nodes or > 10 million edges; Large (L) datasets exceed 100 million nodes or 1 billion edges.; **Attributed** indicates whether both node and edge features are present; **Spatial** denotes whether the graphs carry geometric or coordinate information (e.g. 2D, 3D, geographic coordinate system-GCS); **Temporal** flags static (S) or time-series graph data (T); **Multi** marks support for multiple edges between node pairs; and **RCG** (Rich Connection Geometry) indicates datasets whose edge geometry exhibits nontrivial patterns that go beyond simple straight-line connections. "?" entries indicate information not stated in the original paper. In addition to values extracted from the literature, some benchmark statistics were sourced from Refs. [39; 119; 21; 78].

| Dataset | #Graphs | #Classes | #Nodes | #Edges | Scale | Attributed | Spatial | Temporal | Multi | RCG |
|---|---|---|---|---|---|---|---|---|---|---|
| **Biology** | | | | | | | | | | |
| ENZYMES [5; 75] | 600 | 6 | 32.6 | 62.1 | S | – | – | S | – | – |
| PROTEINS [5; 75] | 1.1K | 2 | 39.1 | 72.8 | S | ✓ | – | S | – | – |
| D&D [75; 20] | 1.2K | 2 | 284.3 | 715.7 | S | – | – | S | – | – |
| ProFold [32] | 76K | – | 8.0 | ? | S | ✓ | 3D | T | – | – |
| NeuroGraph [87] | 23K | 7 | 359.6 | 11K | M | – | – | S | – | – |
| Skeleton (NTU-RGB+D) [88] | 56K | 60 | 25.0 | 24 | M | – | 3D | T | – | – |
| ppa [126; 38] | 158K | 37 | 243.4 | 266.1 | M | ✓ | – | S | – | – |
| Skeleton (Kinetics) [51] | 260K | 400 | 18.0 | 17 | M | – | 2D | T | – | – |
| **Chemistry** | | | | | | | | | | |
| MUTAG [56; 75] | 188 | 2 | 17.9 | 19.8 | S | ✓ | – | S | – | – |
| SIDER [105; 1] | 1.4K | 2 | 33.6 | 35.4 | S | ✓ | – | S | – | – |
| BACE [105; 90] | 1.5K | 2 | 34.1 | 36.9 | S | ✓ | – | S | – | – |
| ClinTox [105; 29] | 1.5K | 2 | 26.2 | 27.9 | S | ✓ | – | S | – | – |
| BBBP [105; 71] | 2.0K | 2 | 24.1 | 25.9 | S | ✓ | – | S | – | – |
| Tox21 [105; 94] | 7.8K | 2 | 18.6 | 19.3 | M | ✓ | – | S | – | – |
| ToxCast [105; 83] | 8.6K | 2 | 18.8 | 19.3 | M | ✓ | – | S | – | – |
| Peptides-func [24] | 15.5K | – | 150.9 | 307.3 | M | ✓ | 3D | S | – | – |
| Peptides-struct [24] | 15.5K | – | 150.9 | 307.3 | M | ✓ | 3D | S | – | – |
| MolHIV [105; 38] | 41.1K | 2 | 25.5 | 27.5 | M | ✓ | – | S | – | – |
| MUV [105; 85] | 93.1K | 2 | 24.2 | 26.3 | M | ✓ | – | S | – | – |
| QM9 [81] | 129K | 12 | 18.0 | 18.6 | M | ✓ | 3D | S | – | – |
| MOSES [80] | 194K | – | 22 | 47 | M | ✓ | 3D | S | – | – |
| MolOpt [48] | 229K | – | 24 | 53 | M | ✓ | 3D | S | – | – |
| ZINC250K [45] | 250K | – | 23 | 50 | M | ✓ | 3D | S | – | – |
| MolPCBA [105; 38] | 437.9K | 2 | 26.0 | 28.1 | M | ✓ | – | S | – | – |
| PCQM-Contact [24] | 529.4K | – | 30.1 | 61.1 | M | ✓ | 3D | S | – | – |
| ChEMBL [72] | 1.8M | – | 27.0 | 58 | M | ✓ | 3D | S | – | – |
| PCQM4Mv2 [39] | 3.7M | – | 14.1 | 14.6 | L | ✓ | 3D | S | – | – |
| **Social Networks** | | | | | | | | | | |
| IMDB-BINARY [75; 116] | 1K | 2 | 19.8 | 96.5 | S | – | – | S | – | – |
| IMDB-MULTI [75; 116] | 1.5K | 3 | 13.0 | 65.9 | S | – | – | S | – | – |
| REDDIT-BINARY [75; 116] | 2K | 2 | 429.6 | 497.8 | S | – | – | S | – | – |
| REDDIT-MULTI-5K [75; 116] | 5.0K | 5 | 508.5 | 594.9 | M | – | – | S | – | – |
| REDDIT-MULTI-12K [75; 116] | 11.9K | 11 | 11.0 | 391.4 | M | – | – | S | – | – |
| CollabNet [92] | 2.3K | – | 303K | 207.6K | L | – | GCS | T | - | - |
| **Computer Science** | | | | | | | | | | |
| CIFAR10 [23; 57] | 60K | 10 | 117.6 | 941.1 | M | ✓ | 2D | S | – | – |
| MNIST [23; 60] | 70K | 10 | 70.6 | 564.5 | M | ✓ | 2D | S | – | – |
| Database [36] | 300.0K | – | ¡100.0 | ? | M | ? | – | S | – | – |
| MalNet [27] | 1.3M | 696 | 15.4K | 35.2K | L | – | – | S | – | – |
| TpuGraphs (Tile) [78] | 12.9M | – | 40.0 | ? | L | ? | – | S | – | – |
| TpuGraphs (Layout) [78] | 31.1M | – | 7.7K | ? | L | ? | – | S | – | – |
| TenSet [124] | 51.6M | – | 5.0–10.0 | ? | L | ? | – | S | – | – |
| **Geography** | | | | | | | | | | |
| METR-LA [46] | 34K | – | 327.0 | 2.4 | M | ✓ | GCS | T | – | – |
| PeMS-BAY [15] | 50K | – | 207 | 1.5 | M | ✓ | GCS | T | – | – |
| OpenStreetMap [4] | 110K | – | 500 | 1.2K | M | ✓ | GCS | S | ✓ | ✓ |
| **Physics** | | | | | | | | | | |
| N-body-spring [54] | 3.4M | – | 5.0 | 10 | M | ✓ | 2D | T | – | – |
| N-body-charged [54] | 3.4M | – | 25.0 | 3 | M | ✓ | 2D | T | – | – |
| T-HSG-5M | 5.1M | 1401 | 13.8 | 28.9 | L | ✓ | ℂ-plane | T | ✓ | ✓ |
| HSG-12M | 11.6M | 1401 | 13.8 | 28.9 | L | ✓ | ℂ-plane | S | ✓ | ✓ |

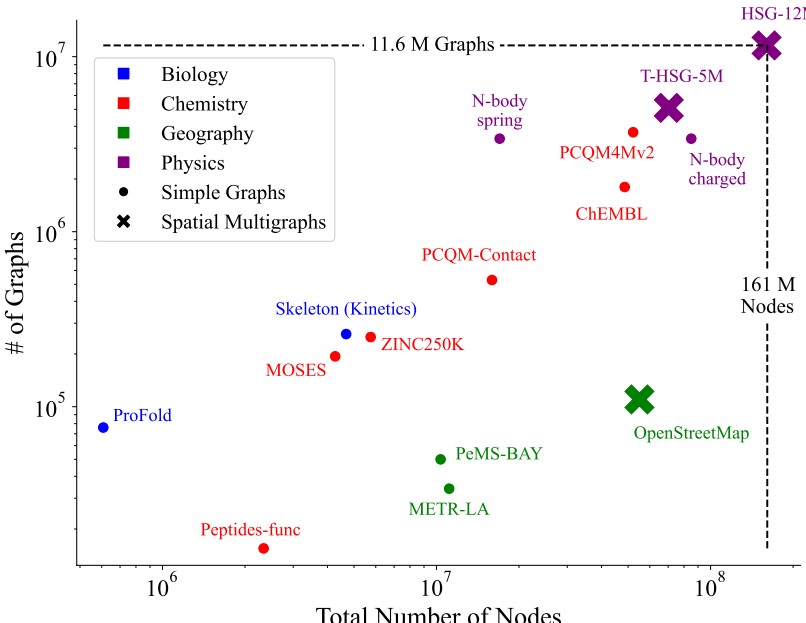

Figure A6: Number of graphs v.s. total number of nodes in HSG-12M compared to other natural science datasets. HSG-12M stands out with the highest data volume across all natural science datasets, including those not designed for classification.

# E  BENCHMARK DETAILS

## E.1  DATA PREPROCESSING (INCLUDING SPATIAL MULTIEDGE FEATURIZATION)

| Feature type | Dim. | Components / Description |
|---|---|---|
| *Node* | 4 | Complex position coordinates (2D) |
| | | Spectral potential at the node (1D) |
| | | DOS at the node (1D) |
| *Edge* | 6 | (Curved) edge length (1D; also used as the edge weight) |
| | | Straight-line distance between endpoints (1D) |
| | | Midpoint coordinates (2D) |
| | | Average spectral potential along the edge (1D) |
| | | Average DOS along the edge (1D) |
| *Graph* | — | None. |

## E.2  TRAINING CONFIGURATION

**Baseline Models.**

- **Graph Convolutional Network (GCN)** (Kipf & Welling, 2017): first-order spectral graph convolution that aggregates normalized neighbor features and applies a shared linear transform.
- **Graph Attention Network (GAT)** (Veličković et al., 2018): masked self-attention over neighborhoods with multi-head weighting to adaptively combine messages.
- **Modified Graph Attention Network (GATv2)** (Brody et al., 2022): reformulated attention with dynamic key–query dependence for greater expressiveness and more stable training.
- **Graph Isomorphism Network (GIN)** (Xu et al., 2019): sum-aggregating message passing with an MLP update, designed for expressivity comparable to the Weisfeiler–Lehman test.
- **Edge-conditioned Graph Isomorphism Network (GINE)** (Hu et al., 2019): GIN variant that injects edge features into the message MLP, improving performance on edge-featured graphs.

- **Molecular Fingerprints (MF)** (Duvenaud et al., 2015): uses learnable neighborhood transforms and softmax pooling, yielding a fixed-length real-valued fingerprint aggregated over nodes and layers.

- **Crystal Graph Convolutional Neural Network (CGCNN)** (Xie & Grossman, 2018): crystal-graph convolutions over periodic structures (periodic crystal graph) that aggregate atom–bond interactions within a distance cutoff for materials prediction.

- **GraphSAGE** (Hamilton et al., 2017): inductive neighbor-sampling with learnable aggregators (e.g., mean) enabling generalization to unseen graphs.

All architectures are pooled via global add pooling, followed by a multi-layer perceptron (MLP) to produce the final class logits.

**Global training setup.** Data splits, training hyperparameters, optimizer, learning rate scheduler, hardwares, trainer settings, and common model hyperparameters are:

| Group | Hyperparameter | Value / Notes |
|---|---|---|
| Data | Splits ratio | Train:Val:Test = 8:1:1 |
| | Split strategy | Stratified random splits (preserve class/target proportions across train/val/test) |
| Training | Batch size | 6000 |
| | Max epochs | 100 |
| | Max steps | 1000 |
| | Seeds | $\{42, 2025, 666\}$ |
| Model (common) | #Conv layers | 4 |
| | #MLP layers | 2 |
| | Activation | ReLU |
| | Dropout | 0.0 |
| | #Heads (GAT/GATv2) | 1 |
| Optimizer | Algorithm | AdamW (AMSGrad) [69] |
| | Init LR $\eta_0$ | $1 \times 10^{-3}$ |
| | Min LR $\eta_{\min}$ | $1 \times 10^{-5}$ |
| | Weight decay | 0.0 |
| Scheduler | Policy | Cosine annealing [68] |
| | Period $T_0$ | 100 |
| Trainer | Devices | Two RTX A5000, 24GB each |
| | Strategy | Distributed Data Parallel (DDP) |

**Dataset-specific hyperparameter: hidden dimension of post-convolution MLP.** The post-convolution MLP hidden dimension is tuned to be larger than the number of classes per dataset, to ensure sufficient capacity for final classification.

| $\mathbf{dim}_h^{\mathbf{MLP}}$ | one-band | two-band | three-band | topology | HSG-12M |
|---|---|---|---|---|---|
| Value | 128 | 256 | 1500 | 1500 | 1500 |

**Model-specific hyperparameters: hidden dimension of graph convolution layers.** To ensure fair comparison, for each dataset, we tune the hidden dimension of graph convolution layers for each model such that the total number of trainable parameters is within 4% relative difference across all models.

| $\dim_h^{\text{Conv}}$ | MF | GCN | SAGE | GAT | GIN | GINE | CGCNN | GATv2 |
|---|---|---|---|---|---|---|---|---|
| HSG-one-band | 100 | 467 | 330 | 452 | 312 | 312 | 202 | 330 |
| HSG-two-band | 200 | 933 | 661 | 933 | 621 | 621 | 410 | 661 |
| HSG-three-band | 300 | 1279 | 963 | 1279 | 852 | 852 | 601 | 963 |
| HSG-topology | 300 | 1279 | 963 | 1279 | 852 | 852 | 601 | 963 |
| HSG-12M | 300 | 1279 | 963 | 1279 | 852 | 852 | 601 | 963 |

### E.3 EVALUATION METRICS

We evaluate single-label multiclass prediction over a dataset $\mathcal{D} = \{(\mathcal{G}_i, y_i)\}_{i=1}^n$ with $y_i \in \{1, \ldots, C\}$. Let $s_{i,c}$ be the model score (logit or probability) for class $c$ on example $i$, and $\hat{y}_i = \arg\max_c s_{i,c}$.

**Accuracy (micro-$F_1$).** Overall fraction of correct predictions:

$$\text{Acc} = \frac{1}{n} \sum_{i=1}^n \mathbb{I}\{\hat{y}_i = y_i\}. \tag{22}$$

In single-label multiclass settings, Accuracy equals the micro-averaged $F_1$. It is intuitive and stable when class frequencies are roughly balanced. The chance baseline is $1/C$.

**Macro $F_1$.** Compute one-vs-rest counts for each class $c$:

$$\text{TP}_c = \sum_i \mathbb{I}\{y_i = c, \ \hat{y}_i = c\}, \quad \text{FP}_c = \sum_i \mathbb{I}\{y_i \neq c, \ \hat{y}_i = c\}, \quad \text{FN}_c = \sum_i \mathbb{I}\{y_i = c, \ \hat{y}_i \neq c\}. \tag{23}$$

Per-class precision/recall and $F_1$, with $0/0 \equiv 0$:

$$P_c = \frac{\text{TP}_c}{\text{TP}_c + \text{FP}_c}, \qquad R_c = \frac{\text{TP}_c}{\text{TP}_c + \text{FN}_c}, \qquad F1_c = \frac{2 P_c R_c}{P_c + R_c}. \tag{24}$$

Macro-average across classes:

$$\text{Macro-}F_1 = \frac{1}{C} \sum_{c=1}^C F1_c. \tag{25}$$

Macro-$F_1$ weights all classes equally and is therefore sensitive to minority-class performance—*crucial when $C$ is large or labels are imbalanced*.

**Top-$k$ Accuracy.** Let $\text{TopK}(\{s_{i,c}\}_{c=1}^C, k)$ denote the indices of the $k$ largest scores. The Top-$k$ metric is

$$\text{Top-}k = \frac{1}{n} \sum_{i=1}^n \mathbb{I}\big\{ y_i \in \text{TopK}(\{s_{i,c}\}_{c=1}^C, \ k) \big\}. \tag{26}$$

We report $k \in \{5, 10\}$. Top-$k$ captures ranking quality and is directly aligned with inverse-design workflows that accept a shortlist for subsequent physics-based re-ranking. Random-guess baselines scale as $k/C$ (e.g., $5/24 \approx 20.83\%$, $10/24 \approx 41.67\%$; for $C = 1401$: $5/1401 \approx 0.357\%$, $10/1401 \approx 0.714\%$).

**Reporting and interpretation.** All metrics are reported as mean$_{\pm\text{std}}$ averaged over seeds on the held-out test split. For smaller or moderately balanced label spaces, Accuracy is informative and easy to compare; for highly imbalanced or very large $C$, Macro-$F_1$ is emphasized to surface minority-class recall, while Top-5/10 quantify the usefulness of the model as a candidate-generator for downstream, physics-constrained refinement.

### E.4 ADDITIONAL BENCHMARK RESULTS AND ANALYSIS

Tables A5-A8 reports additional benchmark results including test loss, test top-5, and training statistics, including throughput and device utilization.

**Overall accuracy and stability.** Across all five static variants, seed variance is consistently small, indicating stable training.

**Edge attributes matter.** Methods that explicitly consume edge features (e.g. GINE) are consistently superior to their edge-agnostic counterparts (e.g. GIN). For example on `two-band`, GINE ($.518_{\pm.049}$) outperforms GIN ($.343_{\pm.084}$). On `HSG-12M`, plain GIN essentially collapses ($.063_{\pm.031}$), while GINE remains competitive ($.460_{\pm.025}$). This aligns with the dataset design: multi-edge geometry (length, straight-line distance, midpoint, average spectral potential, average DOS) carries irreducible spatial information; architectures which propagate and transform edge states are expected to succeed.

**Performance degrades with task difficulty.** Averaging over models, test metrics degrade from `one-band`→`two-band`→`topology`→`three-band`→`HSG-12M`. This monotonic decay is expected: higher-band Hamiltonians induce larger graphs with richer multi-edge geometry and more challenging class diversity (up to 1,401), stressing both representation and optimization. Moreover, as expected, per-graph memory usage scales with dataset complexity for every model (e.g., SAGE: 0.066 on `one-band` → 0.544 on `three-band`).

**Top-k is high—promising for inverse design.** Despite moderate Top-1 accuracy on the largest settings, *Top-10 accuracy is very high* (e.g., on `HSG-12M` SAGE 95.2%, CGCNN 94.8%). For easier subsets, Top-10 essentially saturates (99%+ on `one-band` for all models). This pattern implies that models almost always retrieve a **small candidate set** of plausible Hamiltonian families. **This is encouraging for inverse design workflows (retrieve top-$k$ families, then re-rank/verify physically in experiments, e.g. design a few meta-matetial candidates and observe if targeted spectral properties are obtained).**

**Attention is not a free lunch here.** GAT / GATv2 lag SAGE / CGCNN on all splits, while also incurring the **highest peak GPU memory per graph** (e.g., on `HSG-12M` ∼1.93–2.11 vs SAGE's 0.51). In spatial multigraphs with high effective degrees (many parallel edges), attention softmaxes can become dominated by edge multiplicity/noise and impose additional compute/memory overhead without commensurate accuracy gains under our budgets. In other words, dense multi-edge neighborhoods amplify attention's quadratic costs and may dilute useful geometric cues when our **direction-ignorant summary features** are the only edge signal.

**GraphSAGE excels with limited budget and comparable parameters constraint.** GraphSAGE is consistently the strongest baseline. It attains the best Top-1 accuracy and macro-F$_1$ on every subset. Under matched trainable parameter constraints ($\leq$4% difference) and a fixed training budget (`max_epochs` = 100, `max_steps` = 1000), these consistent gains suggest that:

1. Either, other more expressive architectures (e.g., attentive or edge-MLP-based) require larger training budgets to fully realize their potential;
2. Or, other expressive baselines are more parameter-hungry, and thus under trainable parameter constraints, they are not as efficient as lightweight baselines for large-scale benchmarks;
3. Or surprisingly, GraphSAGE's neighborhood aggregation is a better inductive bias for our spatial multigraphs than attention or vanilla spectral convolutions.

One could explore relaxing the fixed budget and hyperparameter optimization to unlock each architecture's full potential. However this is far beyond the computing resources currently available to us, and we leave this to future work.

**Implications for spatial multigraph learning.** With our **fixed-size, direction-agnostic** edge summaries (length, straight-line distance, midpoint, avg. potential/DOS), relatively **simple, locality-biased** architectures already capture much of the discriminative signal.

However, the persistent Top-1↔Top-10 gap between edge-aware baselines and their edge-agnostic counterparts, and difficulty at high class diversity, together indicate that **fine-grained geometric information along multi-edge curves** (e.g., curvature, torsion, higher-order moments, spline/Bezier edge parameterizations, or sequence encodings of the edge polyline) are promising routes to push performance further, especially on large-scale challenges like `three-band` and `HSG-12M`.

Table A5: Additional Graph-level classification results for **Test Loss** on the HSG dataset variants. Cells show mean$_{\pm\text{std}}$ over three random seeds; best model per dataset in **Bold**.

| Model | one-band | two-band | three-band | topology | HSG-12M |
|---|---|---|---|---|---|
| GCN | $.723_{\pm.018}$ | $1.695_{\pm.047}$ | $2.522_{\pm.063}$ | $2.062_{\pm.062}$ | $2.357_{\pm.089}$ |
| GAT | $.841_{\pm.008}$ | $1.776_{\pm.034}$ | $2.465_{\pm.068}$ | $1.825_{\pm.053}$ | $2.330_{\pm.047}$ |
| GATv2 | $.926_{\pm.018}$ | $1.853_{\pm.018}$ | $2.809_{\pm.171}$ | $1.968_{\pm.025}$ | $2.401_{\pm.011}$ |
| GIN | $.494_{\pm.017}$ | $2.216_{\pm.402}$ | $4.926_{\pm.422}$ | $4.179_{\pm.612}$ | $4.764_{\pm.711}$ |
| GINE | $.554_{\pm.027}$ | $1.466_{\pm.183}$ | $2.298_{\pm.091}$ | $1.316_{\pm.057}$ | $1.799_{\pm.124}$ |
| MF | $1.056_{\pm.019}$ | $2.392_{\pm.089}$ | $2.852_{\pm.026}$ | $2.222_{\pm.051}$ | $2.658_{\pm.063}$ |
| CGCNN | $.474_{\pm.010}$ | $1.218_{\pm.104}$ | $1.730_{\pm.063}$ | $1.191_{\pm.065}$ | $1.485_{\pm.021}$ |
| GraphSAGE | $\mathbf{.355_{\pm.011}}$ | $\mathbf{.932_{\pm.022}}$ | $\mathbf{1.561_{\pm.073}}$ | $\mathbf{1.019_{\pm.002}}$ | $\mathbf{1.434_{\pm.009}}$ |

Table A6: Additional Graph-level classification results for **Top-5 Accuracy** on the HSG dataset variants. Cells show mean$_{\pm\text{std}}$ over three random seeds; best model per dataset in **Bold**.

| Model | one-band | two-band | three-band | topology | HSG-12M |
|---|---|---|---|---|---|
| GCN | $.988_{\pm.001}$ | $.847_{\pm.009}$ | $.693_{\pm.015}$ | $.715_{\pm.007}$ | $.725_{\pm.020}$ |
| GAT | $.981_{\pm.001}$ | $.830_{\pm.007}$ | $.702_{\pm.016}$ | $.752_{\pm.014}$ | $.726_{\pm.009}$ |
| GATv2 | $.973_{\pm.002}$ | $.815_{\pm.003}$ | $.627_{\pm.038}$ | $.723_{\pm.005}$ | $.711_{\pm.001}$ |
| GIN | $.995_{\pm.000}$ | $.746_{\pm.087}$ | $.184_{\pm.066}$ | $.276_{\pm.127}$ | $.222_{\pm.098}$ |
| GINE | $.996_{\pm.001}$ | $.883_{\pm.029}$ | $.755_{\pm.015}$ | $.848_{\pm.013}$ | $.830_{\pm.018}$ |
| MF | $.974_{\pm.001}$ | $.703_{\pm.026}$ | $.616_{\pm.006}$ | $.673_{\pm.011}$ | $.655_{\pm.015}$ |
| CGCNN | $.995_{\pm.001}$ | $.920_{\pm.012}$ | $.836_{\pm.009}$ | $.870_{\pm.008}$ | $.876_{\pm.003}$ |
| GraphSAGE | $\mathbf{.998_{\pm.000}}$ | $\mathbf{.953_{\pm.002}}$ | $\mathbf{.863_{\pm.012}}$ | $\mathbf{.898_{\pm.002}}$ | $\mathbf{.882_{\pm.002}}$ |

Table A7: Additional Graph-level classification results for **Throughput (graphs sec$^{-1}$)** on the HSG dataset variants. Cells show mean$_{\pm\text{std}}$ over three random seeds; best model per dataset in **Bold**.

| Model | one-band | two-band | three-band | topology | HSG-12M |
|---|---|---|---|---|---|
| GCN | $113580_{\pm1840}$ | $83975_{\pm717}$ | $50803_{\pm238}$ | $44065_{\pm53}$ | $52945_{\pm199}$ |
| GAT | $103757_{\pm879}$ | $62717_{\pm302}$ | $31885_{\pm25}$ | $25862_{\pm29}$ | $34193_{\pm283}$ |
| GATv2 | $104282_{\pm2065}$ | $63092_{\pm216}$ | $30074_{\pm11}$ | $23825_{\pm15}$ | $32051_{\pm112}$ |
| GIN | $111905_{\pm900}$ | $84681_{\pm354}$ | $61019_{\pm469}$ | $55879_{\pm98}$ | $62134_{\pm955}$ |
| GINE | $112075_{\pm2619}$ | $81356_{\pm1343}$ | $52408_{\pm186}$ | $45984_{\pm81}$ | $53332_{\pm99}$ |
| MF | $112493_{\pm3786}$ | $\mathbf{85980_{\pm1154}}$ | $\mathbf{65223_{\pm710}}$ | $\mathbf{61507_{\pm232}}$ | $\mathbf{66829_{\pm1729}}$ |
| CGCNN | $105543_{\pm1818}$ | $62122_{\pm280}$ | $26752_{\pm17}$ | $20414_{\pm19}$ | $28671_{\pm226}$ |
| GraphSAGE | $\mathbf{113781_{\pm1797}}$ | $85590_{\pm637}$ | $54473_{\pm266}$ | $48479_{\pm73}$ | $56064_{\pm533}$ |

Table A8: Additional Graph-level classification results for **Average Peak GPU Memory (MB/-graph)** on the HSG dataset variants. Cells show mean$_{\pm\text{std}}$ over three random seeds; best model per dataset in **Bold**.

| Model | one-band | two-band | three-band | topology | HSG-12M |
|---|---|---|---|---|---|
| GCN | $.0851_{\pm.0001}$ | $.3308_{\pm.0002}$ | $.7274_{\pm.0014}$ | $.8824_{\pm.0015}$ | $.6713_{\pm.0017}$ |
| GAT | $.2418_{\pm.0001}$ | $1.0276_{\pm.0003}$ | $2.3011_{\pm.0021}$ | $2.8788_{\pm.0040}$ | $2.1065_{\pm.0056}$ |
| GATv2 | $.2100_{\pm.0002}$ | $.8809_{\pm.0002}$ | $2.1102_{\pm.0014}$ | $2.6580_{\pm.0036}$ | $1.9301_{\pm.0056}$ |
| GIN | $.0882_{\pm.0002}$ | $.2954_{\pm.0002}$ | $.6375_{\pm.0014}$ | $.7535_{\pm.0014}$ | $.5918_{\pm.0009}$ |
| GINE | $.1209_{\pm.0001}$ | $.4816_{\pm.0001}$ | $1.0676_{\pm.0016}$ | $1.3142_{\pm.0022}$ | $.9812_{\pm.0026}$ |
| MF | $\mathbf{.0494_{\pm.0002}}$ | $\mathbf{.1746_{\pm.0001}}$ | $\mathbf{.4054_{\pm.0006}}$ | $\mathbf{.4701_{\pm.0006}}$ | $\mathbf{.3772_{\pm.0011}}$ |
| CGCNN | $.1876_{\pm.0000}$ | $.7981_{\pm.0004}$ | $1.9284_{\pm.0016}$ | $2.4368_{\pm.0041}$ | $1.7636_{\pm.0050}$ |
| GraphSAGE | $.0669_{\pm.0002}$ | $.2407_{\pm.0001}$ | $.5512_{\pm.0012}$ | $.6542_{\pm.0011}$ | $.5108_{\pm.0008}$ |

### E.5 Physical Motivation of the Benchmark Task

In crystalline, mesoscopic solids and meta-materials, observable spectral signatures—band dispersions, density of states $\rho(E)$—are generated by an underlying Hamiltonian $\boldsymbol{H}(z)$ whose structure is dictated by crystal geometry, orbital content, and the pattern of allowed hoppings. The forward map

$$\text{poly2graph} : \boldsymbol{H} \longmapsto \mathcal{G}(\boldsymbol{H}) \tag{27}$$

from a tight-binding (or effective) Hamiltonian to a *spectral graph* $\mathcal{G}$ is deterministic yet typically many-to-one: different microscopic parameterizations inside the same Hamiltonian family—more precisely, the same *hopping-pattern* can induce very similar spectra.

**For materials discovery and interpretation of experiments, one interesting and practically relevant question is the *inverse* problem of Hamiltonian inference from spectral data**:

> *Given a desired spectral signature (the Hamiltonian spectral graph), what "class" of Hamiltonians—what material structure/hopping pattern—could realize it?*

We cast this inverse-design query as supervised *categorical retrieval* from spectral graphs to a discrete characteristic polynomial (ChP) class (a "hopping-pattern"). Learning a predictor

$$f_\theta : \ \mathcal{G} \mapsto \mathcal{C}_P \tag{28}$$

coarse-grains the ill-posed inverse map into a small set of plausible Hamiltonian families. High Top-$k$ accuracy means we can enumerate $\text{TopK}(f_\theta(G), k)$ as a compact candidate list, thus reducing searching over a huge number of hopping patterns to a manageable shortlist.

This framing is physically meaningful for two reasons. *First*, the spectral features that guide human intuition (e.g. gaps and gap sizes) are controlled primarily by lattice symmetries and connectivity rather than precise parameter values; predicting the *family* is therefore the right first step. *Second*, our spatial multigraph featurization of spectra (node- and edge-level geometric and spectral statistics) is engineered to expose invariants that tie back to local real-space structure, allowing GNNs to learn robust surrogates of poly2graph$^{-1}$.

Beyond enabling inverse design, the HSG provides a large-scale resource to study how Hamiltonian parameters control spectral-graph morphology and to pre-train scientific foundation models on physically grounded graph signals, contributing to AI4Science community (Yan, 2025; Yan & Pan, 2025). In short, the mapping Spectral Graph → ChP Class directly operationalizes a task that materials physicists and chemists already perform by hand, but at scale and with principled uncertainty via Top-$k$ retrieval.

## F  Universality of Spectral Graphs Through Toeplitz Decomposition

At the heart of our framework is the **Poly2Graph** algorithm, a function that establishes a direct mapping from the algebraic domain of polynomials and matrices to the structural domain of spectral graphs. This connection is most naturally illustrated with Toeplitz matrices. As demonstrated in appendix B, a generic Toeplitz matrix can be interpreted as a single-band tight-binding Hamiltonian (Eq.8), which corresponds to a unique signature spectral graph.

The significance of this specific result is vastly amplified by a foundational theorem in linear algebra:

> *Any matrix can be expressed as a product of Toeplitz matrices (Ye & Lim, 2015).*

Specifically, for any matrix $\boldsymbol{M} \in \mathbb{C}^{n \times n}$, a decomposition into a product of $r$ Toeplitz matrices exists, where $\lfloor n/2 \rfloor + 1 \le r \le 2n + 5$. While this decomposition is not unique without further constraints, its existence is guaranteed.

Consequently, by associating a spectral graph with each Toeplitz component, we can represent any arbitrary matrix as a *multiset* of these graphs. Although the decomposition is not unique, one can define the smallest *multiset* as the *canonical* topology signature of the input system. Henceforth, this provides a universal procedure for translating complex matrices into a graph-based representation:

1. Start with a generic matrix $M \in \mathbb{C}^{n \times n}$.

2. Decompose $M$ into a product of $r$ Toeplitz matrices, $M = T_1 T_2 \cdots T_r$.

3. Apply the `Poly2Graph` algorithm to each Toeplitz factor $T_i$ to extract its corresponding spectral graph $\mathcal{G}_i$. The resulting multiset $\{\mathcal{G}_1, \ldots, \mathcal{G}_r\}$ is the spectral graph representation of $M$.

This procedure highlights the remarkable generality of our framework. Since the characteristics of most scientific systems can be expressed as a matrix, polynomial, or more generically a vector (which can be standardized and treated as the coefficients of a univariate polynomial), our approach offers a novel analytical lens. This opens up new avenues for research across numerous fields, a direction we are actively exploring and invite the broader community to join.

## G  TUTORIAL OF POLY2GRAPH PACKAGE

`poly2graph` is a Python package for automatic *Hamiltonian spectral graph* construction. It takes in a characteristic polynomial or a Bloch Hamiltonian and returns the spectral graph.

### G.1  FEATURES

- High-performance
    - Fast construction of spectral graph from any one-dimensional models
    - Adaptive resolution to reduce floating operation cost and memory usage
    - Automatic backend for computation bottleneck. If `tensorflow` / `torch` is available, any device (e.g. `/GPU:0`, `/TPU:0`, `cuda:0`, etc.) that they support can be used for acceleration.
- Cover generic topological lattices
    - Support generic one-band and multi-band models
    - Flexible multiple input choices, be they characteristic polynomials or Bloch Hamiltonians; formats include strings, `sympy.Poly`, and `sympy.Matrix`
- Automatic and Robust
    - By default, no hyper-parameters are needed. Just input the characteristic of your model and `poly2graph` handles the rest
    - Automatic spectral boundary inference
    - Relatively robust on multiband models that are prone to "component fragmentation"
- Helper functionalities generally useful
    - `skeleton2graph` module: Convert a skeleton image to its graph representation
    - `hamiltonian` module: Conversion among different Hamiltonian representations and efficient computation of a range of properties

### G.2  INSTALLATION

You can install the package via pip:

```
1  $ pip install poly2graph
```

or clone the repository and install it manually:

```
1  $ git clone https://github.com/sarinstein-yan/poly2graph.git
2  $ cd poly2graph
3  $ pip install -e .
```

Optionally, if TensorFlow or PyTorch is available, `poly2graph` will make use of them automatically to accelerate the computation bottleneck. Priority: `tensorflow > torch > numpy`.

Check the installation:

```python
import poly2graph as p2g
print(p2g.__version__)
```

## G.3 USAGE

See the Poly2Graph Tutorial JupyterNotebook for a quick interactive start.

`p2g.SpectralGraph` and `p2g.CharPolyClass` are the two main classes in the package.

`p2g.SpectralGraph` investigates the spectral graph topology of **a specific** given characteristic polynomial or Bloch Hamiltonian. `p2g.CharPolyClass` investigates **a class** of **parametrized** characteristic polynomials or Bloch Hamiltonians, and is optimized for generating spectral properties in parallel.

```python
import numpy as np
import networkx as nx
import sympy as sp
import matplotlib.pyplot as plt
from matplotlib import colors
# always start by initializing the symbols for k, z, and E
k = sp.symbols('k', real=True)
z, E = sp.symbols('z E', complex=True)
```

### G.3.1 A GENERIC **ONE-BAND** EXAMPLE (P2G.SPECTRALGRAPH):

Characteristic polynomial:

$$P(E, z) := h(z) - E = z^4 - z - z^{-2} - E$$

Its Bloch Hamiltonian (Fourier transformed Hamiltonian in momentum space) is a scalar function:

$$h(z) = z^4 - z - z^{-2}$$

where the phase factor is defined as $z := e^{ik}$.

Expressed in terms of crystal momentum $k$:

$$h(k) = e^{4ik} - e^{ik} - e^{-2ik}$$

The valid input formats to initialize a `p2g.SpectralGraph` object are:

1. Characteristic polynomial in terms of `z` and `E`:
    - as a string of the Poly in terms of `z` and `E`
    - as a `sympy.Poly` with $\{z, 1/z, E\}$ as generators
2. Bloch Hamiltonian in terms of `k` or `z`
    - as a `sympy.Matrix` in terms of `k`
    - as a `sympy.Matrix` in terms of `z`

All the following `characteristics` are valid and will initialize to the same characteristic polynomial and therefore produce the same spectral graph:

```
1  char_poly_str = '-z**-2 - E - z + z**4'
2
3  char_poly_Poly = sp.Poly(
4      -z**-2 - E - z + z**4,
5      z, 1/z, E # generators are z, 1/z, E
6  )
7
8  phase_k = sp.exp(sp.I*k)
9  char_hamil_k = sp.Matrix([-phase_k**2 - phase_k + phase_k**4])
10
11 char_hamil_z = sp.Matrix([-z**-2 - E - z + z**4])
```

Let us just use the string to initialize and see a set of properties that are computed automatically:

```
1  sg = p2g.SpectralGraph(char_poly_str, k=k, z=z, E=E)
```

**Characteristic polynomial**:

```
1  sg.ChP
```

$$>>> \text{Poly}\left(z^4 - z - \tfrac{1}{z^2} - E,\ z, \tfrac{1}{z}, E,\ domain = \mathbb{Z}\right)$$

**Bloch Hamiltonian**:

- For one-band model, it is a unique, rank-0 matrix (scalar)

```
1  sg.h_k
```

$>>>$
$$\left[e^{4ik} - e^{ik} - e^{-2ik}\right]$$

```
1  sg.h_z
```

$>>>$
$$\left[-\tfrac{-z^6 + z^3 + 1}{z^2}\right]$$

**The Frobenius companion matrix of `P(E)(z)`**:

- treating `E` as parameter and `z` as variable

- Its eigenvalues are the roots of the characteristic polynomial at a fixed complex energy `E`. Thus it is useful to calculate the GBZ (generalized Brillouin zone), the spectral potential (Ronkin function), etc.

```
1  sg.companion_E
```

>>>

$$\begin{bmatrix} 0 & 0 & 0 & 0 & 0 & 1 \\ 1 & 0 & 0 & 0 & 0 & 0 \\ 0 & 1 & 0 & 0 & 0 & E \\ 0 & 0 & 1 & 0 & 0 & 1 \\ 0 & 0 & 0 & 1 & 0 & 0 \\ 0 & 0 & 0 & 0 & 1 & 0 \end{bmatrix}$$

**Number of bands & hopping range**:

```python
print('Number of bands:', sg.num_bands)
print('Max hopping length to the right:', sg.poly_p)
print('Max hopping length to the left:', sg.poly_q)
```

>>>

```
Number of bands: 1
Max hopping length to the right: 2
Max hopping length to the left: 4
```

**A real-space Hamiltonian of a finite chain and its energy spectrum**:

```python
H = sg.real_space_H(
    N=40,         # number of unit cells
    pbc=False,    # open boundary conditions
    max_dim=500   # maximum dimension of the Hamiltonian matrix (for
    ↪ numerical accuracy)
)

energy = np.linalg.eigvals(H)

fig, ax = plt.subplots(figsize=(3, 3))
ax.plot(energy.real, energy.imag, 'k.', markersize=5)
ax.set(xlabel='Re(E)', ylabel='Im(E)', \
xlim=sg.spectral_square[:2], ylim=sg.spectral_square[2:])
plt.tight_layout(); plt.show()
```

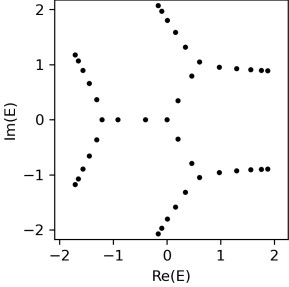

THE SET OF SPECTRAL FUNCTIONS

```python
phi, dos, binaried_dos = sg.spectral_images()
```

```
3  fig, axes = plt.subplots(1, 3, figsize=(8, 3), sharex=True,
   ↪  sharey=True)
4  axes[0].imshow(phi, extent=sg.spectral_square, cmap='terrain')
5  axes[0].set(xlabel='Re(E)', ylabel='Im(E)', title='Spectral Potential')
6  p2, p98 = np.percentile(dos, (2, 98))
7  # ˆ Clip extreme DOS to increase visibility.
8  norm = colors.Normalize(vmin=p2, vmax=p98)
9  axes[1].imshow(dos, extent=sg.spectral_square, cmap='viridis',
   ↪  norm=norm)
10 axes[1].set(xlabel='Re(E)', title='Density of States')
11
12 axes[2].imshow(binaried_dos, extent=sg.spectral_square, cmap='gray')
13 axes[2].set(xlabel='Re(E)', title='Graph Skeleton')
14 plt.tight_layout()
15 plt.show()
```

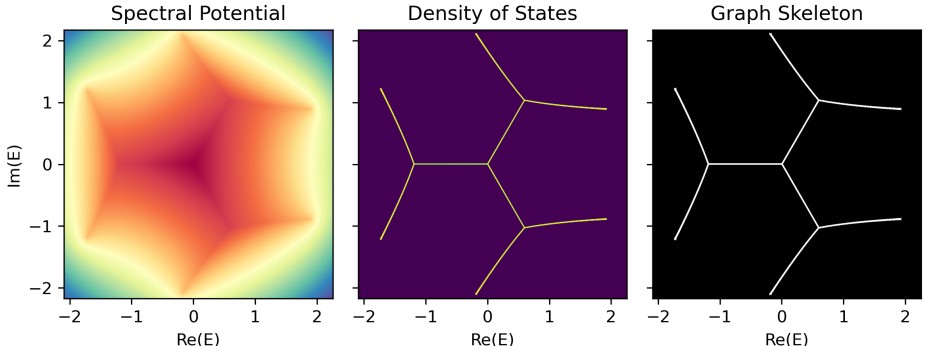

### THE SPECTRAL GRAPH $\mathcal{G}$

```
1  graph = sg.spectral_graph()
2
3  fig, ax = plt.subplots(figsize=(3, 3))
4  pos = nx.get_node_attributes(graph, 'pos')
5  nx.draw_networkx_nodes(graph, pos, alpha=0.8, ax=ax,
6              node_size=50, node_color='#A60628')
7  nx.draw_networkx_edges(graph, pos, alpha=0.8, ax=ax,
8              width=5, edge_color='#348ABD')
9  plt.tight_layout(); plt.show()
```

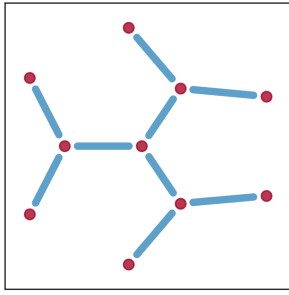

If `tensorflow` or `torch` is available, `poly2graph` will automatically use them and run on **CPU** by default. If other device, e.g. **GPU / TPU** is available, one can pass `device = {device string}` to the method `spectral_images` and `spectral_graph`:

```
SpectralGraph.spectral_images(device='/cpu:0')
SpectralGraph.spectral_graph(device='/gpu:1')
SpectralGraph.spectral_images(device='cpu')
SpectralGraph.spectral_graph(device='cuda:0')
...
```

However, some functions may not have gpu kernel in `tf`/`torch`, in which case the computation will fallback to CPU.

### G.3.2   A GENERIC **MULTI-BAND** EXAMPLE (`P2G.SPECTRALGRAPH`):

Characteristic polynomial (four bands):
$$P(E, z) := \det(\mathbf{h}(z) - E\,\mathbf{I}) = z^2 + 1/z^2 + Ez - E^4$$

One of its possible Bloch Hamiltonians in terms of $z$:
$$\mathbf{h}(z) = \begin{bmatrix} 0 & 0 & 0 & z^2 + 1/z^2 \\ 1 & 0 & 0 & z \\ 0 & 1 & 0 & 0 \\ 0 & 0 & 1 & 0 \end{bmatrix}$$

```
sg_multi = p2g.SpectralGraph("z**2 + 1/z**2 + E*z - E**4", k, z, E)
```

**Characteristic polynomial**:

```
sg_multi.ChP
```

$$>>> \text{Poly}\left(z^2 + zE + \tfrac{1}{z^2} - E^4,\ z,\ \tfrac{1}{z}, E,\ domain = \mathbb{Z}\right)$$

**Bloch Hamiltonian**:

- For multi-band model, if the `p2g.SpectralGraph` is not initialized with a `sympy Matrix`, then `poly2graph` will use the companion matrix of the characteristic polynomial `P(z)(E)` (treating `z` as parameter and `E` as variable) as the Bloch Hamiltonian – this is one of the set of possible band Hamiltonians that possesses the same energy spectrum and thus the same spectral graph.

```
sg_multi.h_k
```

>>>
$$\begin{bmatrix} 0 & 0 & 0 & 2\cos(2k) \\ 1 & 0 & 0 & e^{ik} \\ 0 & 1 & 0 & 0 \\ 0 & 0 & 1 & 0 \end{bmatrix}$$

```
sg_multi.h_z
```

>>>

$$\begin{bmatrix} 0 & 0 & 0 & z^2 + \frac{1}{z^2} \\ 1 & 0 & 0 & z \\ 0 & 1 & 0 & 0 \\ 0 & 0 & 1 & 0 \end{bmatrix}$$

---

**The Frobenius companion matrix of `P(E)(z)`**:

```
sg_multi.companion_E
```

>>>

$$\begin{bmatrix} 0 & 0 & 0 & -1 \\ 1 & 0 & 0 & 0 \\ 0 & 1 & 0 & E^4 \\ 0 & 0 & 1 & -E \end{bmatrix}$$

---

**Number of bands & hopping range**:

```
print('Number of bands:', sg_multi.num_bands)
print('Max hopping length to the right:', sg_multi.poly_p)
print('Max hopping length to the left:', sg_multi.poly_q)
```

>>>

```
Number of bands: 4
Max hopping length to the right: 2
Max hopping length to the left: 2
```

---

**A real-space Hamiltonian of a finite chain and its energy spectrum**:

```python
H_multi = sg_multi.real_space_H(
    N=40,          # number of unit cells
    pbc=False,     # open boundary conditions
    max_dim=500    # maximum dimension of the Hamiltonian matrix (for
      ↪ numerical accuracy)
)

energy_multi = np.linalg.eigvals(H_multi)

fig, ax = plt.subplots(figsize=(3, 3))
ax.plot(energy_multi.real, energy_multi.imag, 'k.', markersize=5)
ax.set(xlabel='Re(E)', ylabel='Im(E)', \
xlim=sg_multi.spectral_square[:2], ylim=sg_multi.spectral_square[2:])
plt.tight_layout(); plt.show()
```

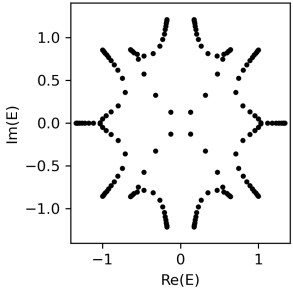

### THE SET OF SPECTRAL FUNCTIONS

```
phi_multi, dos_multi, binaried_dos_multi =
↪   sg_multi.spectral_images(device='/cpu:0')

fig, axes = plt.subplots(1, 3, figsize=(8, 3), sharex=True,
↪   sharey=True)
axes[0].imshow(phi_multi, extent=sg_multi.spectral_square,
↪   cmap='terrain')
axes[0].set(xlabel='Re(E)', ylabel='Im(E)', title='Spectral Potential')
axes[1].imshow(dos_multi, extent=sg_multi.spectral_square,
↪   cmap='viridis', norm=norm)
axes[1].set(xlabel='Re(E)', title='Density of States')
axes[2].imshow(binaried_dos_multi, extent=sg_multi.spectral_square,
↪   cmap='gray')
axes[2].set(xlabel='Re(E)', title='Graph Skeleton')
plt.tight_layout(); plt.show()
```

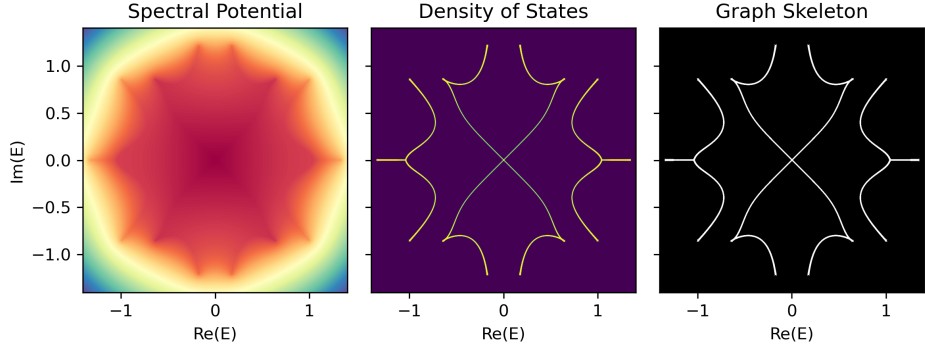

### THE SPECTRAL GRAPH $\mathcal{G}$

```
graph_multi = sg_multi.spectral_graph(
    short_edge_threshold=20,
    # ^ node pairs or edges with distance < threshold pixels are merged
)

fig, ax = plt.subplots(figsize=(3, 3))
pos_multi = nx.get_node_attributes(graph_multi, 'pos')
nx.draw(graph_multi, pos_multi, ax=ax,
        node_size=10, node_color='#A60628',
        edge_color='#348ABD', width=2, alpha=0.8)
```

```
11  plt.tight_layout(); plt.show()
```

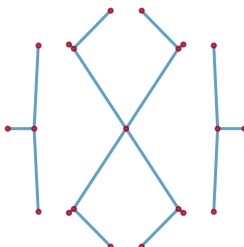

### G.3.3  NODE AND EDGE ATTRIBUTES OF THE SPECTRAL GRAPH OBJECT

The spectral graph is a `networkx.MultiGraph` object.

- Node Attributes
    1. `pos` : (2,)-numpy array
        - the position of the node $(\mathrm{Re}(E), \mathrm{Im}(E))$
    2. `dos` : float
        - the density of states at the node
    3. `potential` : float
        - the spectral potential at the node
- Edge Attributes
    1. `weight` : float
        - the weight of the edge, which is the **length** of the edge in the complex energy plane
    2. `pts` : (w, 2)-numpy array
        - the positions of the points constituting the edge, where w is the number of points along the edge, i.e., the length of the edge, equals `weight`
    3. `avg_dos` : float
        - the average density of states along the edge
    4. `avg_potential` : float
        - the average spectral potential along the edge

```
1  node_attr = dict(graph.nodes(data=True))
2  edge_attr = list(graph.edges(data=True))
3  print('The attributes of the first node\n', node_attr[0], '\n')
4  print('The attributes of the first edge\n', edge_attr[0][-1], '\n')
```

>>>

```
1   The attributes of the first node
2    {'pos': array([-0.20403848, -2.11668106]),
3     'dos': 0.0011466597206890583,
4     'potential': -0.655870258808136}
5
6   The attributes of the first edge
7    {'weight': 1.4176547247784077,
8     'pts': array([[-2.04038482e-01, -2.11668106e+00],
9          [-1.99792382e-01, -2.11243496e+00],
10         ...
11         [ 5.94228396e-01, -1.02967935e+00]]),
```

```
12    'avg_dos': 0.10761458,
13    'avg_potential': -0.5068641}
```

### G.3.4 A GENERIC **MULTI-BAND** CLASS (P2G.CHARPOLYCLASS):

Let us add two parameters $\{a, b\}$ to the aforementioned multi-band example and construct a `p2g.CharPolyClass` object:

```
1  a, b = sp.symbols('a b', real=True)
2
3  cp = p2g.CharPolyClass(
4      "z**2 + a/z**2 + b*E*z - E**4",
5      k=k, z=z, E=E,
6      params={a, b}, # pass parameters as a set
7  )
```

>>>

```
1  Derived Bloch Hamiltonian `h_z` with 4 bands.
```

View a few auto-computed properties

**Characteristic polynomial**:

```
1  cp.ChP
```

$>>> \mathrm{Poly}\left(z^2 + a\frac{1}{z^2} + bzE - E^4, z, \frac{1}{z}, E, domain = \mathbb{Z}[a, b]\right)$

**Bloch Hamiltonian**:

```
1  cp.h_k
```

>>>

$$\begin{bmatrix} 0 & 0 & 0 & (a + e^{4ik})e^{-2ik} \\ 1 & 0 & 0 & be^{ik} \\ 0 & 1 & 0 & 0 \\ 0 & 0 & 1 & 0 \end{bmatrix}$$

```
1  cp.h_z
```

>>>

$$\begin{bmatrix} 0 & 0 & 0 & \frac{a}{z^2} + z^2 \\ 1 & 0 & 0 & bz \\ 0 & 1 & 0 & 0 \\ 0 & 0 & 1 & 0 \end{bmatrix}$$

**The Frobenius companion matrix of `P(E)(z)`**:

```
1  cp.companion_E
```

>>>

$$
\begin{bmatrix}
0 & 0 & 0 & -a \\
1 & 0 & 0 & 0 \\
0 & 1 & 0 & E^4 \\
0 & 0 & 1 & -Eb
\end{bmatrix}
$$

### AN ARRAY OF SPECTRAL FUNCTIONS

To get an array of spectral images or spectral graphs, we first prepare the values of the parameters $\{a,b\}$

```python
a_array = np.linspace(-2, 1, 6)
b_array = np.linspace(-1, 1, 6)
a_grid, b_grid = np.meshgrid(a_array, b_array)
param_dict = {a: a_grid, b: b_grid}
print('a_grid shape:', a_grid.shape,
      '\nb_grid shape:', b_grid.shape)
```

>>>

```
a_grid shape: (6, 6)
b_grid shape: (6, 6)
```

Note that **the value array of the parameters should have the same shape**, which is also **the shape of the output array of spectral images**

```python
phi_arr, dos_arr, binaried_dos_arr, spectral_square = \
    cp.spectral_images(param_dict=param_dict)
print('phi_arr shape:', phi_arr.shape,
    '\ndos_arr shape:', dos_arr.shape,
    '\nbinaried_dos_arr shape:', binaried_dos_arr.shape)
```

>>>

```
phi_arr shape: (6, 6, 1024, 1024)
dos_arr shape: (6, 6, 1024, 1024)
binaried_dos_arr shape: (6, 6, 1024, 1024)
```

```python
from mpl_toolkits.axes_grid1 import ImageGrid

fig = plt.figure(figsize=(13, 13))
grid = ImageGrid(fig, 111, nrows_ncols=(6, 6), axes_pad=0,
                 label_mode='L', share_all=True)

for ax, (i, j) in zip(grid, [(i, j) for i in range(6) for j in
 ↪  range(6)]):
    ax.imshow(phi_arr[i, j], extent=spectral_square[i, j],
     ↪  cmap='terrain')
    ax.set(xlabel='Re(E)', ylabel='Im(E)')
    ax.text(
        0.03, 0.97, f'a = {a_array[i]:.2f}, b = {b_array[j]:.2f}',
        ha='left', va='top', transform=ax.transAxes,
        fontsize=10, color='tab:red',
        bbox=dict(alpha=0.8, facecolor='white')
    )

```

```
17  plt.tight_layout()
18  plt.savefig('./assets/ChP_spectral_potential_grid.png', dpi=72)
19  plt.show()
```

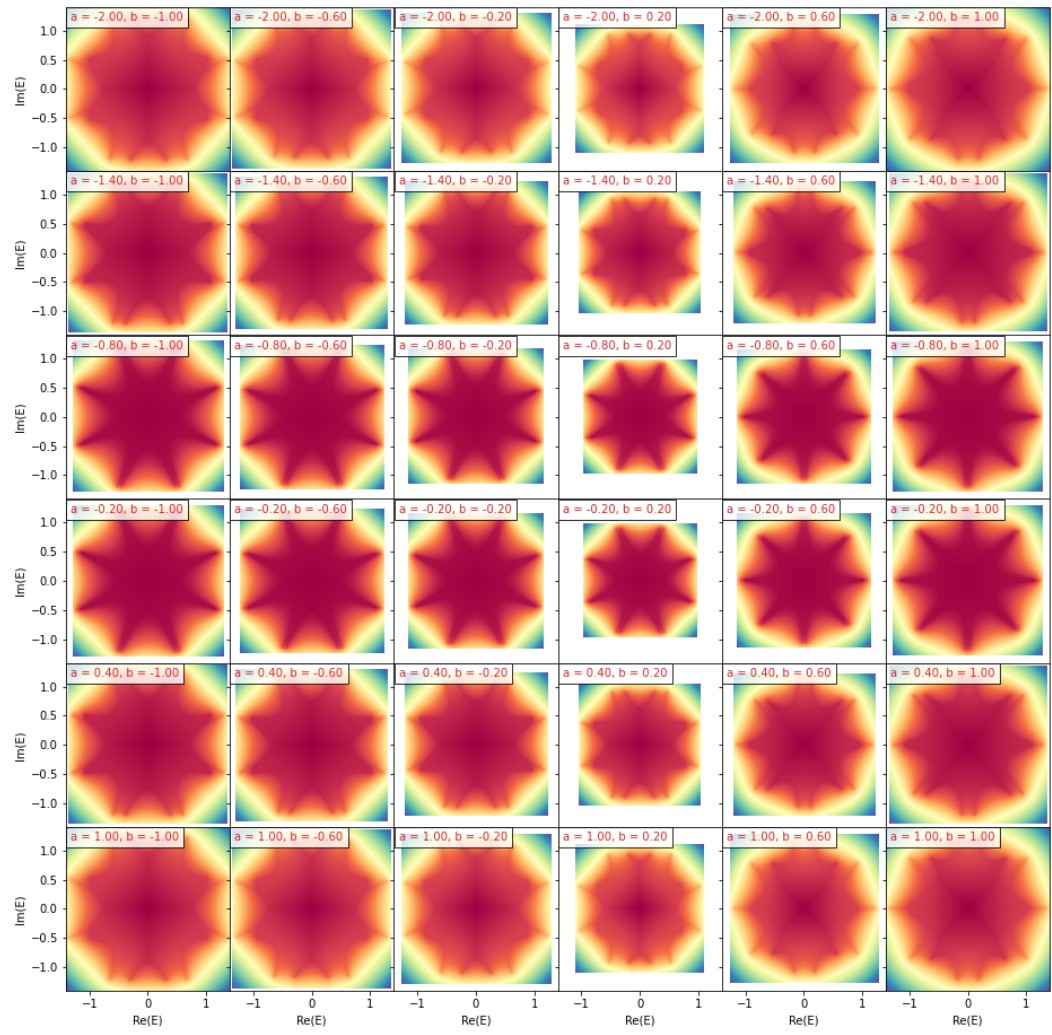

### AN ARRAY OF SPECTRAL GRAPHS

```
1  graph_flat, param_dict_flat = cp.spectral_graph(param_dict=param_dict)
2  print(graph_flat, '\n')
3  print(param_dict_flat)
```

```
1  [<networkx.classes.multigraph.MultiGraph object at 0x000001966DFCD190>,
2  <networkx.classes.multigraph.MultiGraph object at 0x000001966DFCECF0>,
3  ...
4  <networkx.classes.multigraph.MultiGraph object at 0x000001966DFCE750>]
5
6  {a:
```

```
7  array([-2. , -1.4, -0.8, -0.2,  0.4,  1. , -2. , -1.4, -0.8, -0.2,
   ↪   0.4,  1. , -2. , -1.4, -0.8, -0.2,  0.4,  1. , -2. , -1.4, -0.8,
   ↪  -0.2,  0.4,  1. , -2. , -1.4, -0.8, -0.2,  0.4,  1. , -2. , -1.4,
   ↪  -0.8, -0.2,  0.4,  1. ]),
8  b:
9  array([-1. , -1. , -1. , -1. , -1. , -1. , -0.6, -0.6, -0.6, -0.6,
   ↪  -0.6, -0.6, -0.2, -0.2, -0.2, -0.2, -0.2, -0.2,  0.2,  0.2,  0.2,
   ↪   0.2,  0.2,  0.2,  0.6,  0.6,  0.6,  0.6,  0.6,  0.6,  1. ,  1. ,
   ↪   1. ,  1. ,  1. ,  1. ])}
```

The spectral graph is a `networkx.MultiGraph` object, which cannot be directly returned as a multi-dimensional numpy array of `MultiGraph`, except for the case of 1D array. Instead, we return a flattened list of `networkx.MultiGraph` objects, and the accompanying `param_dict_flat` is the dictionary that contains the corresponding flattened parameter values.

It's recommended to pass the values of the parameters as `vectors` (1D arrays) instead of higher dimensional `ND arrays` to avoid the overhead of reshaping the output and the difficulty to retrieve / postprocess the spectral graphs.

## H   LIMITATIONS AND FUTURE WORK

**Component fragmentation.** Our extraction pipeline struggles when the hopping range or band number becomes large (e.g. for pure theoretical interests that fall outside realistic domain), because extremely low densities of states make the graph skeleton fragile, occasionally fragmenting a connected component (as shown in the bottom row in figure A4). We term this phenomenon *component fragmentation* and note that it is an intrinsic limitation of the spectral graph per se (see appendix C.6).

**Better designed, more comprehensive benchmark.** Our contribution centers on the dataset and its generator; the benchmark is intended to catalyze follow-up work. We invite the community to perform comprehensive, large-scale, and carefully designed evaluations.

**Representation gap.** Our reference PyG conversion uses fixed-size, direction-agnostic edge summaries, which can discard full continuous details of multi-edge geometry. Future encoders could operate directly on edge coordinate sequences, e.g. explore spline/Bezier bases, curvature/shape descriptors.

**Multi-edge modeling.** Vanilla attention and pooling are not tailored for heavy edge multiplicity. Multi-edge–aware mechanisms—typed/bundled edges, edge-gated updates, sparsified geometric attention, or dual-graph pooling over edges—may better exploit information carried by parallel curves.

**Temporal modeling (T-HSG-5M).** Our static benchmark does not cover dynamic tasks. `T-HSG-5M` enables early-sequence classification, temporal extrapolation, and change-point detection that leverage continuous geometric evolution along Hamiltonian parameters.

