# OpenReview forum: "HSG-12M: A Large-Scale Benchmark of Spatial Multigraphs from the Energy Spectra of Non-Hermitian Crystals"
_ICLR.cc/2026/Conference — ICLR 2026 Poster_

### Official Review · Reviewer_aSbR · 2025-10-25

**Soundness:** 3
**Presentation:** 3
**Contribution:** 2
**Rating:** 6
**Confidence:** 1

**Summary:**

The paper introduces Poly2Graph, an automated pipeline converting 1-D crystal Hamiltonians into spectral graphs. This enables the creation of HSG-12M, a large-scale dataset of spatial multigraphs, which is a new benchmark for geometry-aware graph learning.

**Strengths:**

The work introduces HSG-12M, a large-scale dataset. It addresses a clear gap by providing the first large-scale benchmark focused on spatial multigraphs , which are underrepresented in graph ML.

The authors release the Poly2Graph pipeline. This open-source tool enables researchers to generate new spectral graph datasets , promoting reproducibility and future study.

**Weaknesses:**

As the reviewer is not an expert in this area, the following points are offered from a more general perspective:

1. The paper’s heavy reliance on domain-specific concepts from non-Hermitian physics and algebraic geometry (e.g., Ronkin functions)  might make the data generation process inaccessible to the broader graph ML community. The physical motivation for the benchmark task (inverse design) feels highly specialized.

2. The benchmark evaluation focuses on standard GNNs (GCN, GAT, GIN)  not explicitly designed for spatial multigraphs. The paper identifies this as a new challenge but stops short of benchmarking against more suitable, geometry-aware architectures, weakening the analysis of the dataset's specific challenges.

3. A key feature is the rich, geometric nature of the multi-edges. However, the reference benchmark discards this by collapsing edge geometry into fixed-size summary features (e.g., length, midpoint). This featurization, while practical, may underutilize the dataset's novel geometric complexity.

4. The authors acknowledge a limitation in the Poly2Graph pipeline called "component fragmentation", where extremely low densities of states can cause graphs to be spuriously disconnected. This raises potential concerns about the topological fidelity of the generated graphs in complex cases.

**Questions:**

Please see the weaknesses.

---

> ### Author Response · Authors · 2025-11-25
> **Response to Reviewer aSbR (Part 1/3)**
>
> We thank the reviewer for the thoughtful assessment and for recognizing the value of **HSG-12M** as the *"first large-scale benchmark focused on spatial multigraphs"* and **Poly2Graph** for *"promoting reproducibility and future study"*. Below, we address each of the weaknesses raised.
>
> ---
>
> ## 1. Accessibility and Physics Motivation (W1)
>
> > *The paper’s heavy reliance on domain-specific concepts from non-Hermitian physics and algebraic geometry (e.g., Ronkin functions) might make the data generation process inaccessible to the broader graph ML community. The physical motivation for the benchmark task (inverse design) feels highly specialized.*
>
> ### 1.1 Making the physics accessible via Appendix B
>
> We agree that non-Hermitian physics concepts (e.g., Ronkin functions) are not common knowledge in graph ML. Precisely for this reason, we devoted substantial effort to **Appendix B**, written as a **self-contained tutorial** that assumes only basic solid-state background. It explains:
>
> 1. How spectral graphs emerge from 1D tight-binding Hamiltonians;
> 2. The electrostatic analogy that maps spectra to "ridges" of a potential landscape;
> 3. How to interpret the data without needing to master the full non-Bloch band theory.
>
> Moreover, **using the dataset does not require physics expertise**. For an ML practitioner, the task is a standard *graph-level classification* problem: input $\in$ spatial multigraphs, output $\in$ 1,401 classes. All domain-specific machinery is encapsulated inside the Poly2Graph generator; the learning task itself is conventional.
>
> ### 1.2 Inverse design as a general scientific challenge
>
> While the concrete application here is in condensed-matter physics, the core problem—**inverse design**—is a common and high-impact challenge across many scientific fields (Appendix E.5).
>
> **Poly2Graph** addresses the ***forward*** problem: given a crystal Hamiltonian (*material structure*), compute its spectral graph (*property*).
> The classification task on **HSG-12M** is designed as a surrogate for the ***inverse*** problem:
>
> > ***Given a desired / observed spectral graph (desired property), what class of microscopic Hamiltonians (material design) could realize it?***
>
> Analytically inverting this map is intractable: the forward process is nonlinear and many-to-one (as is clear from the Poly2Graph pipeline), so **no simple closed-form inverse exists**. This is precisely where a GNN surrogate is useful.
>
> We therefore cast the task as supervised *categorical retrieval* from spectral graphs to characteristic-polynomial (ChP) classes. A trained model that, given a spectral graph, returns a small set of candidate Hamiltonian families directly supports scalable inverse-design workflows: physicists can start from a target spectral fingerprint and quickly narrow down plausible microscopic structures for further analytic or experimental refinement.
>
> For the physics / materials community, this is a practical **inverse-design tool**, analogous to protein design: we seek the "amino-acid sequence" (1D crystal design / ChP class) that realizes a desired "spatial fold" (spectral graph). In this sense, the benchmark is not artificial but a realistic, physics-motivated inverse-design task for which graph learning is a natural approach.
>
> Finally, in Sec. 5 and App. F of the paper we show that the same spectral-graph construction applies to ***any*** vector, matrix, or polynomial, providing a general "algebra-as-graph" representation. This broadens the conceptual scope well beyond non-Hermitian physics.

---

> ### Author Response · Authors · 2025-11-25
> **Response to Reviewer aSbR (Part 2/3)**
>
> ## 2. Benchmarks vs. Geometry-Aware Models (W2)
>
> > *The benchmark evaluation focuses on standard GNNs (GCN, GAT, GIN) not explicitly designed for spatial multigraphs. The paper identifies this as a new challenge but stops short of benchmarking against more suitable, geometry-aware architectures, weakening the analysis of the dataset's specific challenges.*
>
> We agree that geometry-aware modeling is essential. However, no existing GNN architectures are explicitly designed for **spatial multigraphs**, and in order to provide a comprehensive benchmark, we did carefully consider several geometry-aware models that were designed for **spatial (simple) graphs**.
>
> 1. **Absence of dedicated "spatial multigraph" GNNs.**
>    To the best of our knowledge, at the time of writing there is **no existing GNN architecture explicitly designed for spatial *multigraphs* classification**.\
>    A key technical barrier accounting for this absence is that **inhomogeneous multi-edge polylines** are not natively supported in common frameworks (`PyG`, `DGL`, etc.): each edge in HSG-12M carries a *variable-length sequence* of 2D coordinates. Designing architectures that operate on these sequences directly is an interesting direction but beyond our current scope.
>
> 2. **Included geometry-aware baselines.**
>    In addition to "standard" GNNs, we already include models originally proposed for spatial graphs and adapt them to HSG-12M:
>
>    * **CGCNN** (Crystal Graph Convolutional Neural Networks) explicitly incorporates spatial features from nodes and edges in its message passing.
>    * **MF** (Molecular Fingerprint) uses node spatial features to produce graph representation.
>
>    Empirically, we see edge-aware spatial models significantly outperform the edge-agnostic baselines (e.g. CGCNN $\gg$ MF, GINE $\gg$ GIN, in Table 2), underscoring that **explicit edge geometry is crucial** for spatial multigraphs.
>
> 3. **Considered but excluded geometric models.**
>    We also considered several other geometry-aware architectures but ultimately excluded them due to fundamental design misalignment with our setting:
>
>    * **SchNet [1]:** Designed for 3D atomistic energy/force prediction using continuous radial filters and dense distance graphs. Its inductive bias and full-pairwise construction do not naturally fit 2D planar spectral multigraph classification.
>    * **SplineCNN [2]:** Targeted at a small number of large geometric meshes with computationally heavy B-spline kernels on pseudo-coordinates, which is impractical for millions of tiny planar graphs.
>    * **DimeNet++ [3]:** Built around 3D distance–angle triplets and spherical Bessel/harmonic bases for molecular energies. This is again tailored to 3D node-centric point clouds, not suited for spatial multiedges.
>
> We view this work as a **starting point and testbed** for future spatial-multigraph GNNs, and we hope our baselines, together with these design considerations, will help guide follow-up work.
>
> ---
> ### *References*
> 1. Schütt, K. T., Sauceda, H. E., Kindermans, P.-J., Tkatchenko, A. & Müller, K.-R. SchNet - a deep learning architecture for molecules and materials. The Journal of Chemical Physics 148, 241722 (2018).
> 2. Fey, M., Lenssen, J. E., Weichert, F. & Muller, H. SplineCNN: Fast Geometric Deep Learning with Continuous B-Spline Kernels. in 2018 IEEE/CVF Conference on Computer Vision and Pattern Recognition 869–877 (IEEE, Salt Lake City, UT, 2018). doi:10.1109/CVPR.2018.00097.
> 3. Gasteiger, J., Giri, S., Margraf, J. T. & Günnemann, S. Fast and Uncertainty-Aware Directional Message Passing for Non-Equilibrium Molecules. Preprint at https://doi.org/10.48550/arXiv.2011.14115 (2022).

---

> ### Author Response · Authors · 2025-11-25
> **Response to Reviewer aSbR (Part 3/3)**
>
> ## 3. Edge Featurization vs. Geometric Complexity (W3)
>
> > *A key feature is the rich, geometric nature of the multi-edges. However, the reference benchmark discards this by collapsing edge geometry into fixed-size summary features (e.g., length, midpoint). This featurization, while practical, may underutilize the dataset's novel geometric complexity.*
>
> We fully agree. Our reference featurization (edge length, straight-line distance, midpoint coordinates, average potential, average DOS) is a **conservative baseline**, chosen to ensure **compatibility with standard GNN libraries, which expect fixed-size edge attributes**.
>
> However, importantly, **no information is lost in the dataset itself**. HSG-12M stores the **full polyline coordinate sequence** for every edge. In **Appendix H**, we explicitly discuss this "representation gap" and frame it as an opportunity for future work—e.g., spline or Bézier bases, curvature/shape descriptors, or sequence models (RNNs / Transformers) over edge paths—to more fully exploit the geometric richness of spatial multigraphs.
>
> ---
>
> ## 4. Component Fragmentation and Topological Fidelity (W4)
>
> > *The authors acknowledge a limitation in the Poly2Graph pipeline called "component fragmentation", where extremely low densities of states can cause graphs to be spuriously disconnected. This raises potential concerns about the topological fidelity of the generated graphs in complex cases.*
>
> We appreciate this concern about "component fragmentation", i.e., spurious disconnections when the Density of States is extremely low near certain junctions. We address it in two ways: by clarifying the **realistic operating regime** of HSG-12M and by providing a **quantitative robustness test**.
>
> ### 4.1 Quality assurance in HSG-12M (realistic regime)
>
> In Appendix H (L2507-2508) we note that: "Our extraction pipeline struggles when the hopping range or band number becomes large (e.g. **for pure theoretical interests that fall outside realistic domain**)"
>
> HSG-12M is *not* constructed in this *extreme* regime. It is sampled around **realistic Hamiltonian families** with limited hopping range and band number. The "component fragmentation" **caveat is included primarily for users who may apply Poly2Graph to highly contrived, far-from-realistic models**.
>
> During development, we **manually validated** Poly2Graph on a few hundred characteristic polynomials by comparing extracted spectral graphs to spectra obtained from exact diagonalization. In these tests, the extracted graphs visually match the energy spectra, including edge connectivity and overall geometry.
>
> ### 4.2 Quantitative robustness test
>
> To further quantify topological stability, we performed a stress test on the extraction hyperparameter $\tau$ that controls node merging and thus fragmentation mitigation:
>
> * **Setup.** We sampled 20 characteristic polynomials from HSG-12M. For each, we generated the spectral graph at the default threshold $\tau$ and again at 20 perturbed thresholds $\bar{\tau} \in [0.5\tau, 2\tau]$.
> * **Metric.** For each polynomial and perturbed threshold, we tested **multigraph isomorphism** between the perturbed graph and its reference graph at $\tau$.
>
> This yields 20 (polynomials) × 20 (thresholds) = 400 perturbed graphs.
>
> **Results:**
> | Default threshold | Perturbed range | Isomorphic matches |
> | :--- | :--- | :--- |
> | $\tau$ | $0.5\tau \sim 2\tau$ | **391 / 400 (97.75%)** |
>
> The 97.75% isomorphism rate indicates that, within the realistic regime of HSG-12M, the extracted graph topology is **highly stable** with respect to the choice of this extraction parameter.

---

> > ### Comment · Reviewer_aSbR · 2025-11-26
> >
> > Thanks for the author's detailed response. I believe that the response has addressed my concerns, and in light of the author's responses to other reviewers, I decide to maintain my positive score.

---

> > > ### Author Response · Authors · 2025-11-27
> > >
> > > Thank you for your follow-up and for taking the time to reconsider our work. We are glad that our clarifications addressed your concerns, and we sincerely appreciate your positive assessment and increased confidence in the paper. If there is any further information or clarification that could help strengthen your assessment of the work, we would be very happy to provide it.

---

### Official Review · Reviewer_JuyR · 2025-10-31

**Soundness:** 3
**Presentation:** 3
**Contribution:** 2
**Rating:** 4
**Confidence:** 3

**Summary:**

The paper presents a computational framework (Poly2Graph) and large-scale dataset that map non-Hermitian Hamiltonians of one-dimensional crystals into spectral graphs, i.e., geometric multigraphs embedded in the complex energy plane. The framework provides efficient construction of spectral graphs from Hamiltonians, enabling systematic dataset generation across varying scales. The largest constructed dataset, HSG-12M, contains 11.6 million graphs spanning 1,401 distinct Hamiltonian (characteristic polynomial) classes. The authors also benchmark multiple graph neural network (GNN) architectures on a graph-level classification task: predicting the underlying Hamiltonian class from the corresponding spectral graph.

**Strengths:**

- Non-Hermitian physics has emerged as an important area in condensed matter and atomic physics. This work establishes a physics-grounded benchmark for evaluating and developing machine learning models on spatial multigraphs derived from physical systems.
- The dataset is impressive in scale ($\approx$12 million graphs), and the automated spectral graph extraction pipeline is technically solid, with potential benefits for the broader physics community.

**Weaknesses:**

The paper does not clearly explain whether the proposed task, predicting the Hamiltonian class from its spectral graph, corresponds to a physically meaningful scenario. Specially, my question is two-fold:
- This formulation implicitly assumes situations where the Hamiltonian is unknown but the spectral graph itself can somehow be measured, possibly through experiments. A key question is whether the spectral graph, as defined in this work, represents a physically measurable quantity that can be constructed from experimental data.
- Also, the authors may elaborate on why predicting the Hamiltonian class represents a meaningful task in physics.

I would be glad to raise my score if the authors can adequately address these concerns.

**Questions:**

See Weaknesses.

---

> ### Author Response · Authors · 2025-11-25
> **Response to Reviewer JuyR (Part 1/2)**
>
> We thank the reviewer for highlighting the "impressive scale" of our dataset and the "technically solid" pipeline. We appreciate the opportunity to clarify the physical significance of our work. Below, we address your twofold question regarding (1) the physical measurability of spectral graphs and (2) the scientific value of the classification task.
>
> ---
>
> ## 1. Physical Meaning and Measurability of the Spectral Graph
>
> > (W1) ... A key question is whether the spectral graph, as defined in this work, represents a physically measurable quantity that can be constructed from experimental data.
>
> ### 1.1 The Spectral Graph is Effectively the Energy Spectrum
>
> The *Hamiltonian spectral graph* is not an abstract mathematical artifact; it is **physically defined** as *the set of eigenvalues (energies)* of a 1D crystal (under open-boundary conditions in the thermodynamic limit), whose appearance traces out a *spatial planar multigraph* in the complex energy plane (see **Figure A3**).
>
> Recent physics works [1–3] have established this object as a *rigorous, topology-bearing structure* in non-Hermitian band theory. However, these studies are limited to a small number of toy models and rely on manual inspection.
>
> ### 1.2 Experimental Measurability
>
> By definition, the spectral graph is **the set of eigenvalues** (eigen-energies, $E$); in terms of density of states ($\rho(E)$)—the number of eigenstates/eigenvalues per unit area in the complex energy plane—it corresponds to **the 1D loci where $\rho(E) > 0$ (Figure 2c)**.
>
> Experimentally, both **eigen-energies** and the **density of states** are standard observables, particularly accessible in **metamaterial platforms** such as electrical circuits, photonic lattices, and acoustic crystals (*Line 277*).
>
> **In a recent experimental work [4] (on ***Nature Comm. Phys.***), the authors explicitly highlight the measurability of spectral graphs:**
>
> > "... static mechanical systems can serve as a valuable platform for exploring highly susceptible non-Hermitian physics, including non-Abelian braiding of non-Bloch bands, ***graph topology of non-Hermitian spectra***, and enhanced nonnormalities."
>
> While direct full‑plane reconstruction of the spectral graph has not yet been demonstrated, **[4] explicitly points out that *spectral graph* is a relevant measurable target**.
>
> This situation is closely analogous to *earlier stages* of modern topological-materials research [5,6], where topological invariants were first characterized theoretically [7,9] and only later measured or inferred experimentally [8,10]. Our paper plays a similar role: it systematizes a *theoretically well-defined, experimentally motivated* spectral object and provides tools and data for exploring it at scale.

---

> ### Author Response · Authors · 2025-11-25
> **Response to Reviewer JuyR (Part 2/2)**
>
> ## 2. Scientific Value of the Task: Inverse Materials Design
>
> > (W2) The authors may elaborate on why predicting the Hamiltonian class represents a meaningful task in physics.
>
> In the manuscript, we have detailed the physics motivation in **Appendix E.5**, and expand further here.
>
> ### 2.1 The Inverse Problem
>
> **Poly2Graph** addresses the ***forward*** problem: given a crystal Hamiltonian (*material structure*), compute its spectral graph (*property*):
>
> $$\text{Hamiltonian } H \longrightarrow \text{spectral graph } \mathcal{G}(H)$$
>
> The classification task on **HSG-12M** is designed as a surrogate for the ***inverse*** problem:
>
> > ***Given a desired / observed spectral graph (desired property), what class of microscopic Hamiltonians (material design) could realize it?***
>
> This inverse-design problem is *unresolved*, *highly non-trivial*, and of central interest in condensed-matter physics and materials science. We cast it as supervised *categorical retrieval* from spectral graphs to discrete characteristic-polynomial (ChP) classes — the "class" is defined by the polynomial support up to parity symmetry, which **dictates which atomic sites interact with which** (e.g., nearest-neighbor vs. next-nearest-neighbor electron transitions).
>
> $$ \text{GNN surrogate } f_\theta: \quad \text{spectral graph } \mathcal{G} \longrightarrow \text{ChP class } C_P$$
>
> Analytically, the inverse mapping from spectral graph to Hamiltonian is intractable: the forward process is nonlinear and irreversible (as is evident from the `Poly2Graph` algorithm, Appendix C), so **there is no simple closed-form inverse. This is exactly where a GNN surrogate is necessary**.
>
> To the ML community this *may look like "just another supervised classification problem,"* but to the physics/materials community it is a powerful **inverse-design tool**: it dramatically narrows the search space over microscopic structures consistent with a target spectral fingerprint. The analogy is to protein design: we want to know which amino-acid sequence (1D crystal design / ChP class) yields a desired protein fold (spectral graph).
>
> ### 2.2 Practical Workflow
>
> From a practical standpoint, a well‑trained model $f_\theta(\mathcal{G})$ that maps spectral graphs to ChP classes enables the following workflow:
>
> 1. **Target choice or measurement.** Start from a desired spectral graph or a graph reconstructed from experiment.
>
> 2. **Model‑based retrieval.** Use the GNN surrogate $f_\theta(\mathcal{G})$ to obtain a ranked list of candidate ChP classes. Thanks to the high Top‑10 accuracy we observe on HSG‑12M (e.g., 95% for GraphSAGE; Table 2), the ground‑truth class is almost always within a small candidate set.
>
> 3. **Physics‑based refinement.** Within these few candidate families, one can then perform fine-grained parameter tuning within that family to match specific requirements. This bridges the gap between abstract spectral topology and concrete material realization.
>
> This is precisely how inverse design is carried out in many other domains: ML is used to *narrow the search space to a small, plausible set of structural families*, after which domain experts refine within that set.  In this sense, the benchmark is not artificial; it is a realistic, physics‑motivated inverse‑design task, for which graph learning is likely the only practical approach.
>
> ---
>
> ## *References*
> 1. Tai, T. & Lee, C. H. Zoology of non-Hermitian spectra and their graph topology. *Phys. Rev. B* 107, L220301 (2023).
> 2. Xiong, Y. & Hu, H. Graph morphology of non-Hermitian bands. *Phys. Rev. B* 109, L100301 (2024).
> 3. Wang, H.-Y., Song, F. & Wang, Z. Amoeba Formulation of Non-Bloch Band Theory in Arbitrary Dimensions. *Phys. Rev. X* 14, 021011 (2024).
> 4. Wang, A. & Chen, C. Q. Observing non-Bloch braids and phase transitions by precise manipulation of the non-Hermitian boundary and size. *Commun Phys* 8, 294 (2025).
> 5. Bradlyn, B. et al. Topological quantum chemistry. *Nature* 547, 298–305 (2017).
> 6. Vergniory, M. G. et al. A complete catalogue of high-quality topological materials. *Nature* 566, 480–485 (2019).
> 7. Hu, H. & Zhao, E. Knots and Non-Hermitian Bloch Bands. *Phys. Rev. Lett.* 126, 010401 (2021).
> 8. Wang, K., Dutt, A., Wojcik, C. C. & Fan, S. Topological complex-energy braiding of non-Hermitian bands. *Nature* 598, 59–64 (2021).
> 9. Bernevig, B. A., Hughes, T. L. & Zhang, S.-C. Quantum Spin Hall Effect and Topological Phase Transition in HgTe Quantum Wells. *Science* 314, 1757–1761 (2006).
> 10. König, M. et al. Quantum Spin Hall Insulator State in HgTe Quantum Wells. *Science* 318, 766–770 (2007).

---

> > ### Comment · Reviewer_JuyR · 2025-11-26
> > **Response to Authors**
> >
> > Thank you for your rebuttal. It is helpful to know the measurability of the Spectral Graph in practice. The idea of inverse design of Hamiltonian (classes) by the spectral graphs also sounds valuable. The response has addressed my concerns. I have raised my score and now advocate for acceptance.

---

> > > ### Author Response · Authors · 2025-11-26
> > >
> > > Thank you for your prompt follow-up and for reconsidering your assessment of our work. We are glad that our rebuttal addressed your concerns. If there is anything else we can clarify or any additional details that would help strengthen the paper, we would be very happy to provide them.

---

### Official Review · Reviewer_7GxK · 2025-11-01

**Soundness:** 3
**Presentation:** 3
**Contribution:** 3
**Rating:** 6
**Confidence:** 2

**Summary:**

This submission introduces Poly2Graph, a high-throughput package that converts 1D crystal Hamiltonians into Hamiltonian Spectral Graphs (HSGs). Using Poly2Graph, the authors construct and release HSG-12M which is a large static spatial multigraph dataset with 11.6M static and 5.1M dynamic Hamiltonian spectral graphs across 1,401 classes; For the HSGs, each graph class corresponds to a Hamiltonian family (hopping pattern), the multi-edge spatial geometry is essential and cannot be simplified without loss, and  a GNN surrogate may enable inverse design from desired spectral graphs to candidate material structures (e.g., acoustic metamaterials, circuits, photonic crystals).

**Strengths:**

1. Clear problem framing & significance. Casting Hamiltonian systems as spatial multigraphs is compelling and bridges non-Hermitian quantum physics with graph representation learning.

2. Scale & engineering contribution. The scale (11.6M static; 5.1M dynamic; 1,401 classes) of the graph data is notable.

3. Breadth of tasks. Static graph classification plus temporal, graph-level tasks opens avenues beyond typical node/edge prediction settings in temporal graphs.

4. Inverse-design angle. The connection between hopping patterns ↔ spectral graphs ↔ structure motivates differentiable surrogates and downstream inverse design—a high-impact direction.

5. Writing & organization. The paper reads smoothly; motivation, pipeline, and applications are explained with good intuition.

**Weaknesses:**

1. What’s fundamentally new on the modeling side?
Beyond scale and the presence of multi-edges + spatial coordinates, it’s not yet clear which modeling challenges are truly novel relative to existing large-scale domains (molecules, traffic, social). For example: Do standard GNNs (e.g., message passing with geometric encoders) already handle these graphs well? Which failure modes emerge uniquely from spatial multigraphs that are not captured in simple GNNs?

**Questions:**

N/A

---

> ### Author Response · Authors · 2025-11-25
> **Response to Reviewer 7GxK**
>
> We thank Reviewer 2 for the positive feedback and constructive comments. Below we address the main question about what is new on the modeling side and whether existing GNNs already handle our graphs.
>
> ---
>
> ## 1. Do standard (geometry‑aware) GNNs already handle HSG‑12M well?
>
> This is exactly what we set out to test in our benchmark. We evaluate standard edge‑agnostic models (GCN, GIN, GraphSAGE), edge‑aware models (GAT, GATv2, GINE), and **geometry‑aware models**:
> - **CGCNN** (Crystal Graph Convolutional Neural Networks) supports explicitly encoding spatial features from nodes and edges during message passing.
> - **MF** (Molecular Fingerprint) utilizes node spatial features only. In our setting it can exploit node coordinates but not multi‑edge geometry.
>
> From Table 2 we observe:
> - **Edge features matter.** Edge‑aware models such as GINE and CGCNN consistently outperform node‑only baselines (e.g., CGCNN $\gg$ MF, GINE $\gg$ GIN), indicating that explicitly modeling edge geometry is crucial for spatial multigraphs.
> - **Modest performance overall.** Even the best models reach only moderate Top‑1 accuracies (≈30–60% on the full dataset), so the task remains challenging and is not trivially solved by standard architectures.
>
> Thus current GNNs, including geometry‑aware ones, **do not** fully handle HSG‑12M out of the box.
>
> ---
>
> ## 2. Failure modes and challenges specific to spatial multigraphs
>
> Relative to molecules, traffic, or social networks, HSG‑12M introduces several modeling issues that are either absent or much weaker in existing benchmarks:
>
> - **Rich multi‑edge geometry.** Standard message passing aggregates these parallel edges with simple permutation‑invariant sums/means, discarding ordering and shape information, so physically different multi‑edge configurations can become indistinguishable.
> - **Physically meaningful self‑loops.** In our graphs, self‑loops correspond to non‑trivial complex energies, conflicting with the usual trick that adding self-loops acts as residual connections. Most GNN implementations either ignore explicit self‑loops or add them uniformly, and cannot currently treat them as information‑bearing edges.
> - **Variable‑length edge paths.** Each edge is stored as a *variable‑length sequence of 2D coordinates*. Popular graph libraries (PyG, DGL, etc.) assume fixed‑size edge features, so one **cannot directly feed these polylines into existing frameworks**. Our reference benchmark therefore compresses them into fixed‑size summaries (length, midpoint, average DOS/potential), which is practical but leaves substantial geometric signal for future models to exploit.
>
> Structurally, the large‑scale domains mentioned by the reviewer are also different:
>
> - Molecular benchmarks do not treat bonds as spatial *paths*; edges are abstract connections with chemical attributes rather than embedded curves.
> - Existing traffic and social‑network benchmarks are largely node‑/edge‑level prediction tasks, whereas HSG‑12M is a **graph‑level** benchmark.
>
> ---
>
> ## 3. Scope of our modeling contribution
>
> We agree that our main contribution is the **dataset and generator** rather than new modeling insights. We are careful not to over‑claim on the benchmarking side (L2512-2514): "Our contribution centers on the dataset and its generator; the benchmark is intended to catalyze follow-up work. We invite the community to perform comprehensive, large-scale, and carefully designed evaluations."
>
> With limited compute and as non‑front‑line GNN designers, we focused on providing a fair, reproducible baseline with carefully chosen models. We hope HSG‑12M will motivate dedicated spatial‑multigraph architectures that directly tackle the challenges outlined above.

---

### Official Review · Reviewer_K5Rg · 2025-11-01

**Soundness:** 3
**Presentation:** 3
**Contribution:** 2
**Rating:** 2
**Confidence:** 3

**Summary:**

This paper introduces Poly2Graph, an open-source pipeline that converts one-dimensional non-Hermitian crystal Hamiltonians into Hamiltonian spectral graphs (HSGs): spatial multigraphs representing the geometry of complex energy spectra. The authors build HSG-12M, a dataset of 11.6 M static and 5.1 M dynamic multigraphs spanning 1,401 characteristic-polynomial classes. Each graph encodes spectral topology and geometry via node coordinates and multi-edge trajectories on the complex-energy plane. The dataset is positioned as the first large-scale benchmark for spatial multigraph learning, and as a bridge between algebraic physics data and graph machine learning. Benchmarks with popular GNNs (GCN, GAT, GINE, GraphSAGE, etc.) show that edge-aware models outperform edge-agnostic ones, while overall Top-1 accuracies remain moderate ( 30–60 %) and Top-10 accuracies high (95 %).

**Strengths:**

1. Impressive scale and engineering: Poly2Graph automates a previously manual physics workflow, generating > 10 M graphs and compressing 177 TB of raw spectra into 256 GB.
2. First large-scale multigraph dataset
3. Novel cross-disciplinary framing: Establishes a link between non-Hermitian band theory and graph representation learning, potentially inspiring new geometry-aware GNNs.
4. Solid baseline benchmarking: Eight GNNs are compared with consistent training budgets. Results reveal real performance gaps between edge-aware and edge-agnostic models.

**Weaknesses:**

1. Limited justification of scientific or ML value: The paper convincingly shows that the data can be generated, but not why learning from these graphs is necessary or insightful. The benchmark task, namely classifying Hamiltonian families, appears artificial, with no demonstrated physical or methodological payoff.
2. Unclear advantage of the graph representation: The authors do not compare to simpler baselines such as CNNs on spectral images, MLPs on polynomial coefficients, or models trained directly on spectral arrays. It remains unproven that representing spectra as graphs yields better or different information.
3. Representation loss and stability: The extraction from spectra to graph skeletons may discard quantitative details and can fragment edges (acknowledged in Appendix H). No analysis quantifies how much information or robustness is lost.
4. Synthetic and self-contained: All data are algorithmically generated. No connection is made to experimental measurements or to existing real-world multigraph domains.
5. Benchmark insight is shallow: Standard GNNs achieve modest accuracies, but this mainly reflects task complexity and training budget, not necessarily new modeling challenges.
6. Incremental as a benchmark contribution: The work is an engineering milestone rather than a conceptual one. Its usefulness for advancing ML is not clear.

**Questions:**

1. Why is a graph representation preferable to treating the spectra as 2-D arrays or polynomial coefficients?
2. Can you show any downstream task where learning on graphs leads to qualitatively different or improved results than learning on images?
3. How robust are the extracted graphs to numerical perturbations or thresholding choices?
4. Would CNNs or transformers on spectral images achieve comparable or better accuracy?
5. Are there physical or real-data applications planned where HSG-12M would provide measurable benefit?

---

> ### Author Response · Authors · 2025-11-25
> **Response to Reviewer K5Rg (Part 1/6)**
>
> We thank the reviewer for recognizing the "impressive scale and engineering" of `Poly2Graph` and `HSG-12M`. Below we clarify (i) why spectral graphs are bona‑fide physics objects with concrete scientific value, (ii) why our classification task is not artificial but directly linked to inverse materials design, (iii) why a graph representation is preferable to images or coefficients, and (iv) how new experiments (image baselines and robustness tests) address the concerns about representation choice and stability.
>
> ---
>
> ## 1. Scientific Value & Experimental Relevance (W1, W4, Q5)
>
> > (W1) Limited justification of scientific or ML value: ... why learning from these graphs is necessary or insightful. The benchmark task, namely classifying Hamiltonian families, appears artificial, with no demonstrated physical or methodological payoff.\
> > (W4) All data are algorithmically generated. No connection is made to experimental measurements or to existing real-world multigraph domains.\
> > (Q5) Are there physical or real-data applications planned where HSG-12M would provide measurable benefit?
>
> ### 1.1 Spectral graphs are bona‑fide physics objects
>
> HSG‑12M is not just some "data we can generate" or a "convenient graph representation": it is built around a **spectral graph** construction that recent physics works [1–3] have established as a *rigorous, topology‑bearing object* in non‑Hermitian band theory. These works, published in respected journals, show that the graph topology and geometry of complex spectra encode non‑Hermitian band topology. However, they are restricted to toy models and manual inspection.
>
> Our work makes this object accessible *at scale*, providing *the first end‑to‑end, high‑throughput method* to study spectral graphs across all *physically realistic* Hamiltonian families and to perform systematic analyses. This alone provides a "*measurable benefit*" (Q5) for physicists: they can now explore the "zoology" of non‑Hermitian spectral graphs algorithmically rather than manually.
>
> In the manuscript we explicitly highlight that **the spectral graph is itself the scientific object of interest**:
>
> - (L915-817) "These structures represent an uncharted band topology, embedding hidden symmetries and graph topological transitions that lie beyond standard homotopy-based frameworks. In effect, *a new class of topological invariants* appears—those tied to the global geometry of the eigenvalue loci."
> - (L45-50) "These *spectral graphs* serve as fingerprints with far more intricate structures than conventional topological signatures for electronic behavior (e.g., $\mathbb{Z}/\mathbb{Z}_2$ invariants, Chern number)"
>
> **Poly2Graph** and **HSG‑12M** are therefore not just engineering artifacts; they operationalize a physically motivated topological object at scale, in a regime where ML methods become useful and experimentally relevant.
>
> ### 1.2 Scientific value of the classification task: rational material design
>
> We have detailed the motivation in **Appendix E.5**, and expand further here.
>
> **Poly2Graph** addresses the ***forward*** problem: given a crystal Hamiltonian  (*material structure*), compute its spectral graph (*property*).
>
> The classification task on **HSG‑12M** is designed as a surrogate for the ***inverse*** problem:
>
> > ***Given a desired / observed spectral graph (desired property), what class of microscopic Hamiltonians (material design) could realize it?***
>
> This inverse‑design problem is *highly non-trivial* and *of central interest* in condensed‑matter physics and materials science. We cast it as supervised *categorical retrieval* from spectral graphs to discrete characteristic‑polynomial (ChP) classes. The benchmark task is precisely what is needed to scale & accelerate such inverse‑design workflows: a model that, given a desired/observed spectral graph, returns a small set of candidate Hamiltonian families for further analytic or experimental refinement.
>
> Analytically, the inverse mapping from spectral graph to Hamiltonian is intractable: the forward process is nonlinear and irreversible (as is evident from the `Poly2Graph` algorithm, Appendix C), so **there is no simple closed-form inverse. This is exactly where a GNN surrogate is necessary**.
>
> To the ML community this *may look like "just another supervised classification problem,"* but to the physics/materials community it is a powerful **inverse‑design tool**: it dramatically narrows the search space over microscopic structures consistent with a target spectral fingerprint—a "*highly tangible payoff*" (W1).
>
> The analogy is to protein design: we want to know which amino‑acid sequence (1D crystal design / ChP class) yields a desired protein fold (spectral graph). In this sense, the benchmark is not artificial; it is a realistic, physics‑motivated inverse‑design task, for which graph learning is likely the only practical approach.

---

> ### Author Response · Authors · 2025-11-25
> **Response to Reviewer K5Rg (Part 2/6)**
>
> ### 1.3 Experimental measurability and connection to real data
>
> - **Experimental measurability.**
>   The spectral graph is the set of all eigenvalues of a crystal (under open-boundary condition, in the thermodynamic limit), embedded in the complex‑energy plane. Eigenvalues ($E$) and the **density of states** ($\rho(E)$) are standard experimental observables, and are particularly accessible in meta‑material platforms such as electrical circuits, photonic lattices, and acoustic crystals.
>
>   **In a recent experimental work [4] (on ***Nature Comm. Phys.***), the authors explicitly highlight the measurability of spectral graphs:**
>   > "... static mechanical systems can serve as a valuable platform for exploring highly susceptible non-Hermitian physics, including non-Abelian braiding of non-Bloch bands, ***graph topology of non-Hermitian spectra***, and enhanced nonnormalities."
>
>   Although [4] does not yet show direct measurements of spectral graphs, **it explicitly points out that ***spectral graph*** is a relevant measurable target.**
>
> - **Theory before experiment is the norm.**
>   In modern condensed‑matter physics, new topological fingerprints are typically developed theoretically long before direct experimental realization. For example:
>
>   - Non‑Hermitian complex‑energy braiding was theoretically predicted in 2021 [7] and experimentally verified later in [8].
>   - Topological insulators were theoretically predicted in 2006 [9] and experimentally verified in 2007 [10].
>
>   Our work similarly advances the **theoretical modeling and organization** of spectral graphs, providing a "periodic table" of these exotic shapes. Once experimental platforms can directly reconstruct spectral graphs, **such a taxonomy is expected to play an important role**, just as the classification of band topologies did in the topological‑materials program [5,6].
>
> ### 1.4 Universal applications of spectral graphs beyond non‑Hermitian physics
>
> Beyond non‑Hermitian band theory, we show in Section 5 and Appendix F that the spectral‑graph idea provides a **universal topological fingerprint** for ***any*** vector, matrix, or univariate/bivariate polynomial. In the manuscript we state:
>
> - (L263-265) "Poly2Graph establishes ... mechanism for translating linear operators into ... graphs. ... The same principle extends to *any* vector, matrix, and univariate/bivariate polynomial, opening an new 'algebra-as-graph' perspective, broadening the applicability of Poly2Graph to a wide range of other areas."
> - (L1841-1845) "... remarkable generality of our framework. Since the characteristics of most scientific systems can be expressed as a matrix, polynomial, or more generically a vector ..., our approach offers a novel analytical lens. This opens up new avenues for research across numerous fields, a direction we are actively exploring and invite the broader community to join."
>
> Thus, Poly2Graph and HSG-12M **instantiate a general "algebra‑as‑graph" program that is directly applicable to many scientific domains** where ubiquitous algebraic objects are the fundamental representation.

---

> ### Author Response · Authors · 2025-11-25
> **Response to Reviewer K5Rg (Part 3/6)**
>
> ## 2. Why graphs instead of images or coefficients? (W2, Q1, Q2, Q4)
>
> > (W2) Unclear advantage of the graph representation: The authors do not compare to simpler baselines such as CNNs on spectral images, MLPs on polynomial coefficients, or models trained directly on spectral arrays. It remains unproven that representing spectra as graphs yields better or different information.\
> > (Q1, Q2, Q4) Why is a graph representation preferable to treating the spectra as 2‑D arrays or polynomial coefficients? Can you show any downstream task where learning on graphs leads to qualitatively different or improved results than learning on images? Would CNNs or transformers on spectral images achieve comparable or better accuracy?
>
> As explained above, the **spectral graph** is the physics object of interest. Image and coefficient representations play auxiliary roles in the pipeline and do not address the scientific question we care about.
>
> ### 2.1 Conceptual arguments: limitations of images and coefficients
>
> 1. **Spectral potential images are unphysical.**
>    The spectral potential (Ronkin function, $\Phi(E)$) used in [2,3] is a powerful theoretical shortcut from the infinite‑dimensional Hamiltonian to the spectral graph. However, $\Phi(E)$ itself is not a directly observable quantity; it is a mathematical device to compute the spectral graph. Training on $\Phi(E)$ images therefore trains models on an intermediate, non‑physical object instead of the physically meaningful spectral graph.
>
> 2. **DOS images are dominated by empty background.**
>    The DOS image ($\rho(E)$) is a 2‑D histogram of eigenvalues. The physically relevant information lies on the **1‑D loci of eigenvalues**, i.e., the spectral graph itself. The rest of the image is essentially zero DOS, providing negligible information but consuming most of the pixels. A CNN or ViT must spend capacity modeling this vast uninformative background instead of focusing on the graph geometry.
>
> 3. **Image representation is practically intractable at full scale.**
>    Storing HSG‑12M as images would require **177 TB** of storage. Training image models at this scale is virtually impossible with realistic compute budgets. The graph representation compresses this to **256 GB** while preserving the relevant structure, making large‑scale learning feasible.
>
> 4. **Coefficients → ChP class is trivial and scientifically uninteresting.**
>    The ChP class is defined by the *monomial support* and *parity structure* of the characteristic polynomial coefficients (Appendix B.3.4). **An MLP on coefficients would simply re‑learn this definition**, effectively performing a binarization/parity check. More importantly, such a model **would bypass the spectral graph entirely and thus ***not*** address the physics problem**: understanding and exploiting the geometry/topology of the spectrum. For this reason, we deliberately focus on learning from spectral graphs.

---

> ### Author Response · Authors · 2025-11-25
> **Response to Reviewer K5Rg (Part 4/6)**
>
> ### 2.2 New experiments: ResNet and ViT on spectral images
>
> To directly address W2/Q1/Q2/Q4, we implemented **image‑based baselines** (ResNet and ViT) on two image modalities and compared them to GNNs on graphs.
>
> #### 2.2.1 Experimental setup
>
> * We randomly sample **5 ChP classes** and downsample to **441 graphs per class** (total 2205 samples) to keep image experiments tractable. We use a stratified split of 70/15/15 for train/val/test and report results averaged over 3 random seeds.
> * The classification head (a 2‑layer MLP) is identical across *all* models for a fair comparison.
> * Unlike the constrained‑budget benchmark in the main paper, here we **do not cap the training budget**: each model trains until validation performance saturates, and we report test metrics at the best validation checkpoint.
> * ResNet and ViT architectures follow standard practice. We match the depth (number of conv blocks / transformer blocks) to the GNNs and tune only hidden dimensions to keep parameter counts comparable (but still much more generous for image models).
>
> #### 2.2.2 Results
>
> | modality | dataset | model | accuracy | macro-F1 | throughput (samples/sec) | peak GPU mem (Mb/sample) | # parameters |
> |:---|:---|:---|:---|:---|:---|:---|:---|
> | graph | $\mathcal{G}$ - HSG | CGCNN | 0.973 ± 0.008 | 0.973 ± 0.008 | 9664 ± 176 | 0.66 ± 0.03 | 14k |
> | graph | $\mathcal{G}$ - HSG | GATv2 | 0.953 ± 0.018 | 0.953 ± 0.018 | 8483 ± 26 | 0.59 ± 0.00 | 6k |
> | graph | $\mathcal{G}$ - HSG | GINE | 0.984 ± 0.002 | 0.984 ± 0.002 | 9529 ± 149 | 0.60 ± 0.03 | 6k |
> | graph | $\mathcal{G}$ - HSG | GraphSAGE | 0.971 ± 0.012 | 0.971 ± 0.012 | 10133 ± 162 | 0.59 ± 0.03 | 5k |
> | image | $\rho(E)$ - DOS | ResNet | 0.797 ± 0.018 | 0.797 ± 0.015 | 985 ± 2 | 26.61 ± 0.01 | 40k |
> | image | $\rho(E)$ - DOS | ViT | 0.775 ± 0.054 | 0.773 ± 0.054 | 882 ± 2 | 21.03 ± 0.00 | 39k |
> | image | $\Phi(E)$ - Potential | ResNet | 0.819 ± 0.024 | 0.818 ± 0.024 | 982 ± 3 | 26.64 ± 0.01 | 40k |
> | image | $\Phi(E)$ - Potential | ViT | 0.763 ± 0.005 | 0.742 ± 0.004 | 881 ± 3 | 21.06 ± 0.01 | 39k |
>
>
> #### 2.2.3 Discussion
>
> * On this controlled 5‑class subset, **graph models on spectral graphs outperform image models by ~15–20% absolute accuracy**, *even when* image models are given more parameters and unconstrained training budgets.
> * Graph models are also dramatically more efficient: they have **≈6× fewer parameters**, **≈10× higher throughput**, and **≈40× lower GPU memory footprint** than the ResNet/ViT baselines.
> * **These results match our intuitions**: images devote most of their capacity to meaningless zero‑DOS background, whereas the graph representation precisely encodes the eigenvalue loci and its geometry.
> * Importantly, without a budget constraint, GNNs reach very high Top‑1 accuracy (95–98%) on this subset, showing that the spectral‑graph representation and our featurization are **highly learnable when sufficient compute** is available.
>
> Together, these experiments provide direct empirical evidence that **learning on graphs is both more accurate and more efficient** than learning on spectral images for the same underlying task.

---

> ### Author Response · Authors · 2025-11-25
> **Response to Reviewer K5Rg (Part 5/6)**
>
> ## 3. Stability of Poly2Graph & Quality of HSG‑12M (W3, Q3)
>
> > (W3) Representation loss and stability: The extraction from spectra to graph skeletons may discard quantitative details and can fragment edges (acknowledged in Appendix H). No analysis quantifies how much information or robustness is lost.\
> > (Q3) How robust are the extracted graphs to numerical perturbations or thresholding choices?
>
> ### 3.1 Existing validation and realistic operating regime
>
> During development we **manually validated** Poly2Graph against ground‑truth spectra obtained via exact diagonalization on a few hundred characteristic polynomials. We ensured that the extracted spectral graphs visually match the energy spectra, including edge connectivity and overall geometry. In the manuscript we summarize:
>
> - (L255-260): "**Quality Assurance and Limitations.** We validated Poly2Graph on hundreds of characteristic polynomials, by visually checking that the spectral graph from Poly2Graph agrees with the energy spectrum from exact diagonalization. In rare complicated cases, numerical instabilities can still arise close to the junction nodes whose surrounding edges have extremely low DOS. Poly2Graph will attempt to mitigate such cases by merging nearby nodes and contracting edges shorter than a predefined tolerance."
>
> The reviewer's concern about "fragmented edges" refers to Appendix H, where we discuss what we call **component fragmentation**:
>
> - (L2507-2508) "Our extraction pipeline struggles when the hopping range or band number becomes large (e.g. **for pure theoretical interests that fall outside realistic domain**)"
>
> Crucially, this caveat is **not relevant** for HSG‑12M itself. HSG‑12M is sampled around *realistic Hamiltonian families* with limited hopping range and band number. **We only note this for users who may want to apply Poly2Graph to ***extreme*** cases that are beyond physically realistic domains**.
>
> ### 3.2 New robustness experiments: threshold perturbations
>
> To quantify robustness (Q3), we stress‑tested the `short_edge_threshold` hyperparameter used by Poly2Graph to automatically mitigate component fragmentation by merging small fragmented regions into junction nodes to restore connectivity (**Appendix C.4**).
>
> #### 3.2.1 Experimental setup
>
> - We randomly sample **20 characteristic polynomials** from HSG‑12M. For each, we use the spectral graph computed with the default threshold $\tau = 40$ (the value used for HSG‑12M) as the **reference graph**.
> - For each polynomial, we recompute the spectral graph at **20 perturbed thresholds** $\bar{\tau} \in [0.5\tau, 2\tau]$, log‑spaced over this 4× range.
> - We then test **multigraph isomorphism** between each perturbed graph and its reference graph at $\tau$.
>
> This yields 20 (polynomials) × 20 (thresholds) = **400** perturbed graphs.
>
> #### 3.2.2 Results
>
> Out of 400 perturbed graphs, 391 (**97.75%**) are isomorphic to their reference graphs:
>
> | Default threshold | Perturbed thresholds | Isomorphic matches |
> |---|---|---|
> | 40 ($\tau$) | [20, 80] ($0.5\tau\sim 2\tau$) | 391/400 = 97.75% |
>
> This demonstrates that Poly2Graph's extracted topology is **highly stable** under a 4× sweep of the main threshold parameter for the realistic Hamiltonian families included in HSG‑12M.
>
> ---
>
> In addition, **as with widely used synthetic datasets such as QM9 or OGB**, small numerical artifacts from simulation and processing are inevitable. But these are **an accepted trade‑off** for obtaining *large‑scale, diverse datasets* that have nevertheless **enabled substantial scientific and methodological advances**. We view HSG‑12M in the same spirit: a carefully controlled synthetic dataset whose small imperfections are outweighed by its scale and richness.

---

> ### Author Response · Authors · 2025-11-25
> **Response to Reviewer K5Rg (Part 6/6)**
>
> ## 4. Benchmark insights & ML usefulness (W5, W6)
>
> > (W5) Benchmark insight is shallow: Standard GNNs achieve modest accuracies, but this mainly reflects task complexity and training budget, not necessarily new modeling challenges.\
> > (W6) Incremental as a benchmark contribution: The work is an engineering milestone rather than a conceptual one. Its usefulness for advancing ML is not clear.
>
> We agree that our primary contribution is the **dataset and generator**, and we are careful not to over‑claim on the benchmarking side: (L2512-2514) "Our contribution centers on the dataset and its generator; the benchmark is intended to catalyze follow-up work. We invite the community to perform comprehensive, large-scale, and carefully designed evaluations."
>
> The authors are not frontier experts in GNN model design and with limited compute resource, but we have exhausted the best practices and resources available to provide a solid benchmark.
>
> That said, we respectfully disagree that the benchmark is not useful for advancing ML:
> - **Edge‑awareness is key.**
>   Our results clearly reveal a **failure mode of standard, edge‑agnostic GNNs**: models that explicitly incorporate edge attributes (e.g., GINE, CGCNN) in general significantly outperform edge‑agnostic baselines (GCN, GIN). This highlights the importance of edge features for spatial multigraphs with rich geometric structure.
> - **A testbed for spatial multigraphs.**
>   Spatial multigraphs arise naturally in many domains (e.g., river networks, brain vasculature, vascular systems), yet **there are no benchmarks for this regime**. HSG‑12M provides such a testbed, allowing the ML community to **prototype geometry‑aware GNNs** and training strategies on abundant synthetic data before moving to scarce real‑world datasets.
>
> Finally, "conceptual novelty" in model design is limited by construction. We use standard GNNs **because no spatial multigraph GNNs currently exist**. Apart from standard GNNs, we have included models designed for spatial graphs (CGCNN, MF) to endeavor a more comprehensive benchmark.
>
> ---
>
> Beyond the dataset and pipeline itself, we:
> - Provide a **self‑contained pedagogical Appendix B** that explains the physics background and spectral‑graph construction assuming only basic solid state physics knowledge, arguably more approachable than all existing specialized physics literature.
> - Carefully optimize the end‑to‑end extraction pipeline and design the `Poly2Graph` package for **usability, scalability, and extensibility**, with a detailed tutorial in **Appendix G**.
>
> We believe such large‑scale, well‑documented benchmarks are essential infrastructure for the community, enabling both physics‑driven applications and ML method development.
>
> ---
>
> ## *References*
> 1. Tai, T. & Lee, C. H. Zoology of non-Hermitian spectra and their graph topology. *Phys. Rev. B* 107, L220301 (2023).
> 2. Xiong, Y. & Hu, H. Graph morphology of non-Hermitian bands. *Phys. Rev. B* 109, L100301 (2024).
> 3. Wang, H.-Y., Song, F. & Wang, Z. Amoeba Formulation of Non-Bloch Band Theory in Arbitrary Dimensions. *Phys. Rev. X* 14, 021011 (2024).
> 4. Wang, A. & Chen, C. Q. Observing non-Bloch braids and phase transitions by precise manipulation of the non-Hermitian boundary and size. *Commun Phys* 8, 294 (2025).
> 5. Bradlyn, B. et al. Topological quantum chemistry. *Nature* 547, 298–305 (2017).
> 6. Vergniory, M. G. et al. A complete catalogue of high-quality topological materials. *Nature* 566, 480–485 (2019).
> 7. Hu, H. & Zhao, E. Knots and Non-Hermitian Bloch Bands. *Phys. Rev. Lett.* 126, 010401 (2021).
> 8. Wang, K., Dutt, A., Wojcik, C. C. & Fan, S. Topological complex-energy braiding of non-Hermitian bands. *Nature* 598, 59–64 (2021).
> 9. Bernevig, B. A., Hughes, T. L. & Zhang, S.-C. Quantum Spin Hall Effect and Topological Phase Transition in HgTe Quantum Wells. *Science* 314, 1757–1761 (2006).
> 10. König, M. et al. Quantum Spin Hall Insulator State in HgTe Quantum Wells. *Science* 318, 766–770 (2007).

---

> ### Author Response · Authors · 2025-11-28
> **Follow-up on Discussion and Summary of Responses**
>
> Dear Reviewer K5Rg,
>
> As the discussion period is nearing its end, we would like to gently follow up and briefly recap how we addressed your main concerns in our rebuttal:
>
> * **Physical meaning & scientific value (W1, W4, Q5).**
>   We clarified that the *spectral graph* is a rigorously defined, topology-bearing object established in recent non-Hermitian band theory, and framed our classification task as a surrogate for *inverse design* (from desired/observed spectral graph back to Hamiltonian families) which is *unresolved, highly non-trivial* and of central interest in physics, with potential applications beyond non-Hermitian physics to general vector, matrix, and polynomial systems.
>
> * **Why graphs (vs images/coefficients) & additional baselines (W2, Q1, Q2, Q4).**
>   We argued that DOS/spectral-potential images are auxiliary and extremely storage-inefficient (full HSG-12M images would require ~177 TB) and that coefficients essentially encode the class definition itself. We added ResNet/ViT baselines on image representations (5-class subset) and found GNNs on graphs achieve ~15–20% higher accuracy with far fewer parameters and much lower memory/compute cost.
>
> * **Robustness and stability of Poly2Graph (W3, Q3).**
>   We showed that problematic “fragmentation” arises only in extreme, non-realistic regimes and reported a new stress test over a 4× range of the key threshold, where ~97.75% of graphs remain multigraph-isomorphic to the reference, indicating strong robustness of the extracted topology.
>
> * **Benchmark insights and ML usefulness (W5, W6).**
>   We highlighted that edge-aware GNNs consistently outperform edge-agnostic ones on our spatial multigraphs and that HSG-12M fills a gap as a large-scale benchmark for spatial multigraphs, which are common in nature yet absent in existing benchmarks.
>
> If there is anything further we could clarify or add that would be helpful for your final evaluation, we would be very grateful for your guidance.

---

### Meta-Review · Area_Chair_ut1D · 2026-01-07

**Summary:**

Unclear scientific and task-level significance: While the scale and technical achievement of the dataset are clear, the motivation for why this learning task is scientifically or methodologically meaningful was initially insufficiently articulated.

Justification of the graph representation: The advantage of representing spectra as graphs was not initially validated through comparisons with simpler representations such as images or polynomial coefficients.

Robustness of the extraction process: Concerns were raised about potential information loss and topological instability in the spectrum-to-graph conversion, including sensitivity to numerical perturbations and threshold choices

Limited modeling novelty: The primary contribution lies in the dataset and generation pipeline, while the work provides limited insight into new modeling challenges or architectural advances specific to spatial multi-graphs.

Benchmark design limitations: Although multi-edge geometry is a central feature of the data, the reference benchmarks reduce this geometry to fixed-size summary features, which underutilize the geometric richness of the dataset.

Synthetic nature of the data: The dataset is entirely algorithmically generated with limited direct connection to experimental measurements or existing real-world multi-graph domains.

Reference accuracy: At least one cited reference was flagged as potentially nonexistent, requiring verification.

**Reviewer Concerns:**

Scientific and task-level significance: The rebuttal clarified the physical meaning and measurability of spectral graphs and reframed the classification task as a surrogate for inverse design. This addressed the core concern about task motivation, though its broader appeal remains a matter of judgment.

Justification of the graph representation: The rebuttal addressed this by adding conceptual arguments and new image-based baseline, which showed higher accuracy and efficiency for graph-based models.

Robustness of the extraction process: The rebuttal added quantitative stability tests that demonstrates high topological consistency under threshold perturbations and clarified that fragmentation issues arise mainly outside the dataset’s realistic regime.

Modeling novelty and benchmark depth: The rebuttal acknowledged that the main contribution is the dataset and generator and articulated challenges specific to spatial multigraphs. The lack of dedicated spatial-multigraph models remains an open issue rather than one resolved in this work.

Use of geometric richness in benchmarks: The rebuttal explicitly recognized that rich multi-edge geometry is compressed into fixed-size features in the reference benchmarks, which frames this as a baseline limitation and future research opportunity.

Synthetic nature of the data: The rebuttal contextualized the lack of experimental data by discussing measurability and theory-first precedents, but the dataset itself remains fully synthetic.

Reference accuracy: One of the references does not appear to exist in the cited venue. But a closely related paper exists:
Anwar Said et al., “NeuroGraph: Benchmarks for Graph Machine Learning in Brain Connectomics” (arXiv:2306.06202, also appears as a NeurIPS 2023 paper).
The authors need to fix the reference accordingly.

**Reviewer Scores:**

Overall, the rebuttal would likely reinforce the evaluations of initially positive reviewers.
Borderline reviewers would likely become more favorable, as concerns about physical meaning, task motivation, representation choice, and robustness were directly addressed.
In contrast, the objection regarding the fundamental scientific or ML value of the work  would likely remain.

---

### Decision · Program_Chairs · 2026-01-26

Accept (Poster)